

# Boundary SymTFT

**Lakshya Bhardwaj, Christian Copetti, Daniel Pajer and Sakura Schäfer-Nameki**

Mathematical Institute, University of Oxford, Andrew Wiles Building,
Woodstock Road, Oxford, OX2 6GG, UK

## Abstract

We study properties of boundary conditions (BCs) in theories with categorical (or non-invertible) symmetries. We describe how the transformation properties, or (generalized) charges, of BCs are captured by topological BCs of Symmetry Topological Field Theory (SymTFT), which is a topological field theory in one higher spacetime dimension. As an application of the SymTFT characterization, we discuss the symmetry properties of boundary conditions for (1+1)d gapped and gapless phases. We provide a number of concrete examples in spacetime dimensions $d = 2, 3$. We furthermore expand the lattice description for (1+1)d anyon chains with categorical symmetries to include boundary conditions carrying arbitrary 1-charges under the symmetry.

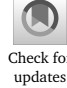

# 1   Introduction

Non-invertible or categorical symmetries [1–14] (for recent reviews with more complete references on this topic see [15, 16]) have hugely resonated in the realm of high energy theory, condensed matter theory, and mathematics. Exploring their implications has become one of the key challenges and exciting directions in these fields.

One – by now well-documented – implication is their imprint on the phase structure of quantum systems, constraining both gapped and gapless phases. A particularly powerful tool to systematically explore the phase diagrams dictated by categorical symmetries has been the Symmetry Topological Field Theory (SymTFT) [17–20] also known as topological holography [21–25]. Categorical symmetries have many intricate features: starting with the multiplets of generalized charges, which e.g. contain both genuine and non-genuine operators [26–30], they lead to a new, categorical Landau paradigm [31], including gapped and gapless phases [2, 3, 31–43] and phase transitions [22, 24, 44, 45]. The SymTFT has played a central role in clarifying the conceptual foundation and supplying a systematic computational framework to address these exciting questions.

Much of these developments thus far have focused on theories on spacetimes without boundaries. As is well-documented, in both gapped and gapless phases with standard, group-symmetries, the symmetry transformations of boundary conditions provide useful insights into their physics, classifying possible boundary conditions, but also constraining RG-flows in the presence of boundaries. A classic example that goes beyond group-symmetries is the set of topological defect (Verlinde) lines in rational conformal field theories (CFTs), which act non-trivially on the boundary conditions, i.e. Cardy states, of the CFT [34, 46–48]. Another well-studied case for group symmetries are gapped phases, in particular SPTs, which are known to have characteristic edge modes. Recent works on the study of categorical symmetries in the presence of boundaries have appeared in [49, 50] and some of the results in the present paper were given a sneak preview in [51, 52].

The main goal of this paper is to develop the description of boundary conditions and more generally interfaces/defects, in the presence of categorical symmetries. Crucial to this is the SymTFT, which in order to capture the transformation properties of the BCs under the symmetry has an additional (gapped) boundary. We will refer to this extension of the SymTFT as the **Boundary SymTFT**, see figure 1 and for a simplified version that we will use throughout the paper, see figure 2. In particular we show that BCs are determined in terms of so-called generalized charges, more precisely, $(d-1)$-charges for a theory in $d$ spacetime dimensions.

To explain this in more detail, recall that the SymTFT $\mathfrak{Z}(\mathcal{S})$ of a theory $\mathfrak{T}_d$ in $d$ dimensions with symmetry given by a fusion $(d-1)$-category $\mathcal{S}$, has the property that compactifying it on an interval with a gapped boundary condition $\mathfrak{B}^{\mathrm{sym}}_{\mathcal{S}}$ and a physical boundary condition $\mathfrak{B}^{\mathrm{phys}}_{\mathfrak{T}_d}$, gives back the original theory $\mathfrak{T}_d$ with the symmetry action of $\mathcal{S}$. One should think of the symmetry boundary as the location, where the symmetry topological defects of $\mathcal{S}$ are localized. All the dynamics of the theory are confined to the physical boundary. The symmetry action on the theory is realized in terms of the topological defects of the SymTFT – some will realize the symmetry, others the charges. In particular **generalized $(p-1)$-charges** are defined in terms of $p$-dimensional defects $Q_p$ of the SymTFT, that stretch between the two boundaries [29]. As with ordinary symmetries, they organize into multiplets under the symmetry $\mathcal{S}$.

A particular subset of such defects are the $(d-1)$-charges, obtained by stretching $d$-dimensional topological defects of the SymTFT, $Q_d$, between $\mathfrak{B}^{\mathrm{sym}}$ to $\mathfrak{B}^{\mathrm{phys}}$. Due to their dimensionality, these correspond to **boundary conditions**. This provides the first crucial identification:

$(d-1)$-charges $Q_d$ of $\mathcal{S}$ are in 1-1 correspondence with
topological BCs of the SymTFT, admitting topological interfaces with $\mathfrak{B}^{\mathrm{sym}}_{\mathcal{S}}$.

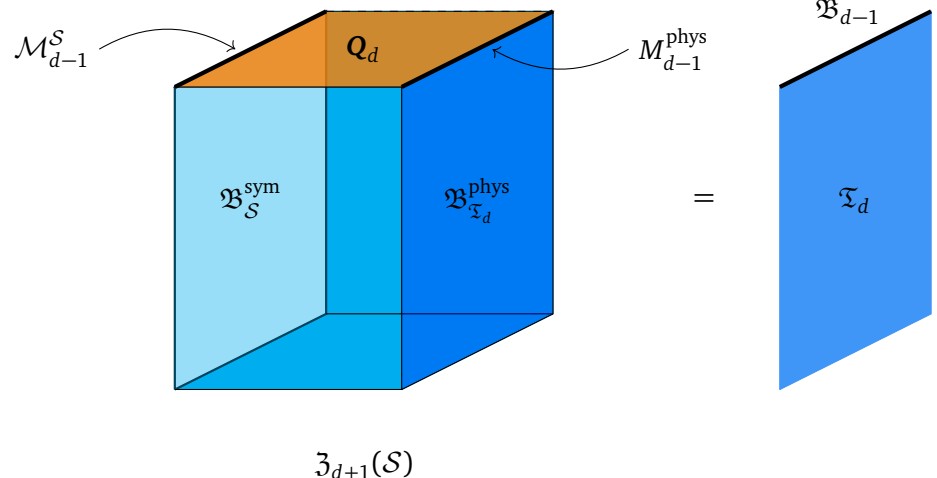

$$\mathfrak{Z}_{d+1}(\mathcal{S})$$

Figure 1: 3d depiction of the Boundary SymTFT: The bulk SymTFT is given in terms of the $(d+1)$-dimensional TQFT $\mathfrak{Z}_{d+1}(\mathcal{S})$ and the two boundary conditions: the gapped symmetry boundary $\mathfrak{B}^{\mathrm{sym}}_{\mathcal{S}}$ and (not necessarily gapped) physical boundary $\mathfrak{B}^{\mathrm{phys}}_{\mathfrak{T}_d}$. The boundary SymTFT extends this to another gapped boundary condition – indicated here by $\boldsymbol{Q}_d$, which has interfaces $\mathcal{M}^{\mathcal{S}}_{d-1}$ and $M^{\mathrm{phys}}_{d-1}$ with the other two boundary conditions. The interval compactification shown on the right hand side, results in a theory $\mathfrak{T}_d$ with boundary $\mathfrak{B}_{d-1}$. From the perspective of generalized charges of the symmetry $\mathcal{S}$, the SymTFT description of boundary conditions is in terms of the ends of $\boldsymbol{Q}_d$, i.e. so-called $(d-1)$-charges.

Given the bulk SymTFT and its topological defects (the Drinfeld center of $\mathcal{S}$), we can determine the $\mathcal{S}$-symmetry properties by studying the defects $\boldsymbol{Q}_d$, and their interfaces with the $\mathfrak{B}^{\mathrm{sym}}$ and $\mathfrak{B}^{\mathrm{phys}}$ boundaries. The SymTFT description allows immediately a generalization to interfaces which are $(d-1)$-charges as well: an interface $\mathcal{I}$ between a theory with symmetry $\mathcal{S}$ and another with symmetry $\mathcal{S}'$, can be mapped to a gapped BC for $\mathfrak{Z}_{d+1}(\mathcal{S}) \boxtimes \overline{\mathfrak{Z}_{d+1}(\mathcal{S}')}$, by folding.

The interfaces between the two gapped boundary conditions of the SymTFT, $\boldsymbol{Q}_d$ and $\mathfrak{B}^{\mathrm{sym}}_{\mathcal{S}}$, can also be thought of as $(d-1)$-module categories $\mathcal{M}$ for $\mathcal{S}$. From this point of view, the topological defects localized on the $(d-1)$-charge $\boldsymbol{Q}_d$ form the category $\mathcal{S}^*_{\mathcal{M}}$, i.e. the category of $\mathcal{S}$-endofunctors $\mathcal{M} \to \mathcal{M}$. In down to earth terms this means it is the category of symmetries that one obtains after gauging $\mathcal{M}$ in $\mathcal{S}$.

Equipped with this formulation of the symmetry transformation properties of boundary conditions, we apply this to fusion category symmetries and fusion 2-categories, i.e. boundary conditions in the presence of symmetries in (1+1)d and (2+1)d. We also provide explicit computations of boundary multiplets under fusion category symmetries $\mathcal{S}$ in (1+1)d, both group-like and non-invertible.

As mentioned just above, concrete examples in this paper rely on fusion 1-categories ($d = 2$) and fusion 2-categories ($d = 3$). For $d = 2$ the necessary notions—fusion categories, module categories, dual (convolution) categories and Drinfeld centres—are classical [53, 54]. For $d = 3$ a rigorous framework is now available: fusion 2-categories were introduced in [55], while module 2-categories and their duals are developed in [56]; key structural results such as Morita theory and internal Homs appear in [57]. For $n \geq 3$ no fully general definition of fusion $n$-category, module $n$-category or Drinfeld center has yet been completed. Existing examples—e.g. the fusion 3-categories for duality defects constructed in [58]—form a special



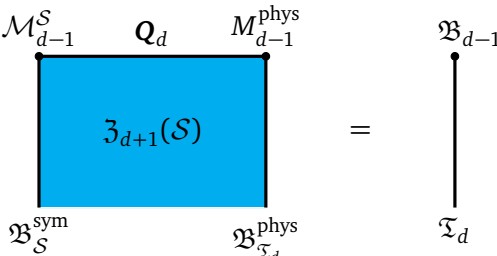

Figure 2: Simplified depiction of the Boundary SymTFT used throughout the paper. Note that the interface between the $(d-1)$-charge $\boldsymbol{Q}_d$ and $\mathfrak{B}^{\text{sym}}$ is a $(d-1)$-module category. The same applies to the interface between $\boldsymbol{Q}_d$ and $\mathfrak{B}^{\text{phys}}$ in case the theory is a gapped phase.

subclass obtained by iterative condensation and do not settle issues of strictness, pivotality or sphericality in full generality. Accordingly, any statement in this paper that involves fusion $n$-categories with $n > 2$ should be regarded as conjectural, motivated by physical considerations and analogy with the well-established $n \leq 2$ cases.

A particularly interesting application is to gapped phases in (1+1)d, for which we have a description in terms of the SymTFT detailed in [38], by requiring the physical boundary to also be a gapped boundary. In this case the interfaces between $\mathfrak{B}^{\text{sym}}$ and $\boldsymbol{Q}_2$ as well as $\boldsymbol{Q}_2$ and $\mathfrak{B}^{\text{phys}}$ are both given in terms of module categories, $\mathcal{M}$ and $\mathcal{N}$, respectively. The symmetry along the $\boldsymbol{Q}_2$ gapped boundary is the symmetry obtained after gauging $\mathcal{M}$, i.e. $\mathcal{S}^*_{\mathcal{M}}$, which acts on $\mathcal{M}$ from the right and $\mathcal{N}$ from the left. Each boundary condition is then labeled by $\mathfrak{B}^{m,n}$, where $m \in \mathcal{M}$ and $n \in \mathcal{N}$. Let us point out that, for gapped phases, this proposal describes *enriched* boundary conditions, which are in general decomposable. Indecomposable boundary conditions are described via a wedge compactification [59]. We expand on this remark in section 2.3.

We can furthermore study boundary changing operators, as shown in figure 3. Given an interface $\mathfrak{B}^m$ between $\mathfrak{B}^{\text{sym}}$ and $\boldsymbol{Q}_2$ labeled by $m \in \mathcal{M}$, a symmetry defect can end on the module category and potentially change $\mathfrak{B}^m$ to $\mathfrak{B}^{m'}$, if $\text{Hom}(S \otimes m, m')$ is non-empty for $v \in \mathcal{S}^*_{\mathcal{M}}$.

In section 3.7 we apply this paradigm to gapless phases, generalizing the club sandwich construction of [44]. This generalizes the SymTFT description of the so-called Kennedy-Tasaki transformation to include boundary conditions and gives constraints for their identification along certain RG flows. We consider the boundary multiplets for intrinsically gapless SPTs (igSPTs) [24, 25, 45, 60–66], exemplifying this for the $\mathbb{Z}_4$ igSPT.

Finally, we extend the lattice models with fusion category symmetries to include boundary conditions. For this we start with the anyon chain models [67–75]. The boundary conditions are encoded in 1-charges $\boldsymbol{Q}_2$, which in turn correspond to module categories $\mathcal{M}$. These provide a simple description for allowed boundaries in the anyon chain.

## 2 Boundary SymTFT and generalized charges

### 2.1 Generalized charges of boundary conditions and interfaces

Let $\mathcal{S}$ be a fusion $(d-1)$-category and $\mathfrak{T}_d$ be an $\mathcal{S}$-symmetric $d$-dimensional QFT. The observables of $\mathfrak{T}_d$ organize themselves in terms of representations (or charges) under the symmetry

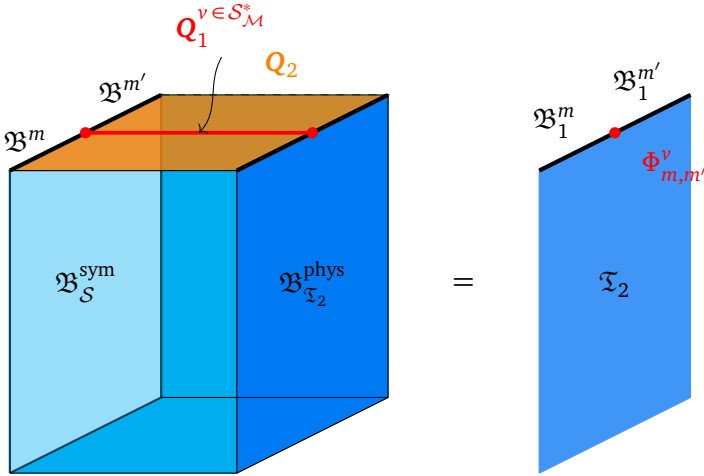

Figure 3: Boundary changing operators in (1+1)d. The interface between $\mathfrak{B}^{\text{sym}}$ and $Q_2$ (shown in orange) is the module category $\mathcal{M}$, with $m, m' \in \mathcal{M}$. The boundary changing operator is 0-charge $Q_1^v$, where $v \in \mathcal{S}_{\mathcal{M}}^*$, which stretches between $\mathcal{M}$ and $M^{\text{phys}}$. The interaction points become the boundary changing operator $\Phi$ after interval compactification.

$\mathcal{S}$. For observables of codimension bigger than one, the possible charges have been studied in [29]. Here we extend their analysis to include charges for observables of codimension-1, which include in particular the boundary conditions of $\mathfrak{T}_d$. The charges of $p$-dimensional observables for $0 \leq p \leq d-2$ were referred to as $p$-charges in [29]. Extending their terminology, we will refer to the **charges of boundary conditions as $(d-1)$-charges**.

**The SymTFT sandwich.** As argued in [29] the charges are simply encoded in the topological defects of the associated $(d+1)$-dimensional SymTFT $\mathfrak{Z}_{d+1}(\mathcal{S})$. Let us recall their construction, which we will extend to describe the SymTFT encoding of $(d-1)$-charges.

An $\mathcal{S}$-symmetric theory $\mathfrak{T}_d$ can be constructed as an interval compactification of the SymTFT $\mathfrak{Z}_{d+1}(\mathcal{S})$ with two BCs placed at the ends of the interval. On the left end, we place the symmetry BC $\mathfrak{B}_{\mathcal{S}}^{\text{sym}}$, which is topological, and on the right end we place a BC $\mathfrak{B}_{\mathfrak{T}_d}^{\text{phys}}$ which contains dynamical information about the physical theory (and may or may not be topological, depending on $\mathfrak{T}_d$). This will be schematically drawn as follows

$$
\begin{array}{cc}
\boxed{\mathfrak{Z}_{d+1}(\mathcal{S})} & = \quad \Big| \qquad . \\
\mathfrak{B}_{\mathcal{S}}^{\text{sym}} \qquad \mathfrak{B}_{\mathfrak{T}_d}^{\text{phys}} & \qquad \mathfrak{T}_d \; \circlearrowright \; \mathcal{S}
\end{array}
\tag{1}
$$

The topological defects of the symmetry boundary $\mathfrak{B}_{\mathcal{S}}^{\text{sym}}$ form the fusion $(d-1)$-category $\mathcal{S}$. Their image under the interval compactification endows $\mathfrak{T}_d$ with a collection of topological defects characterized by the category $\mathcal{S}$. The choice of these topological defects of $\mathfrak{T}_d$ is a choice of $\mathcal{S}$-symmetry of the theory $\mathfrak{T}_d$. This construction is referred to as the SymTFT **sandwich construction**.



**Generalized charges of defects.** For $0 \leq p \leq d-2$, any $p$-dimensional operator $\mathcal{O}_p$ of $\mathfrak{T}_d$ is constructed in the sandwich as described below:

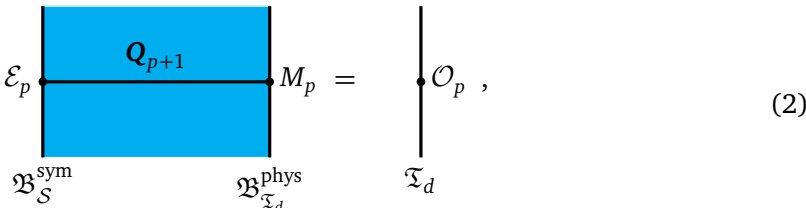

$$\tag{2}$$

where $\mathcal{E}_p$ is a topological end along $\mathfrak{B}_{\mathcal{S}}^{\mathrm{sym}}$ of a topological $(p+1)$-dimensional defect $\boldsymbol{Q}_{p+1}$ of $\mathfrak{Z}_{d+1}(\mathcal{S})$, and $M_p$ is an end of $\boldsymbol{Q}_{p+1}$ along $\mathfrak{B}_{\mathfrak{T}_d}^{\mathrm{phys}}$. The end $M_p$ is topological or non-topological depending on whether the operator $\mathcal{O}_p$ is topological or non-topological.

This construction of $\mathcal{O}_p$ reveals a lot about how it interacts with the symmetry $\mathcal{S}$, in particular fully characterizing its $p$-charge. First of all, note that cycling through all possible ends $\mathcal{E}_p$ of $\boldsymbol{Q}_{p+1}$ while keeping the 'dynamical' end $M_p$ fixed gives rise to a collection of $p$-dimensional operators of $\mathfrak{T}_d$. We label this collection also as $M_p$ for brevity and refer to it as a multiplet of $p$-dimensional operators. The $\mathcal{S}$-symmetry is isolated in the sandwich construction in terms of the topological defects of $\mathfrak{B}_{\mathcal{S}}^{\mathrm{sym}}$. These naturally only interact with the ends $\mathcal{E}_p$ of $\boldsymbol{Q}_{p+1}$, in particular permuting them into each other. This action of $\mathcal{S}$ on $\mathcal{E}_p$ descends directly to an action of $\mathcal{S}$ on the multiplet $M_p$ of operators in $\mathfrak{T}_d$. As $\mathcal{O}_p$ is one of the operators in the multiplet $M_p$, this fully describes the action of $\mathcal{S}$ on $\mathcal{O}_p$. Note that the action is fully characterized in terms of how the topological defect $\boldsymbol{Q}_{p+1}$ appearing in the sandwich construction of $\mathcal{O}_p$ can end along the topological BC $\mathfrak{B}_{\mathcal{S}}^{\mathrm{sym}}$, and hence the $p$-charge of $\mathcal{O}_p$, or more precisely of the multiplet $M_p$ in which $\mathcal{O}_p$ lives, is captured by the $(p+1)$-dimensional topological defect $\boldsymbol{Q}_{p+1}$ of the SymTFT $\mathfrak{Z}_{d+1}(\mathcal{S})$. This establishes an identification between $p$-charges of symmetry $\mathcal{S}$ and $(p+1)$-dimensional topological defects of the SymTFT $\mathfrak{Z}_{d+1}(\mathcal{S})$.

**Generalized charges of BCs.** This can be extended to the case of $(d-1)$-charges of boundary conditions under the symmetry $\mathcal{S}$, and we find that such $(d-1)$-charges can be identified with $d$-dimensional topological boundary conditions of the SymTFT $\mathfrak{Z}_{d+1}(\mathcal{S})$. Said differently, this allows us to characterize the generalized charges under the symmetry $\mathcal{S}$ of such boundary conditions. The argument is quite similar to above. Let $\mathfrak{B}_{d-1}$ be a BC of $\mathfrak{T}_d$. Its sandwich construction takes the familiar form already shown in figures 1 and 2.

The boundary condition in the SymTFT is modeled there as follows: $\boldsymbol{Q}_d$ is a topological BC of $\mathfrak{Z}_{d+1}(\mathcal{S})$, $\mathcal{M}_{d-1}$ is a topological interface between $\mathfrak{B}_{\mathcal{S}}^{\mathrm{sym}}$ and $\boldsymbol{Q}_d$, and $M_{d-1}$ is an interface between $\boldsymbol{Q}_d$ and $\mathfrak{B}_{\mathfrak{T}_d}^{\mathrm{phys}}$. The interface $M_{d-1}$ is topological or non-topological depending on whether the BC $\mathfrak{B}_{d-1}$ is topological or non-topological.

Cycling through all possible topological interfaces $\mathcal{M}_{d-1}$ between $\boldsymbol{Q}_d$ and $\mathfrak{B}_{\mathcal{S}}^{\mathrm{sym}}$ while keeping the 'dynamical' interface $M_{d-1}$ fixed gives rise to a collection of BCs of $\mathfrak{T}_d$. We label this collection also as $M_{d-1}$ for brevity and refer to it as a **multiplet of BCs**. The $\mathcal{S}$-symmetry is isolated in the sandwich construction in terms of the topological defects of $\mathfrak{B}_{\mathcal{S}}^{\mathrm{sym}}$. These naturally only interact with the topological interfaces $\mathcal{M}_{d-1}$, in particular permuting them into each other. This action of $\mathcal{S}$ on $\mathcal{M}_{d-1}$ descends directly to an action of $\mathcal{S}$ on the multiplet $M_{d-1}$ of BCs of $\mathfrak{T}_d$. As $\mathfrak{B}_{d-1}$ is one of the BCs in the multiplet $M_{d-1}$, this fully describes the action of $\mathcal{S}$ on $\mathfrak{B}_{d-1}$. Note that the action is fully characterized in terms of how the topological BC $\boldsymbol{Q}_d$ appearing in the sandwich construction of $\mathfrak{B}_{d-1}$ interacts with the other topological BC $\mathfrak{B}_{\mathcal{S}}^{\mathrm{sym}}$, and hence the $(d-1)$-charge of $\mathfrak{B}_{d-1}$, or more precisely of the multiplet $M_{d-1}$ in which $\mathfrak{B}_{d-1}$ lives, is captured by the $d$-dimensional topological BC $\boldsymbol{Q}_d$ of the SymTFT $\mathfrak{Z}_{d+1}(\mathcal{S})$. This establishes an identification between:

- $(d-1)$-charges of BCs under the symmetry $\mathcal{S}$

- Topological BCs of the SymTFT $\mathfrak{Z}_{d+1}(\mathcal{S})$ admitting a topological interface[1] with the symmetry boundary $\mathfrak{B}_{\mathcal{S}}^{\mathrm{sym}}$.

**Module categories and Lagrangians.** The topological BCs of $\mathfrak{Z}_{d+1}(\mathcal{S})$ admitting a topological interface with $\mathfrak{B}_{\mathcal{S}}^{\mathrm{sym}}$ are characterized by module $(d-1)$-categories of fusion $(d-1)$-category $\mathcal{S}$. Let $\mathcal{M}$ be such a module $(d-1)$-category characterizing a BC $\boldsymbol{Q}_d$. The simple objects $m$ of $\mathcal{M}$ describe the irreducible interfaces $\mathcal{M}_{d-1}^m$ between $\mathfrak{B}_{\mathcal{S}}^{\mathrm{sym}}$ and $\boldsymbol{Q}_d$ so the Boundary SymTFT can be written as

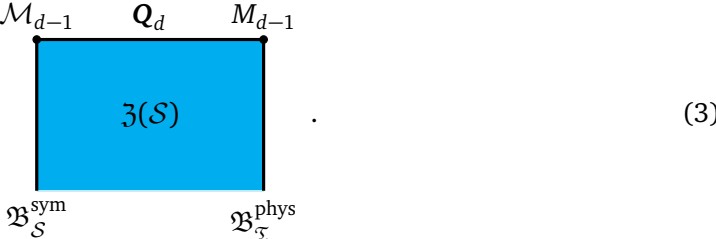

$$\tag{3}$$

Each simple $m \in \mathcal{M}$ gives rise to a boundary condition. These interfaces are acted upon by the topological defects of $\mathfrak{B}_{\mathcal{S}}^{\mathrm{sym}}$ comprising the fusion $(d-1)$-category $\mathcal{S}$, capturing the action of $\mathcal{S}$ on its module category $\mathcal{M}$. Similarly the BCs of the theory $\mathfrak{T}_d$ will also carry labels $m$ and form multiplets under the symmetry $\mathcal{S}$.

An interesting point to note for $d \geq 3$ is that two simple modules $m, m' \in \mathcal{M}$ may not be related by a direct action of $\mathcal{S}$, but instead by a discrete gauging operation. That is, the interface $\mathcal{M}_{d-1}^{m'}$ is obtained from $\mathcal{M}_{d-1}^m$ by gauging topological defects living on $\mathcal{M}_{d-1}^m$, which comprise the fusion $(d-2)$-category of endomorphisms of $m \in \mathcal{M}$.

A further important remark regards the preservation of bulk symmetry by boundary conditions. In (1+1)d a symmetry is preserved if all the topological lines $D_1$ admit topological boundary-preserving junctions, see the in depth discussion in [49]. In higher dimensions a similar idea holds true for extended topological operator. However, if the symmetry is not preserved, it will typically happen that bulk topological defects $D_p$ will descend to non-trivial boundary topological defects $\hat{D}_p$ of the same dimension. This will be relevant in discussing BC in (2+1)D.

Specializing to $d = 2$, such module categories are, in turn, in one-to-one correspondence with Lagrangian algebras $\mathcal{L}$ of the Drinfeld center $\mathcal{Z}(\mathcal{S})$. The Lagrangians define 1-charges $\boldsymbol{Q}_2$ and we will often indicate the 1-charge by a superscript for the associated Lagrangian algebra $\mathcal{L}$. The number of irreducible BCs in a multiplet transforming in a 1-charge is given by the number of isomorphism classes of simple objects in the corresponding module category, which is also the number of linearly independent pairs of ends for topological line defects $\boldsymbol{Q}_1$ of $\mathfrak{Z}(\mathcal{S})$ along the boundaries $\mathfrak{B}^{\mathrm{sym}}$ and $\boldsymbol{Q}_2$ as shown in the following figure:

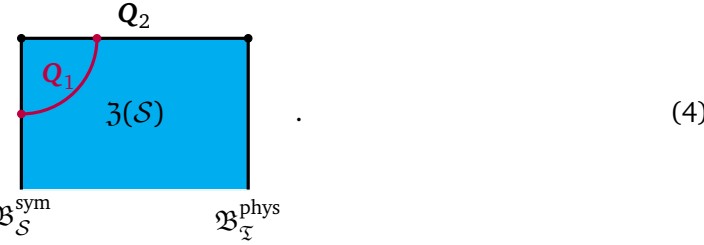

$$\tag{4}$$

---

[1] In $d \geq 3$, not all pairs of topological BCs of a SymTFT admit such a topological interface.

These are readily obtained from the expressions for Lagrangian algebras

$$\mathcal{L}_{\mathfrak{B}^{\text{sym}}_{\mathcal{S}}} = \bigoplus_a n_a \mathbf{Q}_1^a \,,$$
$$\mathcal{L}_{\mathbf{Q}_2} = \bigoplus_a n'_a \mathbf{Q}_1^a \,,$$

(5)

for $\mathfrak{B}^{\text{sym}}_{\mathcal{S}}$ and $\mathbf{Q}_2$ as

$$|\mathcal{M}| = \sum_a n_a n'_a \,.$$

(6)

**$d$-category of $(d-1)$-charges.** We can repeat the above arguments for sub-defects of boundary conditions. Consider a possibly non-topological interface $\mathcal{I}_{d-2}$ between two boundary conditions $\mathfrak{B}_{d-1}$ and $\mathfrak{B}'_{d-1}$ of an $\mathcal{S}$-symmetric $d$-dimensional theory $\mathfrak{T}_d$. If $\mathfrak{B}_{d-1}$ and $\mathfrak{B}'_{d-1}$ lie respectively in multiplets transforming in $(d-1)$-charges $\mathbf{Q}_d$ and $\mathbf{Q}'_d$. As we discussed above $\mathbf{Q}_d$ and $\mathbf{Q}'_d$ are topological BCs of the SymTFT $\mathfrak{Z}_{d+1}(\mathcal{S})$. Then, $\mathcal{I}_{d-2}$ lies in a multiplet transforming under $\mathcal{S}$ in a $(d-2)$-charge $\mathbf{Q}_{d-1}$ which can be identified with a topological interface between the topological BCs $\mathbf{Q}_d$ and $\mathbf{Q}'_d$ of $\mathfrak{Z}_{d+1}(\mathcal{S})$.

In this way, the whole $d$-category formed by topological BCs of $\mathfrak{Z}(\mathcal{S})$ realizes generalized charges of BCs and their sub-defects of an $\mathcal{S}$-symmetric $d$-dimensional theory. This $d$-category can be identified with the $d$-category formed by module $(d-1)$-categories of $\mathcal{S}$, and is often denoted as $\text{Mod}(\mathcal{S})$.

**$(d-1)$ Charges of interfaces and defects.** So far we have only discussed how a particular type of $(d-1)$-dimensional defects, namely BCs, transform under a fusion $(d-1)$-category symmetry $\mathcal{S}$. One can consider other $(d-1)$-dimensional defects similarly. Consider an interface $\mathcal{I}_{d-1}$ between an $\mathcal{S}$-symmetric $d$-dimensional theory $\mathfrak{T}_d$ and an $\mathcal{S}'$-symmetric $d$-dimensional theory $\mathfrak{T}'_d$. Such an interface lies in a multiplet whose $(d-1)$-charge under $(\mathcal{S}, \mathcal{S}')$ is captured by a topological interface $\mathbf{Q}_d$ between the corresponding SymTFTs $\mathfrak{Z}_{d+1}(\mathcal{S})$ and $\mathfrak{Z}_{d+1}(\mathcal{S}')$. By folding, these can be identified with topological BCs of the $(d+1)$-dimensional TFT $\mathfrak{Z}_{d+1}(\mathcal{S}) \boxtimes \overline{\mathfrak{Z}_{d+1}(\mathcal{S}')}$. Such interfaces are for instance relevant the context of gapless phases, as we will discuss in section 3.7

Thus, even though these $(d-1)$-charges seem to be a generalization of the $(d-1)$-charges associated to BCs, their study is actually subsumed within the study of $(d-1)$-charges of BCs. It is for this reason that we focus on $(d-1)$-charges of BCs only, as from them one can understand the structure of $(d-1)$-charges of interfaces.

A particular case of the above is worth mentioning. If we take $\mathfrak{T}'_d = \mathfrak{T}_d$ and $\mathcal{S}' = \mathcal{S}$, then $\mathcal{I}_{d-1}$ are codimension-1 defects of $\mathfrak{T}_d$. Their $(d-1)$-charges under $\mathcal{S}$ are captured by topological codimension-1 defects of the SymTFT $\mathfrak{Z}_{d+1}(\mathcal{S})$, or equivalently topological BCs of the *doubled SymTFT* $\mathfrak{Z}_{d+1}(\mathcal{S}) \boxtimes \overline{\mathfrak{Z}_{d+1}(\mathcal{S})}$.

## 2.2 Examples in (1+1)d and (2+1)d

**0-form symmetry group in (1+1)d.** For a non-anomalous 0-form symmetry $G$ in (1+1)d, which is described by fusion category $\mathcal{S} = \text{Vec}_G$, the module categories are classified by a subgroup $H \subseteq G$ and a class $\beta \in H^2(H, U(1))$ (see e.g. [76]). A multiplet of BCs transforming in the 1-charge corresponding to $(H, \beta)$ comprises of BCs labeled by $H$-cosets $\mathfrak{B}_H$. The action of $G$ permutes the boundaries according to how it acts from the left on such cosets. We can depict the action of the symmetry on the 1d boundary conditions as follows (here we show

only the boundaries and the symmetry action on them):

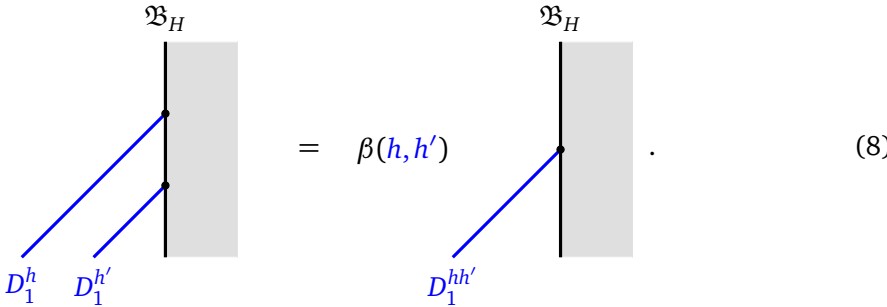

$$(7)$$

In particular, the BC $\mathfrak{B}_H$ corresponding to the identity coset is left invariant by $H$, i.e. the topological line operators $D_1^h$ generating symmetries $h \in H$ can end along $\mathfrak{B}_H$, and induce an $H$ 0-form symmetry along the 1d boundary. The class $\beta$ captures the 't Hooft anomaly for this induced 0-form symmetry and is captured by the following relationship

$$
= \quad \beta(h, h') \qquad \qquad . \tag{8}
$$

**Non-invertible 0-form symmetry in (1+1)d.** For non-Abelian finite groups $G$ the representation category is a non-invertible symmetry $\mathsf{Rep}(G)$. The module categories for $\mathsf{Rep}(G)$ are again labeled by $(H, \beta)$ and we denote them by

$$
\mathcal{M}_{(H,\beta)} = \mathsf{Rep}^\beta(H). \tag{9}
$$

The number of boundary conditions associated to this module category is $|\mathcal{M}_{(H,\beta)}| = |\mathsf{Rep}(H)|$.

Let us apply this to the simplest example of a non-invertible representation category in (1+1)d, namely

$$
\mathcal{S} = \mathsf{Rep}(S_3), \tag{10}
$$

with simple lines $1, P, E$ and fusion $PE = EP = E$ and $P^2 = 1$ and $E^2 = 1 + P + E$. In this case, there are four module categories. The indecomposable module categories are

$$
\begin{aligned}
\mathcal{M}_{\mathsf{Vec}} &= \mathsf{Vec}, \\
\mathcal{M}_{\mathrm{reg}} &= \mathsf{Rep}(S_3), \\
\mathcal{M}_{\mathbb{Z}_2} &= \mathsf{Rep}(\mathbb{Z}_2), \\
\mathcal{M}_{\mathbb{Z}_3} &= \mathsf{Rep}(\mathbb{Z}_3).
\end{aligned} \tag{11}
$$

These are in one-to-one correspondence with Lagrangian algebras of $\mathcal{Z}(\mathcal{S})$. For $\mathsf{Rep}(S_3)$, using the notation in [38] for the simple topological defects of the SymTFT $\boldsymbol{Q}_{[g],R}$, the Lagrangians are

$$
\begin{aligned}
\mathcal{L}_{\mathsf{Vec}} &= \boldsymbol{Q}_{[\mathrm{id}],1} \oplus \boldsymbol{Q}_{[\mathrm{id}],P} \oplus 2\boldsymbol{Q}_{[\mathrm{id}],E}, \\
\mathcal{L}_{\mathrm{reg}} &= \boldsymbol{Q}_{[\mathrm{id}],1} \oplus \boldsymbol{Q}_{[a],1} \oplus \boldsymbol{Q}_{[b],+}, \\
\mathcal{L}_{\mathbb{Z}_2} &= \boldsymbol{Q}_{[\mathrm{id}],1} \oplus \boldsymbol{Q}_{[\mathrm{id}],E} \oplus \boldsymbol{Q}_{[b],+}, \\
\mathcal{L}_{\mathbb{Z}_3} &= \boldsymbol{Q}_{[\mathrm{id}],1} \oplus \boldsymbol{Q}_{[\mathrm{id}],P} \oplus 2\boldsymbol{Q}_{[a],1}.
\end{aligned} \tag{12}
$$

Each such Lagrangian $\mathcal{L}_a$ defines a gapped BC or a 1-charge $\boldsymbol{Q}_2^a$. Note that the symmetry boundary in this case is

$$
\mathsf{B}_{\mathsf{Rep}(S_3)}^{\mathrm{sym}} : \quad \boldsymbol{Q}_2^{\mathrm{reg}}. \tag{13}
$$

The boundary conditions are then:

1. $\mathcal{M} = \text{Vec}$: This is the $\text{Rep}(S_3)$-invariant boundary, and corresponds to the 1-charge $Q_2^{\text{Vec}}$: here we show the 1d boundary (as a line) and the action of the symmetry on it:

$$\Big| \quad \circlearrowleft \text{Rep}(S_3) \ . \tag{14}$$

In more detail, if we call the BC as $\mathfrak{B}$, then we have the actions

$$\begin{aligned} P \otimes \mathfrak{B} &= \mathfrak{B}\,, \\ E \otimes \mathfrak{B} &= \mathfrak{B} \oplus \mathfrak{B}\,. \end{aligned} \tag{15}$$

2. $\mathcal{M}_{\text{reg}} = \text{Rep}(S_3)$: The regular module, which corresponds to the algebra $\mathcal{L}_{\text{Rep}(S_3)}$ gives rise to three boundary conditions, arising from the lines $Q_{[\text{id}],1}$, $Q_{[a],1}$, $Q_{[b],+}$ stretching between $\mathfrak{B}^{\text{sym}}$ and $Q_2^{\text{reg}}$, which are acted upon by $\text{Rep}(S_3)$ as follows

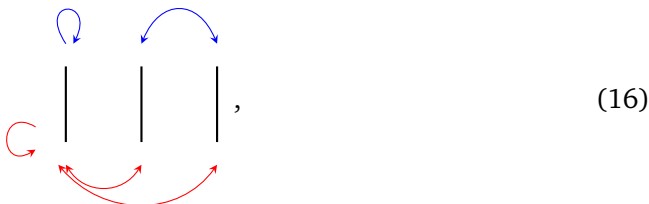

$$\tag{16}$$

where blue indicates the action of the $\mathbb{Z}_2$ and red the action of the non-invertible line $E$. In more detail, we can label the three BCs as $\mathfrak{B}_1$, $\mathfrak{B}_P$ and $\mathfrak{B}_E$, and the action of $\text{Rep}(S_3)$ symmetry on the three boundaries simply follows the $\text{Rep}(S_3)$ fusion rules, e.g.

$$E \otimes \mathfrak{B}_E = \mathfrak{B}_1 \oplus \mathfrak{B}_P \oplus \mathfrak{B}_E\,. \tag{17}$$

3. $\mathcal{M}_{\mathbb{Z}_2} = \text{Rep}(\mathbb{Z}_2)$: The module category $\text{Rep}(\mathbb{Z}_2)$ associated to the Lagrangian algebra $\mathcal{L}_{\mathbb{Z}_2}$ gives rise to two boundary conditions with the $\text{Rep}(S_3)$ action given by

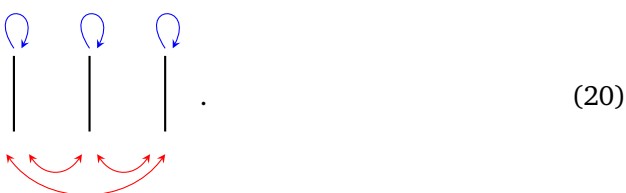

$$\tag{18}$$

In more detail, if we label the two BCs as $\mathfrak{B}_\pm$, then we have

$$\begin{aligned} P \otimes \mathfrak{B}_\pm &= \mathfrak{B}_\mp\,, \\ E \otimes \mathfrak{B}_\pm &= \mathfrak{B}_+ \oplus \mathfrak{B}_-\,. \end{aligned} \tag{19}$$

4. $\mathcal{M}_{\mathbb{Z}_3} = \text{Rep}(\mathbb{Z}_3)$: This module category gives rise to boundary conditions labeled by the lines stretching between $\mathfrak{B}^{\text{sym}}$ and $Q_2^{\mathbb{Z}_3}$, which are $2 \times Q_{[a],1}$ and $Q_{[\text{id}],1}$ including multiplicities as they appear in the associated Lagrangians,

$$\tag{20}$$

In more detail, if we label the three BCs as $\mathfrak{B}_i$ for $i \in \{0, 1, 2\}$, then we have

$$
\begin{aligned}
P \otimes \mathfrak{B}_i &= \mathfrak{B}_i \,, \\
E \otimes \mathfrak{B}_i &= \mathfrak{B}_{i+1} \oplus \mathfrak{B}_{i+2} \,,
\end{aligned}
\tag{21}
$$

where the subscripts are defined modulo 3. Note that it is no coincidence that the action of the symmetry on the boundary condition multiplets follows precisely the action on gapped phases (associated to the same module categories or Lagrangians).

**$\mathbb{Z}_2$ 0-form symmetry in (2+1)d.** Gapped boundary conditions for the SymTFTs of fusion 2-category symmetries were studied recently in [41], in particular for the case we will discuss as an example here: a $\mathbb{Z}_2$ 0-form symmetry with fusion 2-category description

$$
\mathcal{S} = 2\mathrm{Vec}_{\mathbb{Z}_2} \,.
\tag{22}
$$

The corresponding SymTFT $\mathfrak{Z}(2\mathrm{Vec}_{\mathbb{Z}_2})$ is the (3+1)d $\mathbb{Z}_2$ Dijkgraaf-Witten theory, characterized by a $\mathbb{Z}_2$ surface operator $Q_2$ and a $\mathbb{Z}_2$ (magnetic) line operator $Q_1$ with nontrivial mutual linking:

$$
\mathbf{B}_{Q_2 Q_1} = -1 \,.
\tag{23}
$$

Similarly to the case of standard (non-topological) boundary conditions [77, 78], a crucial distinction with respect to the (1+1)d case is the presence of both minimal and non-minimal boundary conditions. The former correspond to the well known Dirichlet and Neumann boundary conditions, whereas the latter can be enriched by a non-trivial TFT on their world-volume. As we will see, this effect propagates to the physical boundary conditions, which will host various types of Fusion Categories on their worldvolume. This will give rise to *minimal* and *non-minimal* multiplets.

The minimal topological boundary conditions for $\mathfrak{Z}(2\mathrm{Vec}_{\mathbb{Z}_2})$ are of two types:

- **Minimal Dirichlet gapped BC $\mathfrak{B}_{\mathbf{Dir}}$:** corresponding to the condensation of $Q_1$ lines. The $Q_2$ surface on the $\mathfrak{B}_{\mathrm{Dir}}$ gapped BC becomes the $\mathbb{Z}_2$ generator $D_2^P$ of the $\mathbb{Z}_2$ zero-form symmetry.

- **Minimal Neumann gapped BC $\mathfrak{B}_{\mathbf{Neu}, \omega}$:** these can be described by gauging the boundary $\mathbb{Z}_2$ symmetry with discrete torsion $\omega \in H^3(\mathbb{Z}_2, U(1)) = \mathbb{Z}_2$, corresponding to the two possible choices of algebras including $Q_2$ surfaces. The boundary symmetry is a $\mathbb{Z}_2$ one-form symmetry $2\mathrm{Rep}(\mathbb{Z}_2)$ generated by the projection $D_1^{\widehat{P}}$ of the bulk $Q_1$ lines. We comment on the relevance of the twist $\omega$ for boundary conditions below.

Fixing $\mathfrak{B}^{\mathrm{sym}} = \mathfrak{B}_{\mathrm{Dir}}$, cycling through minimal $\mathfrak{B}_{\mathrm{Dir}}$ and $\mathfrak{B}_{\mathrm{Neu}, \omega}$ BC can be used to describe the symmetric BCs (2-charges) under $\mathbb{Z}_2^{(0)}$. We can organize these into Doublet or Singlets depending on whether the $\mathbb{Z}_2$ 0-form symmetry is preserved or broken by the boundary condition, respectively.

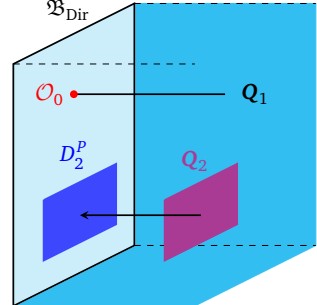
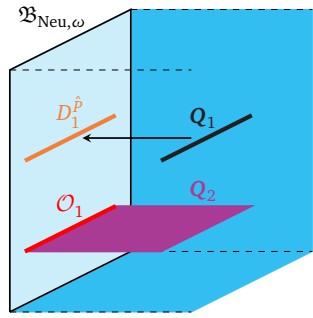

Figure 4: Visualized: Minimal gapped boundary conditions of $\mathfrak{Z}(2\mathrm{Vec}_{\mathbb{Z}_2})$.

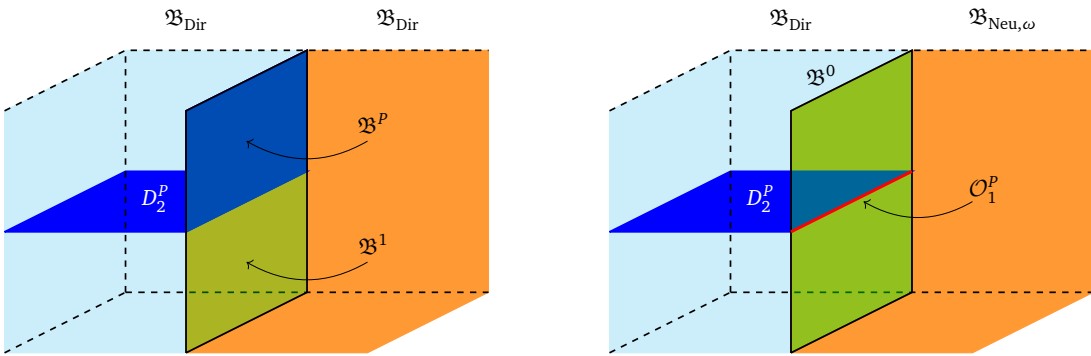

Figure 5: Minimal doublet of BCs (left) and singlet BC (right) for $\mathbb{Z}_2$ 0-form symmetry in (2+1)d.

- **Minimal Doublet BC:** This is a doublet of two BCs exchanged by $\mathbb{Z}_2$ action, such that there are no non-trivial topological line defects living on either of the two BCs. It is described by an interface between two $\mathfrak{B}_{\text{Dir}}$ gapped BCs, possibly decorated by a $\mathbb{Z}_2$ surface $D_2^P$. It also corresponds to the Regular module category for $2\mathsf{Vec}_{\mathbb{Z}_2}$.

- **Minimal Singlet BCs:** There are two such singlets, described by interfaces between $\mathfrak{B}_{\text{Dir}}$ and $\mathfrak{B}_{\text{Neu},\pm}$, respectively. For each of them there is a single BC $\mathfrak{B}$, which is invariant under $\mathbb{Z}_2$, so that the $\mathbb{Z}_2$ generating surface $D_2^P$ can end on it. The topological line $D_1^P$ arising at the end of $D_2^P$ along $\mathfrak{B}$ may or may not have a non-trivial F-symbol, described by $\omega$. The two minimal BCs thus differ by the $(1+1)$d anomaly of the induced $\mathbb{Z}_2$ symmetry.

In addition to the minimal BCs, there are infinitely many non-minimal ones, descending from the non-minimal topological BCs of the SymTFT studied in [41]. These fall in two classes, either stacking the Dirichlet gapped BC with a (2+1)d irreducible TFT:

$$\mathfrak{B}_{\text{Dir}}^{\mathfrak{T}} = \mathfrak{B}_{\text{Dir}} \boxtimes \mathfrak{T}, \tag{24}$$

or by stacking the minimal Dirichlet gapped BC with a $\mathbb{Z}_2^{(0)}$-symmetric TFT $\mathfrak{T}_{\mathbb{Z}_2}$ and gauging the diagonal symmetry:

$$\mathfrak{B}_{\text{Neu}}^{\mathfrak{T}_{\mathbb{Z}_2}} = \frac{\mathfrak{B}_{\text{Dir}} \boxtimes \mathfrak{T}_{\mathbb{Z}_2}}{\mathbb{Z}_2^{(0)}}. \tag{25}$$

Notice that the label $\omega$ becomes spurious, since the discrete torsion can be incorporated by stacking the (2+1)d $\mathbb{Z}_2$ SPT. This distinction extends to 2-charges. as we now explain in detail. Again we can organize them into doublet and singlet types:

- **Non-Minimal Doublet BCs:** These are described by a topological interface between $\mathfrak{B}_{\text{Dir}}$ and $\mathfrak{B}_{\text{Dir}}^{\mathfrak{T}}$. In order for the junction to exist $\mathfrak{T}$ must admit a gapped BC, that is $\mathfrak{T} = \mathfrak{Z}(\mathcal{C})$ for some fusion category $\mathcal{C}$. The corresponding module 2-category $\mathcal{M}$ has simple objects that appear in pairs. Let us pick such a pair. The two simple objects in this pair are indistinguishable, and the endomorphisms of each are described by the fusion category $\mathcal{C}$. Physically, this pair of objects corresponds to two BCs that are exchanged under the $\mathbb{Z}_2$ action, such that each BC carries topological line defects forming the fusion category $\mathcal{C}$.

Fusion categories $\mathcal{C}$ and $\mathcal{C}'$ associated to two pairs of simple objects in $\mathcal{M}$ must be Morita equivalent, that is $\mathcal{Z}(\mathcal{C}) = \mathcal{Z}(\mathcal{C}')$. Physically, a doublet of BCs corresponding to one pair can be obtained from a doublet of BCs corresponding to another pair by a generalized

gauging operation confined to the interface. An example is $\mathcal{C} = \mathsf{Vec}_{S_3}$ and $\mathcal{C}' = \mathsf{Rep}(S_3)$. The minimal doublet is obtained for $\mathcal{C} = \mathsf{Vec}$, for which we have $\mathcal{M} = 2\mathsf{Vec}_{\mathbb{Z}_2}$ or the regular module 2-category.

- **Non-minimal Singlet BCs:** Corresponding to topological interfaces between $\mathfrak{B}_{\mathrm{Dir}}$ and $\mathfrak{B}_{\mathrm{Neu}}^{\mathfrak{T}_{\mathbb{Z}_2}}$. Again the condition for the interface to exist forces $\mathfrak{T}_{\mathbb{Z}_2} = \mathfrak{Z}(\mathcal{C}_0)$ to be the center of a fusion category $\mathcal{C}_0$ with a $\mathbb{Z}_2$ action. The module 2-category $\mathcal{M}$ can have many simple objects, but each simple object appears as a $\mathbb{Z}_2$ singlet. Pick such a simple object and let $\mathfrak{B}$ be the corresponding BC. Since $\mathfrak{B}$ is left invariant by the $\mathbb{Z}_2$ action, the surface $D_2^P$ generating the $\mathbb{Z}_2$ symmetry can end along it. The information of the simple object is captured by a $\mathbb{Z}_2$-graded fusion category $\mathcal{C}$ whose trivial grade $\mathcal{C}_0$ describes the fusion category formed by genuine topological lines living on $\mathfrak{B}$, and the non-trivial grade $\mathcal{C}_1$ describes non-genuine topological lines living on $\mathfrak{B}$ that arise at an end of $D_2^P$ along $\mathfrak{B}$. This ties to the SymTFT description, as the gauging of the $\mathbb{Z}_2$ symmetry in $\mathfrak{Z}(\mathcal{C}_0)$ describes the center of a $\mathbb{Z}_2$-graded fusion category, as explained essentially in [79,80], and the diagonal gauging identifies the $\mathbb{Z}_2$ grading with the ending of $D_2^P$ surface defects.

  BCs $\mathfrak{B}$ and $\mathfrak{B}'$ corresponding to two different simple objects in $\mathcal{M}$ are related by gaugings of genuine topological lines, i.e. those contained in $\mathcal{C}_0$. Such a gauging process modifies the full $\mathbb{Z}_2$-graded fusion category $\mathcal{C}$.

  The minimal singlets are obtained for $\mathcal{C} = \mathsf{Vec}_{\mathbb{Z}_2}$ and $\mathcal{C} = \mathsf{Vec}_{\mathbb{Z}_2}^{\omega}$, which describe the only simple objects for module 2-categories whose underlying 2-category is $\mathcal{M} = 2\mathsf{Vec}$ but with two different module structures for $\mathcal{S} = 2\mathsf{Vec}_{\mathbb{Z}_2}$. Other interesting examples include $\mathcal{C} = \mathsf{Ising}$, or, more generally, $\mathcal{C} = \mathsf{TY}(G)$ for any Abelian group $G$.

**$\mathbb{Z}_2$ 1-form symmetry in (2+1)d.** Now consider non-anomalous $\mathbb{Z}_2$ 1-form symmetry in 3d, for which the symmetry fusion 2-category is

$$\mathcal{S} = 2\mathsf{Rep}(\mathbb{Z}_2), \tag{26}$$

which corresponds to the choice $\mathfrak{B}_{\mathrm{sym}} = \mathfrak{B}_{\mathrm{Neu}}$ The possible 2-charges again can be divided into minimal and non-minimal, each of which are of two types [41], depending on whether the $\mathbb{Z}_2$ one-form symmetry is preserved by the BC or not:

- **Minimal $\mathbb{Z}_2^{(1)}$-preserving BC:** Described by the topological interface between $\mathfrak{B}_{\mathrm{Neu}}$ and $\mathfrak{B}_{\mathrm{Dir}}$. Here we have a single BC $\mathfrak{B}$ on which the line $D_1^P$ generating $\mathbb{Z}_2^{(1)}$ can end topologically. There are no non-trivial topological lines living on $\mathfrak{B}$ and the bulk one-form symmetry is preserved on $\mathfrak{B}$. Charged boundary lines are described by $\mathbb{Z}_2^{(0)}$ surfaces extending on the $\mathfrak{B}_{\mathrm{Dir}}$ side and ending on the gapped interface between $\mathfrak{B}_{\mathrm{Dir}}$ and $\mathfrak{B}_{\mathrm{Neu}}$.

- **Minimal $\mathbb{Z}_2^{(1)}$-breaking BC:** Corresponding to the topological interface between $\mathfrak{B}_{\mathrm{Neu}}$ and $\mathfrak{B}_{\mathrm{Neu},\omega}$. On this BC the $D_1^P$ line becomes a nontrivial topological line $\hat{D}_1^{\ P}$. This topological line may or may not have non-trivial F-symbol, due to anomaly inflow from the second Neumann boundary condition. In other words, $\hat{D}_1^{\ P}$ form a fusion category $\mathcal{C} \in \{\mathsf{Vec}_{\mathbb{Z}_2}, \mathsf{Vec}_{\mathbb{Z}_2}^{\omega}\}$. These two give rise to distinct minimal 2-charges. As the line $D_1^P$ cannot terminate of $\mathfrak{B}$, the one-form symmetry is broken by the BC.

- **Minimal BCs without charged boundary lines:** Described by the topological interface between $\mathfrak{B}_{\mathrm{Neu}}$ and $\mathfrak{B}_{\mathrm{Dir}}$. Here we have a single BC $\mathfrak{B}$ on which the line $D_1^P$ generating $\mathbb{Z}_2^{(1)}$ can end topologically. There are no non-trivial topological lines living on $\mathfrak{B}$ and the bulk one-form symmetry is preserved on $\mathfrak{B}$.

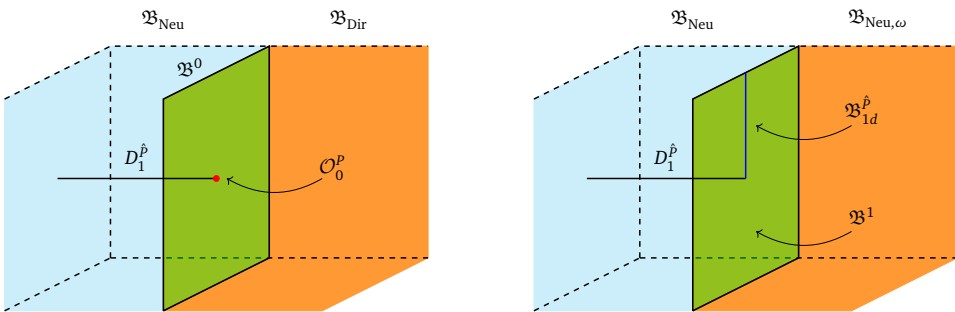

Figure 6: Minimal singlet BC $\mathfrak{B}^0$ (left) and doublet of BCs $\mathfrak{B}^1$, $\mathfrak{B}^{\hat{P}}$ (right) for $\mathbb{Z}_2$ 1-form symmetry in (2+1)d. Note that $\mathfrak{B}^{\hat{P}}_{1d}$ is the BC $\mathfrak{B}^{\hat{P}}$ compactified to a line.

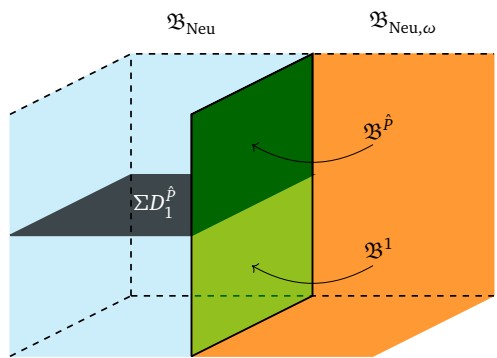

Figure 7: Minimal doublet of BCs for $\mathbb{Z}_2$ 1-form symmetry in (2+1)d. Note that $\Sigma D^{\hat{P}}_1$ is a 2d condensation defect of $D^{\hat{P}}_1$ symmetry lines in $\mathfrak{B}_{\text{Neu}}$.

- **Minimal BCs with charged boundary lines:** Corresponding to the topological interface between $\mathfrak{B}_{\text{Neu}}$ and $\mathfrak{B}_{\text{Neu},\omega}$. On this BC $\mathfrak{B}$ on which the line $D^P_1$ can end, but now we have another order two topological line $D^{\widehat{P}}_1$ living on $\mathfrak{B}$, under which the end $D^P_0$ of $D^P_1$ is charged, or equivalently $D^{\widehat{P}}_1$ is charged under $D^P_1$.

  There are two possibilities for $D^{\widehat{P}}_1$ according to whether or not it carries a trivial F-symbol. In other words, $D^{\widehat{P}}_1$ form a fusion category $\mathcal{C} \in \{\text{Vec}_{\mathbb{Z}_2}, \text{Vec}^{\omega}_{\mathbb{Z}_2}\}$. These two give rise to distinct minimal 2-charges.

  For the case of $\mathcal{C} = \text{Vec}^{\omega}_{\mathbb{Z}_2}$, this is the only possible BC in the multiplet, but for $\mathcal{C} = \text{Vec}_{\mathbb{Z}_2}$ we have another BC $\mathfrak{B}'$ in the multiplet obtained by gauging the line $D^{\widehat{P}}_1$. The bulk line $D^P_1$ no longer ends on $\mathfrak{B}'$ and can be projected to a non-trivial line living on $\mathfrak{B}'$. There are no other topological lines on $\mathfrak{B}'$.

The non-minimal generalizations are as follows:

- **Non-minimal $\mathbb{Z}^{(1)}_2$-preserving BC:** Given by the interface between $\mathfrak{B}_{\text{Neu}}$ and $\mathfrak{B}^{\mathfrak{T}}_{\text{Dir}}$. Here we have a module 2-category $\mathcal{M}$, such that if we pick any simple object of it, then the corresponding BC $\mathfrak{B}$ has the property that the line $D^P_1$ generating $\mathbb{Z}^{(1)}_2$ can end on it topologically. Moreover, there may be non-trivial topological lines living on $\mathfrak{B}$, forming a fusion category $\mathcal{C}$, which is related to $\mathfrak{T}$ by $\mathfrak{T} = \mathfrak{Z}(\mathcal{C})$. All such lines are completely invisible to $D^P_1$. Picking any other simple object of $\mathcal{M}$ we obtain the same structure with some other fusion category $\mathcal{C}'$ Morita equivalent to $\mathcal{C}$. The two BCs $\mathfrak{B}$ and $\mathfrak{B}'$ are related by gaugings of these topological lines localized along the boundary which are uncharged under $\mathbb{Z}^{(1)}_2$.

The minimal 2-charge discussed above has $\mathcal{M} = 2\mathsf{Vec}$ with a single simple object corresponding $\mathcal{C} = \mathsf{Vec}$.

- **Non-minimal $\mathbb{Z}_2^{(1)}$-breaking BC:** Described by the interface between $\mathfrak{B}_{\mathrm{Neu}}$ and $\mathfrak{B}_{\mathrm{Neu}}^{\mathfrak{T}_{\mathbb{Z}_2}}$. In this case the symmetry line $D_1^P$ is mapped to an element $\hat{D}_1 \in \mathcal{C}_1$, which forms the nontrivially-graded part of a $\mathbb{Z}_2$-graded fusion category:

$$\mathcal{C} = \mathcal{C}_0 \oplus \mathcal{C}_1 \,, \tag{27}$$

and

$$\frac{\mathfrak{T}_{\mathbb{Z}_2}}{\mathbb{Z}_2} = \mathfrak{Z}(\mathcal{C}) \,. \tag{28}$$

Discrete gaugings inside $\mathcal{C}_0$ give rise to different simple objects in $\mathcal{M}$. The minimal 2-charges discussed above correspond to $\mathcal{C} = \mathcal{C}_1 \in \{\mathsf{Vec}_{\mathbb{Z}_2}, \mathsf{Vec}_{\mathbb{Z}_2}^{\omega}\}$ with $\mathcal{C}_0 = \mathsf{Vec}$. As these $\mathcal{C}$ are not Morita equivalent, indeed the corresponding boundaries transform in two different 2-charges.

## 2.3 Gapped phases with boundaries

An interesting point is that BCs carrying arbitrary $(d-1)$-charges can be realized by gapped boundaries in each $\mathcal{S}$-symmetric gapped phases of spacetime dimension $d$, provided that the gapped phase admits at least one gapped boundary. This is, for example, in sharp contrast with what happens in (1+1)d CFTs. Notably diagonal unitary minimal models $M_n$ support boundary conditions which only transform in the Regular representation of the symmetry [81]. Another important examples are strongly interacting QFTs, in which even the standard Dirichlet BC–which typically spontaneously breaks some symmetry– might not exists at strong coupling [82, 83]. As we now describe, however, this property of gapped phases comes at a cost: the study of decomposable BC. Finally, there is a clear physical interpretation of the decomposability/enrichment: the BC studied in this section can be mapped to interfaces between different gapped phases–one of which is the bulk gapped phase we consider here– the enrichment of the BC is equivalent to the existence of multiple such interfaces.

While the condition regarding the existence of a gapped boundary is true for arbitrary phases in (1+1)d, it not true in general for higher $d$. For example, topologically ordered phases in (2+1)d do not admit gapped boundaries unless the MTC describing the phase is the Drinfeld center of a fusion category. A simple example is provided by an Ising MTC.

Let us now consider an $\mathcal{S}$-symmetric $d$-dimensional gapped phase $\mathfrak{T}_d$ that does admit at least one gapped boundary $\mathfrak{B}_{d-1}$. This means that we have a boundary SymTFT construction involving three topological BCs $\mathfrak{B}_{\mathcal{S}}^{\mathrm{sym}}$, $\mathfrak{B}_{\mathfrak{T}_d}^{\mathrm{phys}}$ and $Q_d$ of $\mathfrak{Z}_{d+1}(\mathcal{S})$, where $Q_d$ captures the $(d-1)$-charged of $\mathfrak{B}_{d-1}$. This implies that there are topological interfaces between $\mathfrak{B}_{\mathcal{S}}^{\mathrm{sym}}$ and $Q_d$, and between $Q_d$ and $\mathfrak{B}_{\mathfrak{T}_d}^{\mathrm{phys}}$. Combining these facts, we learn that there are topological interfaces between $\mathfrak{B}_{\mathcal{S}}^{\mathrm{sym}}$ and $\mathfrak{B}_{\mathfrak{T}_d}^{\mathrm{phys}}$. In other words, $\mathfrak{B}_{\mathfrak{T}_d}^{\mathrm{phys}}$ is gauge related to $\mathfrak{B}_{\mathcal{S}}^{\mathrm{sym}}$, i.e. $\mathfrak{B}_{\mathfrak{T}_d}^{\mathrm{phys}}$ can be obtained by gauging topological defects living on $\mathfrak{B}_{\mathcal{S}}^{\mathrm{sym}}$ and vice versa.

Similarly, a topological BC $Q_d'$ capturing an arbitrary $(d-1)$-charge is gauge related to $\mathfrak{B}_{\mathcal{S}}^{\mathrm{sym}}$ since there are topological interfaces between $Q_d'$ and $\mathfrak{B}_{\mathcal{S}}^{\mathrm{sym}}$ by definition. This means that $Q_d'$ and $\mathfrak{B}_{\mathfrak{T}_d}^{\mathrm{phys}}$ are also gauge related, and hence there are topological interfaces between them.

In conclusion, we can construct boundary SymTFT configurations involving arbitrary $(d-1)$-charge $\boldsymbol{Q}_d$ with $\mathfrak{B}_{\mathcal{S}}^{\text{sym}}$ and $\mathfrak{B}_{\mathfrak{T}_d}^{\text{phys}}$ fixed:

$$
\begin{array}{ccc}
\mathcal{M} \quad \boldsymbol{Q}_d \quad \mathcal{N} & & \{\mathfrak{B}^{m,n}\} \\
\boxed{\mathfrak{Z}_{d+1}(\mathcal{S})} \quad = & & \bigg| \\
\mathfrak{B}_{\mathcal{S}}^{\text{sym}} \qquad \mathfrak{B}_{\mathfrak{T}_d}^{\text{phys}} & & (\mathfrak{B}_{\mathcal{S}}^{\text{sym}}|\mathfrak{B}_{\mathfrak{T}_d}^{\text{phys}})
\end{array}
\tag{29}
$$

Here $\mathcal{M}$ and $\mathcal{N}$ are $(d-1)$-categories formed by topological interfaces between the adjacent topological BCs, and $m \in \mathcal{M}$ and $n \in \mathcal{N}$ are the labels for objects in these $(d-1)$-categories.[2] This justifies our claim that if an $\mathcal{S}$-symmetric gapped phase admits a gapped boundary, then it admits gapped boundaries transforming in all possible $(d-1)$-charges under $\mathcal{S}$. This is a radical departure from non-topological $\mathcal{S}$-symmetric theories, where not all possible $(d-1)$-charges may exist. In fact, in general there exist many multiplets of gapped BCs transforming in the same $(d-1)$-charge. The different multiplets are labeled by $n \in \mathcal{N}$.

We will often denote the SymTFT sandwich for the gapped phase in terms of

$$
(\mathfrak{B}_{\mathcal{S}}^{\text{sym}}|\mathfrak{B}_{\mathfrak{T}}^{\text{phys}}),
\tag{30}
$$

and the boundary SymTFT

$$
(\mathfrak{B}_{\mathcal{S}}^{\text{sym}}|\boldsymbol{Q}_d|\mathfrak{B}_{\mathfrak{T}}^{\text{phys}}).
\tag{31}
$$

In particular the topological defects on the $\boldsymbol{Q}_d$ boundary corresponds to the symmetry category given by the $\mathcal{S}$-endofunctors of $\mathcal{M}$

$$
\mathcal{S}_{\mathcal{M}}^* := \text{Fun}_{\mathcal{S}}(\mathcal{M}, \mathcal{M}).
\tag{32}
$$

In this setup all three gapped boundary conditions of the SymTFT are related by gauging, or are Morita equivalent. Recall that, using internal-Hom constructions (e.g. for $d = 2$ in [54]), we can associate to the module category $\mathcal{M}$ an algebra object $\mathcal{A}_{\mathcal{M}}$, such that gauging $\mathcal{A}_{\mathcal{M}}$ maps between $\mathcal{S}$ and $\mathcal{S}_{\mathcal{M}}^*$ symmetries.[3] The same operation implements a map between gapped boundaries. We will denote this procedure by $/\mathcal{M}$ for simplicity. Then if we input the symmetry boundary, the physical boundary is related to it by

$$
\mathfrak{B}_{\text{TQFT}}^{\text{phys}} = \mathfrak{B}_{\mathcal{S}}^{\text{sym}}/\mathcal{M}/\mathcal{N} \equiv \mathfrak{B}_{\mathcal{S}}^{\text{sym}}/(\mathcal{M} \boxtimes_{\mathcal{S}_{\mathcal{M}}^*} \mathcal{N}).
\tag{33}
$$

We can also specify what symmetry the physical boundary condition (used as a symmetry boundary) would carry and that is

$$
(\mathcal{S}_{\mathcal{M}}^*)_{\mathcal{N}}^*.
\tag{34}
$$

---

[2]Note that $\mathcal{M}$ is a module category for $\mathcal{S}$ but $\mathcal{N}$ is not a module category for $\mathcal{S}$, in fact $\mathcal{N}$ is a module category over $\mathcal{S}_{\mathcal{M}}^*$ which can be seen as follows. Set $\mathcal{S}_{\mathcal{M}}^* := \text{Fun}_{\mathcal{S}}(\mathcal{M}, \mathcal{M})$, the dual fusion category. A defect $F \in \mathcal{S}_{\mathcal{M}}^*$ sits on the $\boldsymbol{Q}_d$ boundary labelled by $\mathcal{M}$; sliding it through the interface acts on an object $n \in \mathcal{N}$ by $n \mapsto F(n)$, giving a functor $\chi_F : \mathcal{N} \to \mathcal{N}$. Composition of defects matches composition of functors, so $F \mapsto \chi_F$ is a strict monoidal map $\mathcal{S}_{\mathcal{M}}^* \to \text{End}(\mathcal{N})$. Thus $\mathcal{N}$ is a (left) module category over $\mathcal{S}_{\mathcal{M}}^*$.

[3]For $d = 2$ the internal–Hom is classical [53, 54]. For $d = 3$ an analogous construction does exist: finite semisimple module 2-categories over a fusion 2-category possess internal–Homs [55, 57]. For $d > 3$ no general theory has been completed, so statements in those dimensions remain conjectural.

From this perspective all boundary conditions of gapped phases are determined as follows:

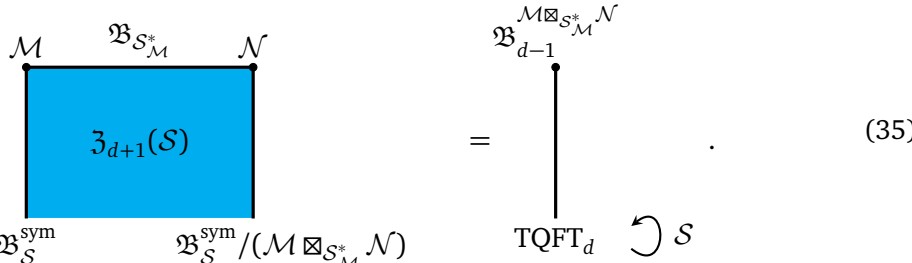

$$\tag{35}$$

Thus, a given gapped boundary condition $\mathfrak{B}^{\text{phys}}_{\mathfrak{T}_d}$ will determine the possible $\mathcal{N}$ module categories that can appear at this interface.

In practice when determining the symmetry properties of boundary conditions of gapped phases we proceed as follows: We fix the symmetry and physical boundaries – as these determine the bulk topological phase – and then cycle through the possible module categories $\mathcal{M}$ and $\mathcal{N}$, which in turn will fix what the $\boldsymbol{Q}_d$ boundary condition is. For (1+1)d theories in particular we label all gapped boundary conditions in terms of Lagrangian algebras $\mathcal{L}$ of the SymTFT, and we will label them accordingly in the next section.

**Wedges and interfaces.** This setup in general describes decomposable boundary conditions for $\mathfrak{T}_d$. These are enriched boundary conditions, meaning that (some of) the defects in $\mathcal{S}^*_{\mathcal{M}}$ stretching trough the upper boundary descend onto non-trivial $(d-2)$-dimensional topological operators localized at the boundary, which are charged under the symmetry $\mathcal{S}$. Notice that in this setup we may only enrich the boundary condition by a symmetry $\mathcal{S}^*_{\mathcal{M}}$. Indecomposable (but not enriched) boundary conditions are instead described by a SymTFT wedge [59][4]

$$\tag{36}$$

We can compactify the horizontal boundary defined by $\boldsymbol{Q}_d$ and map the boundary SymTFT (35) also to a wedge with the following module category

$$\tag{37}$$

Finally, we can compactify (35) to obtain an $\mathcal{S}$-transparent interface between the two

---

[4]One could maybe call this "triangular sandwich", or a "taco". We thank Zhengdi Sun for this suggestion.

gapped phases $(\mathfrak{B}_{\mathcal{S}}^{\mathrm{sym}}|\mathfrak{B}_{\mathfrak{T}}^{\mathrm{phys}})$ and $(\mathfrak{B}_{\mathcal{S}}^{\mathrm{sym}}|\boldsymbol{Q}_d)$:

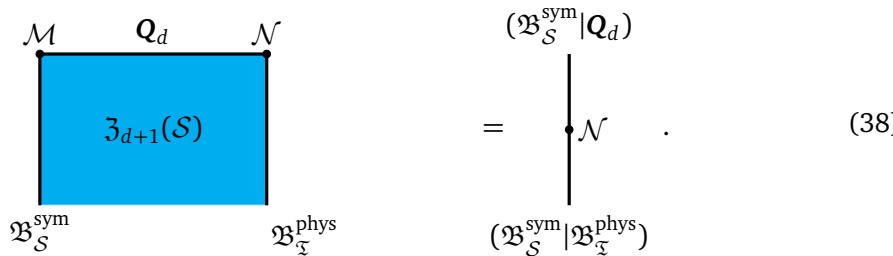

$$\tag{38}$$

Interestingly, this last step can also be performed for a non-topological $\mathfrak{B}^{\mathrm{phys}}$. This allows to reinterpret the symmetry structure of BCs as transparent interfaces between a QFT and a gapped phase $(\mathfrak{B}^{\mathrm{sym}}|\boldsymbol{Q}_d)$.

# 3 Boundary SymTFTs for (1+1)d theories

## 3.1 General setup

For (1+1)d theories, the boundary is one-dimensional, and the SymTFT is a (2+1)d topological order. Its topological lines form the Drinfeld center $\mathcal{Z}(\mathcal{S})$ of the fusion category symmetry $\mathcal{S}$. Gapped boundary conditions are in one-to-one correspondence with Lagrangian algebras $\mathcal{L}$ in the Drinfeld center, which in turn are in one-to-one correspondence with module categories over $\mathcal{S}$ [46]. For instance for group-like symmetries, i.e. $\mathcal{S}$ that are gauge-related to $\mathsf{Vec}_G$, all module categories are labelled $H$ a subgroup of $G$ and $\beta \in H^2(H, U(1))$. Given a Lagrangian algebra $\mathcal{L}$, the SymTFT description of a boundary condition $\mathfrak{B}^{\mathcal{M}}$, associated to a 2-charge $\boldsymbol{Q}_2$ is given by the sandwich:

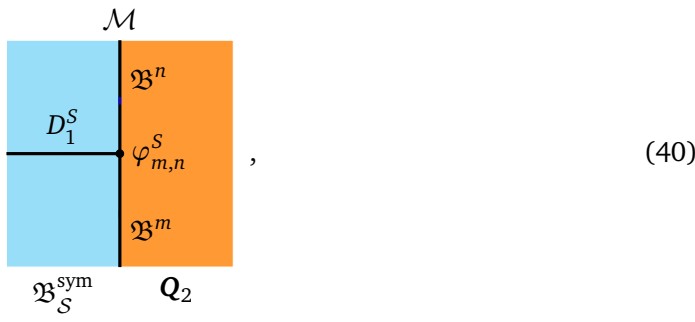

$$\tag{39}$$

The symmetry $\mathcal{S}$ acts on $\mathcal{M}$ on the right, endowing it with the structure of a module category over $\mathcal{S}$ [49]. The boundary condition $\mathfrak{B}^{\mathcal{M}}$ splits, under this action, into irreducible components $\mathfrak{B}^m$, $m \in \mathcal{M}$.

This structure is best described with the aid of topological junctions. Given a simple line $D_1^S$, $S \in \mathcal{S}$ we construct a vector space $V_{m,n}^S$ of topological junctions:[5]

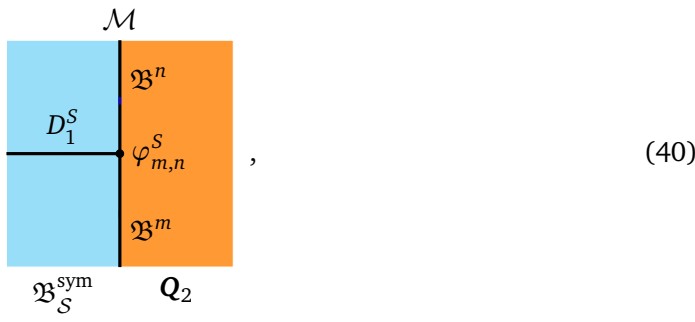

$$\tag{40}$$

---

[5]Here we flatten the boundary of the SymTFT out and project onto the plane – in the initial figure we keep the color coding of the introduction figure 1.

where $\varphi_{m,n}^S \in \mathrm{Hom}_{\mathcal{M}}(S \otimes m, n)$ encodes a choice of basis for $V_{m,n}^S$. The symmetry action on $\mathcal{M}$ is consistent once its boundary F-symbols $\hat{F}$ are specified:

$$
\begin{array}{c}
\includegraphics \end{array}
= \sum_{S_3, \varphi_3, \alpha} (\hat{F}_{m,S_2,S_1}^{n,o,S_3})_{\varphi_1, \varphi_2}^{\varphi_3, \alpha}
\quad \begin{array}{c} \includegraphics \end{array} \; . \tag{41}
$$

Subject to the boundary pentagon equation. This describes the $\mathcal{S}$ action on boundary conditions. Similarly, one can study parallel fusion of $D_1^S$ lines with a boundary condition $\mathfrak{B}^m$. This is described by a NIM matrix $N_{S_m}^{\ n}$ defined through:

$$
D_1^S \otimes \mathfrak{B}^m = \sum_{n \in \mathcal{M}} N_{S_m}^{\ n} \mathfrak{B}^n . \tag{42}
$$

The multiplicities $N_{S_m}^{\ n} \in \mathbb{Z}^+$ count the dimension of the vector space of topological junctions $\varphi_{m,n}^S$.

## 3.2 Boundary changing operators

An $\mathcal{S}$-symmetric boundary condition comes with many interesting observables. Among them, boundary-changing operators $\phi_{m,n}$ are particularly relevant. In (1+1)d these are dynamical local operators interpolating between boundary conditions $\mathfrak{B}^m$ and $\mathfrak{B}^n$. The SymTFT formulation is extremely powerful in elucidating their multiplet structure. Recall that the 2-charge $Q_2$ hosts its own topological symmetry lines $Q_1^\nu$, which are described by elements:

$$
\nu \in \mathcal{S}_{\mathcal{M}}^* , \tag{43}
$$

the category of $\mathcal{S}$-endofunctors of $\mathcal{M}$. These lines may also terminate topologically on $\mathcal{M}$ from the right in a consistent manner:

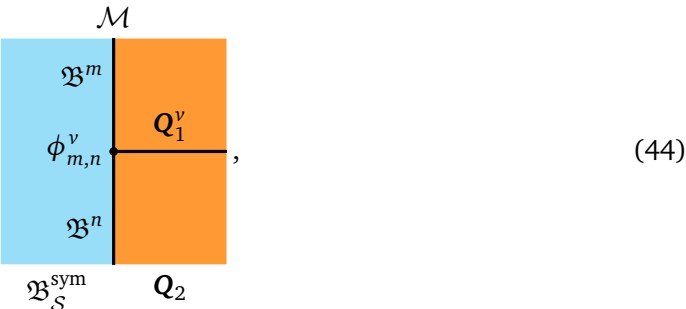

$$
\tag{44}
$$

endowing it with the structure of a right $\mathcal{S}_{\mathcal{M}}^*$ module as well. Thus, they are provided with their own set of boundary F-symbols satisfying a boundary pentagon identity for the $\mathcal{S}_{\mathcal{M}}^*$ symmetry. A line $Q_1^\nu$ extending between $\mathcal{M}$ and $M$ describes a dynamical local operator, $\Phi_{m,n}^\nu$, interpolating between $\mathfrak{B}^m$ and $\mathfrak{B}^n$ boundary conditions –that is, a boundary changing operator. Projecting on the $Q_2$ boundary, we will represent it as follows:

$$
\begin{array}{c} \includegraphics \end{array}
=
\begin{array}{c} \includegraphics \end{array} , \qquad \nu \in \mathcal{S}_{\mathcal{M}}^* \; . \tag{45}
$$

A symmetry line $D_1^S$, $S \in \mathcal{S}$ acts on a boundary-changing operator $\phi_{m,n}^\nu$ via a linear map

$$(S_\nu): V_{m,n}^\nu \longrightarrow V_{m',n'}^\nu. \tag{46}$$

Graphically:

$$D_1^S \left( \phi \begin{matrix} \mathfrak{B}^{m'} \\ \mathfrak{B}^m \ \nu \\ \mathfrak{B}^n \\ \mathfrak{B}^{n'} \end{matrix} \right) = \sum_{\phi_{m',n'}'} (S_\nu)_{\phi_{m,n}}^{\phi_{m',n'}'} \ \phi' \ \Big| \ \nu \ . \tag{47}$$

Endowing $\mathcal{M}$ with the mathematical structure of a $\mathcal{S}-\mathcal{S}_\mathcal{M}^*$ bimodule category. Importantly, the $\mathcal{S}_\mathcal{M}^*$ fusion structure describes the tensor product decomposition for multiplets of boundary-changing operators.

## 3.3 Gapped phases

We now specialize to (1+1)d gapped phases. A gapped phase $\mathfrak{T}$ for a fusion category symmetry $\mathcal{S}$ is specified in terms two Lagrangian algebras of the Drinfeld center:

- $\mathcal{L}_{\text{sym}}$ fixes the symmetry $\mathcal{S}$ and specifies the symmetry boundary $\mathfrak{B}_\mathcal{S}^{\text{sym}}$.

- $\mathcal{L}_{\text{phys}}$ determines $\mathfrak{B}^{\text{phys}}$ and specifies the gapped phase. Cycling through arbitrary $\mathcal{L}_{\text{phys}}$ maps out all $\mathcal{S}$-symmetric gapped phases in (1+1)d.

We will denote the phases by $(\mathcal{L}_{\text{sym}}|\mathcal{L}_{\text{phys}})$. The interval compactification with boundary $\boldsymbol{Q}_2$:

$$\begin{matrix} \mathcal{M} & \mathcal{L}_{\boldsymbol{Q}_2} & \mathcal{N} \\ \mathcal{L}_{\text{sym}} & \mathfrak{Z}(\mathcal{S}) & \mathcal{L}_{\text{phys}} \\ & (\mathcal{L}_\mathcal{S} \mid \mathcal{L}_{\boldsymbol{Q}_2} \mid \mathcal{L}_{\text{phys}}) & \end{matrix} \tag{48}$$

will instead be denoted by $\left(\mathcal{L}_{\text{sym}} \mid \boldsymbol{Q}_2 \mid \mathcal{L}_{\text{phys}}\right)$. Its compactified avatar describes the gapped phase $\mathfrak{T}$ in the presence of a gapped boundary condition $\mathfrak{B}^{\mathcal{M}\boxtimes_{\mathcal{S}_\mathcal{M}^*}\mathcal{N}}$. Apart from the module category $\mathcal{M}$, determining the symmetry action on the boundary condition, we also need to specify a second interface, $\mathcal{N}$, which will determine the spectrum of boundary-changing operators. Notice that the interfaces $\mathcal{M}$ and $\mathcal{N}$ are fixed once the BCs are fixed uniquely by the choice of algebras $\mathcal{L}_{\text{sym}}$, $\mathcal{L}_{\boldsymbol{Q}_2}$, $\mathcal{L}_{\text{phys}}$.

Most of what we have described in the general case goes through also here, however now the second topological interface $\mathcal{N}$ will furnish a set of labels $n \in \mathcal{N}$. A symmetric boundary condition (or 1-charge $\boldsymbol{Q}_2$) for a gapped phase is thus represented by the choice of a pair $(m,n) \in \mathcal{M}\boxtimes\mathcal{N}$. If there exists a topological boundary-changing operator between $(m,n)$ and some $(m',n') \in \mathcal{M}\boxtimes\mathcal{N}$, it can be depicted as a 0-charge $\boldsymbol{Q}_1^s$ with $s \in \mathcal{S}_\mathcal{M}^*$ of the SymTFT as in figure 3. In a projection we will depict it as follows:

$$\begin{matrix} \mathcal{M} & \boldsymbol{Q}_2 & \mathcal{N} \\ \mathfrak{B}^{m'} & \boldsymbol{Q}_1^s & \mathfrak{B}^{n'} \\ \mathfrak{B}^m \ \phi_{m',m}^s & \psi_{n,n'}^s \ \mathfrak{B}^n \end{matrix} = \begin{matrix} \mathfrak{B}^{m',n'} \\ \Phi_{(m,n),(m',n')}^s, \\ \mathfrak{B}^{m,n} \end{matrix} \qquad \nu \in \mathcal{S}_\mathcal{M}^* \ . \tag{49}$$

Notice that the symmetry acts only on the first set of labels $m, m'$ while the boundary-changing operator acts on both. We now discuss several relevant examples.

### 3.4 Examples: Finite groups

#### 3.4.1 Non-anomalous $\mathbb{Z}_2$

We first consider the non-anomalous $\mathbb{Z}_2$ symmetry in (1+1)d generated by the invertible lines 1 and $P$ with $P^2 = 1$. The SymTFT in this case is the (untwisted) 3d $\mathbb{Z}_2$ Dijkgraaf-Witten theory, whose topological line operators can be labelled by their usual names as 1, $e$, $m$ and $f = em$.

This SymTFT has two Lagrangian algebras

$$\mathcal{L}_e = 1 \oplus e\,, \qquad \mathcal{L}_m = 1 \oplus m\,, \tag{50}$$

often referred to as electric and magnetic Lagrangian algebras. $\mathcal{L}_e = 1 \oplus e$ corresponds to the condensation of the electric anyon ($e$) with the magnetic anyon ($m$) being projected to the non-trivial $\mathbb{Z}_2$ symmetry generator $P$. On the other hand, on $\mathcal{L}_m$ the anyon $m$ condenses and projects $e$ to the dual $\hat{\mathbb{Z}}_2$ symmetry generator $\widehat{P}$. Choosing these as symmetry boundaries results in the following symmetry categories

$$
\begin{aligned}
\mathfrak{B}_{\mathcal{S}}^{\text{sym}} = \mathcal{L}_e : &\qquad \mathcal{S} = \text{Vec}_{\mathbb{Z}_2}\,, \\
\mathfrak{B}_{\mathcal{S}}^{\text{sym}} = \mathcal{L}_m : &\qquad \mathcal{S} = \text{Rep}(\mathbb{Z}_2) \cong \text{Vec}_{\widehat{\mathbb{Z}}_2}\,.
\end{aligned}
\tag{51}
$$

The gapped phases for $\mathcal{S} = \text{Vec}_{\mathbb{Z}_2}$ are obtained by choosing

$$
\mathfrak{B}_{\mathcal{S}}^{\text{sym}} = \mathcal{L}_e : \quad
\begin{cases}
\mathfrak{B}^{\text{phys}} = \mathcal{L}_e : & \mathbb{Z}_2 \text{ SSB phase,} \\
\mathfrak{B}^{\text{phys}} = \mathcal{L}_m : & \mathbb{Z}_2 \text{ trivial phase.}
\end{cases}
\tag{52}
$$

**Module categories.** $\mathcal{S} = \text{Vec}_{\mathbb{Z}_2}$ has two module categories

$$
\begin{aligned}
\mathcal{M}_{\text{reg}} &= \text{Vec}_{\mathbb{Z}_2} = \{m_{\pm}\}\,, \\
\mathcal{M}_{\text{Vec}} &= \text{Vec} = \{m_0\}\,,
\end{aligned}
\tag{53}
$$

where we indicated the simple objects. In particular $\text{Vec}_{\mathbb{Z}_2}$ is the module category between the boundary conditions defined by $\mathcal{L}_e$ with itself and $\text{Vec}$ is the module category between $\mathcal{L}_e$ and $\mathcal{L}_m$.

Recall that for 2d TQFTs the indecomposable boundary conditions are in one-to-one correspondence with the vacua. The corresponding BC describes how to end the trivial 2d TFT governing the vacuum. The trivial phase has a single vacuum $v_0$, while the $\mathbb{Z}_2$ broken phase has two vacua $v_{\pm}$.

**BCs for gapped phases.** For each $\mathbb{Z}_2$-symmetric gapped phase, we also have two choices for the boundary $Q_2$ giving rise to four possibilities. We denote a sandwich $(\mathfrak{B}^{\text{sym}} \mid \mathfrak{B}^{\text{phys}})$ with boundary $Q_2$ by

$$(\mathfrak{B}^{\text{sym}} \mid Q_2 \mid \mathfrak{B}^{\text{phys}})\,. \tag{54}$$

The four possibilities are then shown in figure 8.

- $(\mathcal{L}_e | \mathcal{L}_e | \mathcal{L}_m)$: This is the trivial phase, and there are two boundary conditions:

$$\mathfrak{B}^{m_+, m_0} \quad \longleftrightarrow \quad \mathfrak{B}^{m_-, m_0}\,, \qquad m_{\pm} \in \mathcal{M} = \text{Vec}_{\mathbb{Z}_2}\,, \quad m_0 \in \mathcal{N} = \text{Vec}\,, \tag{55}$$

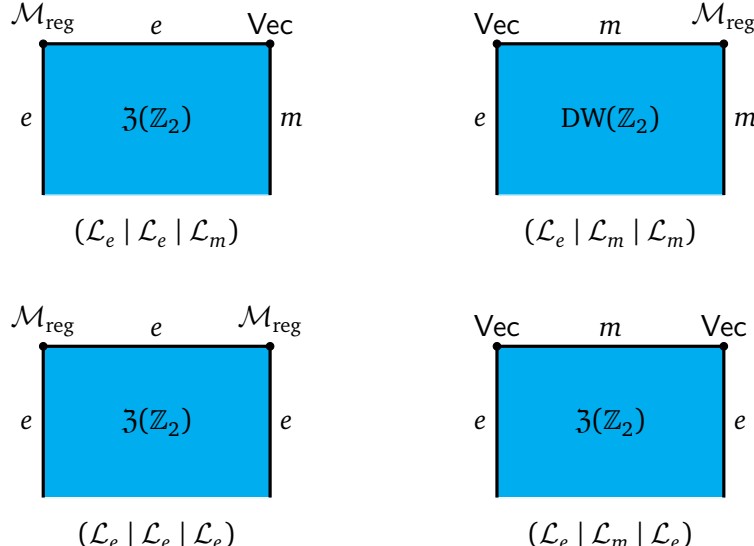

Figure 8: Four Boundary SymTFTs for $\mathbb{Z}_2$ 0-form symmetry in (1+1)d. The top two correspond to the Boundary SymTFTs for the trivial phase, and the bottom two to the $\mathbb{Z}_2$ SSB phase.

which are exchanged under the $\mathbb{Z}_2$ symmetry, as this acts non-trivially on the modules $m_\pm \in \mathrm{Vec}_{\mathbb{Z}_2}$. This can be matched with the known BCs of the trivial TQFT in 2d, which has a single boundary condition $\mathcal{B}$. Here $\mathfrak{B}^{m_\pm, m_0} = \mathcal{B}$. This is consistent with the symmetry action $P\mathcal{B} = \mathcal{B}$ as in this phase the symmetry is simply generated by the identity line, i.e. $P = 1$.

- $(\mathcal{L}_e|\mathcal{L}_m|\mathcal{L}_m)$: The other choice for the trivial gapped phase is to have $Q_2 = \mathcal{L}_m$ in which case there are two boundary conditions that are singlets under the $\mathbb{Z}_2$

$$\mathfrak{B}^{m_0, m_\pm}. \tag{56}$$

Again identifying with the boundary condition $\mathcal{B}$ of the trivial 2d TQFT we must have $\mathfrak{B}^{m_0, m_\pm} = \mathcal{B}$. The two BCs $\mathfrak{B}^{m_0, m_+}$ and $\mathfrak{B}^{m_0, m_-}$ are still differentiated as follows. Consider the end $\mathcal{O}_P$ of $P$ along $\mathcal{B}$. Since the 2d phase is trivial, $\mathcal{B}$ is a 1d TFT and can be identified with trivial 1d TFT whose Hilbert space of states is described by $\mathbb{C}$. Since $P = 1$, we can identify the end $\mathcal{O}_P$ by an operator in this 1d TFT, which can be further identified with complex numbers. The $\mathbb{Z}_2$ requirement on this complex number is

$$\mathcal{O}_P^2 = 1, \tag{57}$$

which fixes

$$\mathcal{O}_P = \pm 1, \tag{58}$$

which differentiates $\mathfrak{B}^{m_0, m_\pm}$. In other words, $\mathfrak{B}^{m_0, m_\pm}$ can be identified with the two one-dimensional irreps of $\mathbb{Z}_2$.

- $(\mathcal{L}_e|\mathcal{L}_e|\mathcal{L}_e)$: For the $\mathbb{Z}_2$ SSB phase there are again two configurations. First consider $Q_2 = \mathcal{L}_e$. Then all module categories are regular module categories and we get four boundary conditions

$$\mathfrak{B}^{m_+, n_\pm} \quad \longleftrightarrow \quad \mathfrak{B}^{m_-, n_\pm}, \tag{59}$$

where the $\mathbb{Z}_2$ symmetry acts on the $m_\pm$ and exchanges them pair-wise.

Again we wish to map this to the boundary conditions of the $\mathbb{Z}_2$ SSB phase directly, which has two underlying boundary conditions $\mathcal{B}_0$ and $\mathcal{B}_1$. The $\mathbb{Z}_2$ generator is realized as $P = 1_{01} + 1_{10}$, and exchanges the two boundary conditions. We can thus identify

$$\mathfrak{B}^{m_+,n_+} = \mathcal{B}_0, \qquad \mathfrak{B}^{m_-,n_+} = \mathcal{B}_1, \tag{60}$$

and

$$\mathfrak{B}^{m_+,n_-} = \mathcal{B}_1, \qquad \mathfrak{B}^{m_-,n_-} = \mathcal{B}_0. \tag{61}$$

- $(\mathcal{L}_e|\mathcal{L}_m|\mathcal{L}_e)$: The second option in the case of the $\mathbb{Z}_2$ SSB phase is $\boldsymbol{Q}_2 = \mathcal{L}_m$. In this case there is precisely one boundary condition

$$\mathfrak{B}^{m_0,n_0}, \tag{62}$$

that is invariant under the $\mathbb{Z}_2$.

In terms of the underlying BCs for the SSB phase, this can be identified as

$$\mathfrak{B}^{m_0,n_0} = \mathcal{B}_0 \oplus \mathcal{B}_1. \tag{63}$$

Indeed the symmetry generator $P$ can end on $\mathfrak{B}^{m_0,n_0}$.

**Boundary-changing operators.** Apart from studying the boundaries, which are represented as simple objects of module categories, there are also boundary-changing operators, or 1-morphisms, which we go over in turn:

- $(\mathcal{L}_e|\mathcal{L}_e|\mathcal{L}_m)$: For $\mathcal{M}_{\mathrm{reg}}$ there are two non-trivial boundary-changing operators $\phi_{+,-}^P \in \mathrm{Hom}_{\mathcal{M}_{\mathrm{reg}}}(P \otimes \mathfrak{B}^{m_+}, \mathfrak{B}^{m_-})$[6] and $\phi_{-,+}^P$ with exchanged indices. These local operators commute with the $\mathbb{Z}_2$ symmetry action of $P$ coming from $\mathfrak{B}^{\mathrm{sym}}$, in fact

$$P: \quad \phi_{+,-}^P \longleftrightarrow \phi_{-,+}^P. \tag{64}$$

By extending $P$ from $\phi_{+,-}^P$ to the physical boundary, one can terminate on the single non-trivial local operator of $\mathcal{M}_{\mathrm{Vec}}$ which is $\psi_{0,0}^P \in \mathrm{End}_{\mathcal{M}_{\mathrm{Vec}}}(\mathfrak{B}^{m_0})$, so that the analog of figure 49 is here – where we label the 1-charge $\boldsymbol{Q}_2$ by the label that specifies the boundary condition, in this case it is $\mathcal{L}_e$, so we denote it by $\boldsymbol{Q}_2^e$:

$$\begin{array}{c}\text{(figure)}\end{array} \tag{65}$$

After collapsing the boundary sandwich one ends up with the two boundaries $\mathfrak{B}^{m_\pm}$ separated by $\Phi_{-,+}^P \in \mathrm{Hom}(\mathfrak{B}^{m_-,m_0}, \mathfrak{B}^{m_+,m_0})$ which is now a 1-morphism in $\mathcal{M}_{\mathrm{reg}} \boxtimes_{\mathrm{Vec}_{\mathbb{Z}_2}} \mathrm{Vec}$, whose symmetry properties are inherited from $\phi_{+,-}^P$,

$$P: \quad \Phi_{+,-}^P \longleftrightarrow \Phi_{-,+}^P. \tag{66}$$

---

[6]As noted previously, in the diagram below $\phi_{+,-}^P$ is a junction operator coming from the right action on $\mathcal{M}_{\mathrm{reg}}$, yet we define it as coming from the left action on $\mathcal{M}_{\mathrm{reg}}$ by defining the notation through a $\pi$ rotation. Rotating the picture by $\pi$, we see that $\mathfrak{B}^{m_+}$ is acted on by $P$ from the left and changes to $\mathfrak{B}^{m_-}$, thus $\phi_{+,-}^P \in \mathrm{Hom}(P \otimes \mathfrak{B}^{m_+}, \mathfrak{B}^{m_-})$.

- $(\mathcal{L}_e|\mathcal{L}_m|\mathcal{L}_m)$: On the level of morphisms, the situation is similar to case a) with the roles of Vec and $\mathcal{M}_{\text{reg}}$ reversed:

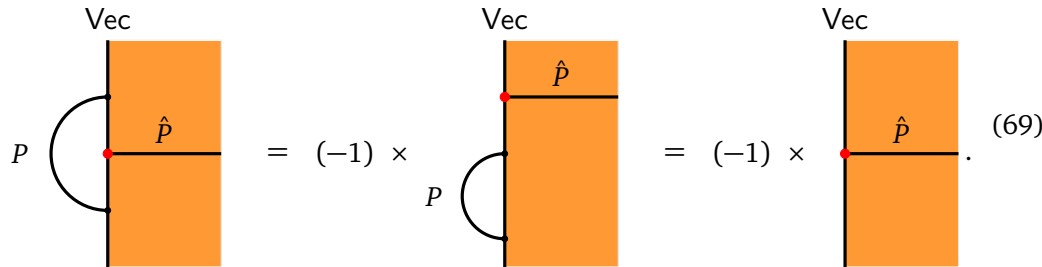

$$
\begin{array}{c}
\text{Vec} \quad \boldsymbol{Q}_2^m \quad \mathcal{M}_{\text{reg}} \\
\end{array}
= \quad \Phi_{-,+}^{\hat{P}} \quad . \tag{67}
$$

However, now the combined boundary-changing operator $\Phi_{-,+}^{\hat{P}}$ is charged under the $\mathbb{Z}_2$ symmetry generated by $P$ on $\mathfrak{B}^{\text{sym}}$

$$
P: \quad \Phi_{-,+}^{\hat{P}} \rightarrow -\Phi_{-,+}^{\hat{P}}. \tag{68}
$$

This is the case as $\boldsymbol{Q}_2 = \mathcal{L}_m$ condenses the bulk magnetic anyon $m$ and instead confines the electric anyon $e$ which projects to line $\hat{P}$ on $\boldsymbol{Q}_2$ which braids non-trivially with $P$ (the projection of the $m$ anyon on $\mathfrak{B}^{\text{sym}}$) and thus

$$
\begin{array}{ccccc}
\text{Vec} & & \text{Vec} & & \text{Vec} \\
\end{array}
$$

$$
P \quad \hat{P} \quad = \quad (-1) \times \quad \hat{P} \quad = \quad (-1) \times \quad \hat{P} \quad . \tag{69}
$$

- $(\mathcal{L}_e|\mathcal{L}_e|\mathcal{L}_e)$: This setup describes a quadruplet of boundary conditions $(m_s, m_q) \in \mathcal{M}_{\text{reg}} \boxtimes_{\text{Vec}_{\mathbb{Z}_2}} \mathcal{M}_{\text{reg}}$, with $s, q = \pm$. The $\mathbb{Z}_2$ symmetry sends $(m_s, m_q)$ to $(m_{-s}, m_q)$, while the non-trivial boundary-changing operators $\Phi_{(-s,-q),(s,q)}^{P}$ (as defined pictorially below) map $(m_{-s}, m_{-q})$ into $(m_s, m_q)$:

$$
\begin{array}{c}
\mathcal{M}_{\text{reg}} \quad \boldsymbol{Q}_2^e \quad \mathcal{M}_{\text{reg}} \\
\end{array}
= \quad \Phi_{(-s,-q),(s,q)}^{P} \quad . \tag{70}
$$

The boundary-changing defects $\Phi_{(-s,-q),(s,q)}^{P}$ are neutral under the bulk $\mathbb{Z}_2$ symmetry as $P$ braids trivially with itself as in case a) but it transforms as

$$
P: \quad \Phi_{(-s,-q),(s,q)}^{P} \leftrightarrow \Phi_{(s,-q),(-s,q)}^{P}. \tag{71}
$$

- $(\mathcal{L}_e|\mathcal{L}_m|\mathcal{L}_e)$: For each Vec there is only one non-trivial morphism in $\text{End}(\mathfrak{B}^{m_0})$, labelled by $\phi_{0,0}^{\hat{P}}$ or $\psi_{0,0}^{\hat{P}}$ below:

$$
\begin{array}{c}
\text{Vec} \quad \boldsymbol{Q}_2^m \quad \text{Vec} \\
\end{array}
= \quad \Phi_{0,0}^{\hat{P}} \quad . \tag{72}
$$

Indeed, we also find that each of these endomorphisms can be used to split the object $\mathfrak{B}^{m_0}$ into its simple components (from the point of view of the $\mathbb{Z}_2$ SSB phase), as clearly $\mathfrak{B}^{m_0} = \mathfrak{B}^{m_+} \oplus \mathfrak{B}^{m_-}$. After collapsing the SymTFT, the combination of the non-trivial morphisms of $\phi_{0,0}^{\hat{P}}$ and $\psi_{0,0}^{\hat{P}}$ gives rise to the nontrivial topological operator $\Phi_{0,0}^{\hat{P}} \in \mathrm{End}(\mathfrak{B}^{m_0,m_0})$, which is charged under the $\mathbb{Z}_2$ symmetry as in case b) as $\hat{P}$ braids non-trivially with $P$:

$$P: \quad \Phi_{0,0}^{\hat{P}} \to -\Phi_{0,0}^{\hat{P}}. \tag{73}$$

### 3.4.2 Anomalous $\mathbb{Z}_2$

Let us now consider the case in which the $\mathbb{Z}_2$ symmetry is anomalous and the anomaly is described by the non-trivial 't Hooft anomaly $\omega \neq 0$,

$$\omega \in H^3(\mathbb{Z}_2, U(1)) = \mathbb{Z}_2, \tag{74}$$

whose only non-trivial element is $\omega(P, P, P)$ which is described by the folowing F-move:

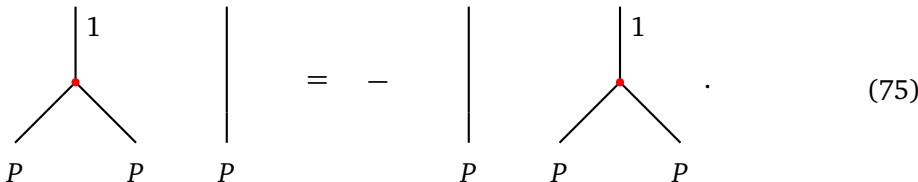

$$\tag{75}$$

The bulk SymTFT is the (2+1)d $\mathbb{Z}_2$ DW theory with a twist, also known as the double semion theory, whose topological lines are commonly labelled by $1$, $s$, $\bar{s}$ and $s\bar{s}$, where the semion $s$ and antisemion $\bar{s}$ have topological spins $+i$ and $-i$ respectively. This SymTFT only allows for a single gapped boundary condition

$$\mathcal{L}_{s\bar{s}} = 1 \oplus s\bar{s}, \tag{76}$$

describing the anomalous symmetry $\mathcal{S} = \mathsf{Vec}_{\mathbb{Z}_2}^{\omega}$, and the one possible gapped phase is a $\mathbb{Z}_2$ gauge theory.

**Module category.** The only module category for the anomalous symmetry is the regular module, as the symmetry cannot be gauged due to the non-trivial 't Hooft anomaly. Hence the only possible case of a boundary sandwich is analogous to the third case seen previously:

$$\tag{77}$$

**Boundary-changing operators.** If we again label the simple objects of $\mathcal{M}_{\mathrm{reg}}$ as $\mathfrak{B}^{m_s}$ and $\mathfrak{B}^{m_q}$ with $s, q = \pm$, we end up with the same boundary changing operators as (70) in case **c)** above. Yet, here $P$ is anomalous with a non-trivial F-symbol $\omega(P, P, P) = (F_{PPP}^P)_{11} = -1$, and

thus acting with the $\mathbb{Z}_2$ symmetry action from the left on $\mathcal{M}_{\text{reg}}$ is now non-trivial:

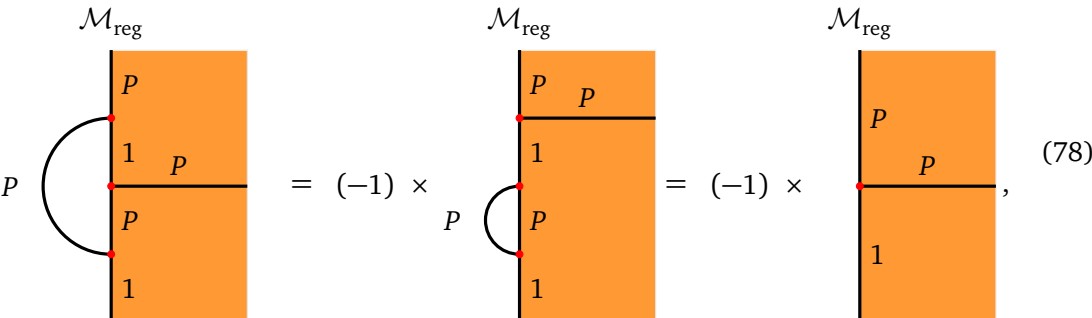

$$(78)$$

where we have abused the notation for boundaries labelling $1 \sim \mathfrak{B}^{m_+}$ and $P \sim \mathfrak{B}^{m_-}$, and the red dot representing the boundary changing operators/junctions $\phi^P$. We have also used the fact that the half-bubble diagram is trivial:

$$(79)$$

Thus the combined boundary-changing operator $\Phi^P$ transforms under $P$ as

$$P : \quad \Phi^P_{(-s,-q),(s,q)} \to -\Phi^P_{(s,-q),(-s,q)} \, , \tag{80}$$

hence the non-trivial anomaly differentiates the symmetry action from the non-anomalous case in (71).

Also, notice that in this case there is no enrichment leading to a singlet boundary condition (even if not a simple one) this is in accordance with the theorem of [48] that an anomalous invertible symmetry admits no invariant boundary condition.

### 3.4.3 Non-anomalous $\mathbb{Z}_2 \times \mathbb{Z}_2$

Next, we study the non-anomalous $\mathbb{Z}_2 \times \mathbb{Z}_2$ symmetry in $(1+1)$d generated by the invertible lines $1$, $P_1$, $P_2$ and $P_1 P_2$ with $(P_1)^2 = (P_2)^2 = 1$. The SymTFT in this case is the (untwisted) 3d $\mathbb{Z}_2 \times \mathbb{Z}_2$ Dijkgraaf-Witten theory, whose 16 topological line operators can be labelled by as $e_1^i e_2^j m_1^k m_2^l$ with $i, j, k, l = 0, 1$ and non-trivial braiding $B(e_1, m_1) = B(e_2, m_2) = -1$.

This SymTFT has six gapped boundary conditions which in the sandwich construction allow for the various SSB patterns of the $\mathbb{Z}_2 \times \mathbb{Z}_2$ symmetry e.g. see [38]. However, here we will only consider phases with single vacua – i.e. trivial phases or SPTs. The relevant BCs are

$$\mathcal{L}_{\text{reg}} = \mathcal{L}_{\mathsf{Vec}_{\mathbb{Z}_2 \times \mathbb{Z}_2}} = 1 \oplus e_1 \oplus e_2 \oplus e_1 e_2 \, , \tag{81}$$

and the other two are the ones which have a trivial intersection with $\mathcal{L}_{\mathsf{Vec}_{\mathbb{Z}_2 \times \mathbb{Z}_2}}$, and thus give rise to SPTs:

$$\begin{aligned} \mathcal{L}_{\text{SPT}^+} &= 1 \oplus m_1 \oplus m_2 \oplus m_1 m_2 \, , \\ \mathcal{L}_{\text{SPT}^-} &= 1 \oplus e_2 m_1 \oplus e_1 m_2 \oplus e_1 e_2 m_1 m_2 \, . \end{aligned} \tag{82}$$

Thus there are two invertible phases with the canonical $\mathbb{Z}_2 \times \mathbb{Z}_2$ symmetry described by the SymTFT sandwiches

$$\begin{aligned} (\mathcal{L}_{\mathsf{Vec}_{\mathbb{Z}_2 \times \mathbb{Z}_2}} | \mathcal{L}_{\text{SPT}^+}) &= \text{SPT}^+ \, , \\ (\mathcal{L}_{\mathsf{Vec}_{\mathbb{Z}_2 \times \mathbb{Z}_2}} | \mathcal{L}_{\text{SPT}^-}) &= \text{SPT}^- \, , \end{aligned} \tag{83}$$

where SPT$^+$ denotes the 2d trivial SPT phase whereas SPT$^-$ is the 2d non-trivial SPT. These two SPTs (or fiber functors for the $\mathbb{Z}_2 \times \mathbb{Z}_2$ symmetry) are characterized by elements of the second group cohomology with coefficients in $U(1)$,

$$\beta \in H^2(\mathbb{Z}_2 \times \mathbb{Z}_2, U(1)) = \mathbb{Z}_2. \tag{84}$$

In both phases, one only finds twisted-sector (string-like) order parameters, characteristic of SPTs. While all these parameters are uncharged under the $\mathbb{Z}_2 \times \mathbb{Z}_2$ symmetry in the trivial SPT$^+$, in the non-trivial SPT$^-$ one finds a $P_1$-twisted operator charged under $P_2$, $P_2$-twisted operator charged under $P_1$, and $P_1 P_2$-twisted operator charged under both $P_1$ and $P_2$.

**Module categories.** One of the module categories of interest to us will again be the regular module category $\mathcal{M}_{\text{reg}}$, whose elements for $\mathbb{Z}_2 \times \mathbb{Z}_2$ we can label as $m_{s,s'}$ for $s, s' = \pm$. The other module category is $\text{Vec}^+$ separating $\mathcal{L}_{\text{Vec}_{\mathbb{Z}_2 \times \mathbb{Z}_2}}$ and $\mathcal{L}_{\text{SPT}^+}$ with a single element $m_0$. Finally, we have the module category $\text{Vec}^-$ defining the interface between $\mathcal{L}_{\text{Vec}_{\mathbb{Z}_2 \times \mathbb{Z}_2}}$ and $\mathcal{L}_{\text{SPT}^-}$ with a single element $m_0$ and a non-trivial commutator described by

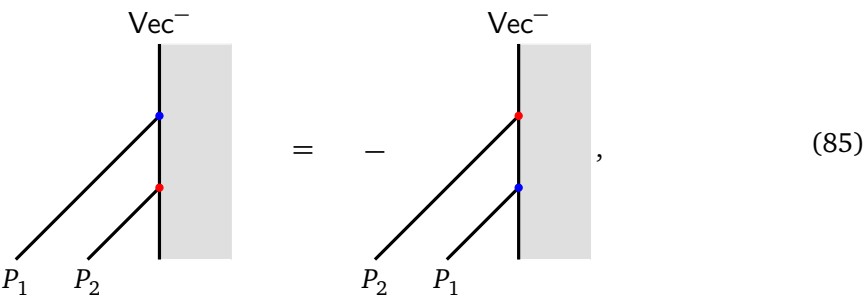

$$\tag{85}$$

which implements whether the $\mathbb{Z}_2 \times \mathbb{Z}_2$ symmetry in 1d is anomaly-free or anomalous, characterized by the 1d anomaly

$$\eta \in H^2(\mathbb{Z}_2 \times \mathbb{Z}_2, U(1)). \tag{86}$$

Notice that, for a boundary system, the total anomaly is given by a combination of the module category $\text{Vec}^\pm$ with anomaly $\eta$ and a bulk inflow term corresponding to the choice of SPT$_\pm$, which is encoded in a sign associated to the trivalent junction $\beta \in \text{Hom}(P_1 \otimes P_2, P_1 P_2)$ describing the SPT.

By only considering invertible topological phases for the $\mathbb{Z}_2 \times \mathbb{Z}_2$ symmetry we need to consider the four boundary sandwiches of the form $(\mathcal{L}_{\text{Vec}_{\mathbb{Z}_2 \times \mathbb{Z}_2}} \mid \mathcal{L}_{\text{SPT}_\pm} \mid \mathcal{L}_{\text{SPT}_\pm})$. This way the underlying theory is SPT$_\pm$ with a singlet boundary transforming in $\text{Vec}^\pm$ under $\mathbb{Z}_2 \times \mathbb{Z}_2$:

- $(\mathcal{L}_{\text{Vec}_{\mathbb{Z}_2 \times \mathbb{Z}_2}} \mid \mathcal{L}_{\text{SPT}^+} \mid \mathcal{L}_{\text{SPT}^+})$: The underlying theory is a trivial SPT with a single vacuum, but there are $|\mathcal{N} = \mathcal{M}_{\text{reg}}| = 4$ boundary conditions. Boundaries will thus transform trivially in the $\text{Vec}^+$ multiplet under the $\mathbb{Z}_2 \times \mathbb{Z}_2$ symmetry as $\text{Vec}^+$ only includes the one irreducible boundary $\mathfrak{B}^{m_0}$. In summary, we get four singlet boundary conditions labelled by

$$\mathfrak{B}^{m_0, n_{\pm, \pm}}, \qquad m_0 \in \text{Vec}^+, \qquad n_{\pm, \pm} \in \mathcal{M}_{\text{reg}}. \tag{87}$$

  The distinction between these BCs is in the ends of $P_1$ and $P_2$ symmetry generating lines along these BCs, as in the $\mathbb{Z}_2$ example. The four BCs can be identified with four 1-dimensional irreps of $\mathbb{Z}_2 \times \mathbb{Z}_2$.

- $(\mathcal{L}_{\text{Vec}_{\mathbb{Z}_2 \times \mathbb{Z}_2}} \mid \mathcal{L}_{\text{SPT}^-} \mid \mathcal{L}_{\text{SPT}^+})$: There is a single boundary condition

$$\mathfrak{B}^{m_0, n_0}, \qquad m_0 \in \text{Vec}^-, \qquad n_0 \in \text{Vec}. \tag{88}$$

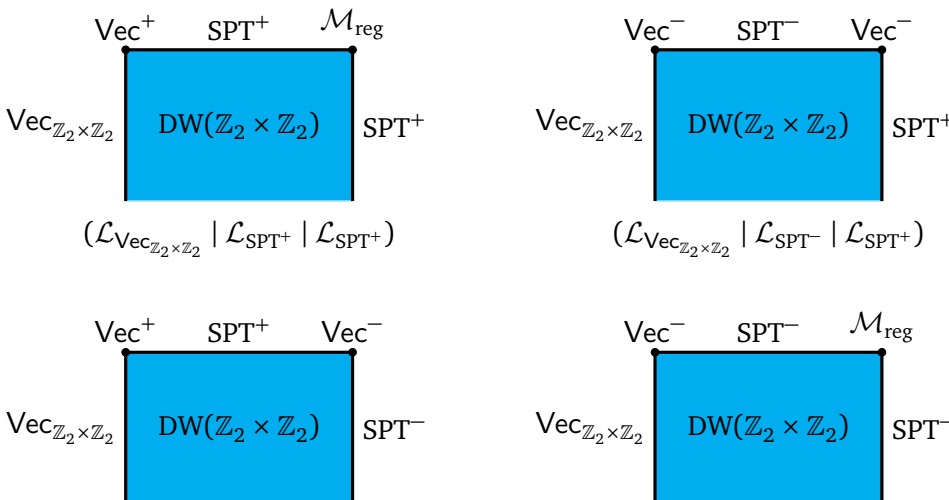

Figure 9: Four Boundary SymTFTs for $\mathbb{Z}_2 \times \mathbb{Z}_2$ 0-form symmetry in (1+1)d. The top two correspond to the Boundary SymTFTs for the trivial phase, and the bottom two to the non-trivial SPT$^-$ phase.

The symmetry acts anomalously as in (85). The resulting BC is identified as

$$\mathfrak{B}^{m_0,n_0} = \mathcal{B} \oplus \mathcal{B}, \tag{89}$$

where $\mathcal{B}$ is the indecomposable BC for the trivial phase. In other words $\mathfrak{B}^{m_0,n_0}$ can be identified with a quantum mechanical system having a two-dimensional Hilbert space which realizes $\mathbb{Z}_2 \times \mathbb{Z}_2$ symmetry with anomaly $\eta$, i.e. a projective representation of $\mathbb{Z}_2 \times \mathbb{Z}_2$.

- $(\mathcal{L}_{\mathsf{Vec}_{\mathbb{Z}_2 \times \mathbb{Z}_2}} \mid \mathcal{L}_{\mathrm{SPT}^+} \mid \mathcal{L}_{\mathrm{SPT}^-})$: The bulk phase is the non-trivial SPT SPT$^-$. For this configuration of module categories, we get a single boundary condition

$$\mathfrak{B}^{m_0,n_0} = \mathcal{B} \oplus \mathcal{B}, \qquad m_0 \in \mathsf{Vec}^+, \quad n_0 \in \mathsf{Vec}. \tag{90}$$

Here the boundary is again identified as a two-dimensional quantum mechanical system. Even though the 1d anomaly $\eta = 0$, we have an anomaly inflow from the 2d bulk for which $\beta \neq 0$. Thus, effectively the 1d system can be identified as a $\mathbb{Z}_2 \times \mathbb{Z}_2$ symmetric system with a non-trivial 't Hooft anomaly. Despite this identification, the 1-charge of the corresponding boundary is identified to be the trivial 1-charge, as the phase originating from the anomaly is encoded as part of the 2d SPT rather than a non-trivial 1-charge of the BC.

- $(\mathcal{L}_{\mathsf{Vec}_{\mathbb{Z}_2 \times \mathbb{Z}_2}} \mid \mathcal{L}_{\mathrm{SPT}^-} \mid \mathcal{L}_{\mathrm{SPT}^-})$: There are four boundary conditions

$$\mathfrak{B}^{m_0,n_i}, \qquad m_0 \in \mathsf{Vec}^-, \quad n_i \in \mathcal{M}_{\mathrm{reg}}. \tag{91}$$

These can be identified with QM systems carrying the four genuine representations of the $\mathbb{Z}_2 \times \mathbb{Z}_2$ symmetry. Still the corresponding 1-charges are non-trivial. This is a consequence of the fact that even though $\beta + \eta$ is trivial, since $\beta$ is non-trivial, $\eta$ must be non-trivial.

**Boundary-changing operators.** On the level of morphisms, we can again investigate all the possible non-trivial boundary-changing operators in the given boundary phases above:

- $(\mathcal{L}_{\mathsf{Vec}_{\mathbb{Z}_2 \times \mathbb{Z}_2}} \mid \mathcal{L}_{\mathrm{SPT}^+} \mid \mathcal{L}_{\mathrm{SPT}^+})$: The four boundary conditions $\mathfrak{B}^{m_0, m_{s,s'}}$ of the trivial SPT transform in a singlet under $\mathbb{Z}_2 \times \mathbb{Z}_2$ and host the following junction operators

$$
\begin{array}{c}
\mathsf{Vec}^+ \quad Q_2^{\mathrm{SPT}^+} \quad \mathcal{M}_{\mathrm{reg}} \\
\end{array}
\qquad = \qquad
\tag{92}
$$

The combined boundary-changing operator $\Phi_{s,s'}^{\hat{P}_1^i \hat{P}_2^j}$ is charged under the $\mathbb{Z}_2 \times \mathbb{Z}_2$ symmetry generated by $\hat{P}_1^i \hat{P}_2^j$ on $\mathfrak{B}^{\mathrm{sym}}$

$$
P_1: \quad \Phi_{s,s'}^{\hat{P}_1^i \hat{P}_2^j} \to (-)^i \Phi_{s,s'}^{\hat{P}_1^i \hat{P}_2^j}, \qquad P_2: \quad \Phi_{s,s'}^{\hat{P}_1^i \hat{P}_2^j} \to (-)^j \Phi_{s,s'}^{\hat{P}_1^i \hat{P}_2^j},
\tag{93}
$$

with analogous reasoning as in (69) but for $P_1$ and $P_2$.

- $(\mathcal{L}_{\mathsf{Vec}_{\mathbb{Z}_2 \times \mathbb{Z}_2}} \mid \mathcal{L}_{\mathrm{SPT}^-} \mid \mathcal{L}_{\mathrm{SPT}^+})$: The singlet boundary condition $\mathfrak{B}^{m_0, m_0}$ of the trivial SPT will give rise to local operators on the 1d boundary $\Phi_0^{\bar{P}_1^i \bar{P}_2^j} \in \mathrm{End}(\mathfrak{B}^{m_0, m_0})$ with $i, j = 0, 1$ as seen below:

$$
\begin{array}{c}
\mathsf{Vec}^- \quad Q_2^{\mathrm{SPT}^-} \quad \mathsf{Vec}^- \\
\end{array}
\qquad = \qquad
\tag{94}
$$

The topological local operators $\Phi_0^{\bar{P}_1^i \bar{P}_2^j}$ will be charged under the anomalous $\mathbb{Z}_2 \times \mathbb{Z}_2$ symmetry in 1d,

$$
P_1: \quad \Phi_0^{\bar{P}_1^i \bar{P}_2^j} \to (-)^i \Phi_0^{\bar{P}_1^i \bar{P}_2^j}, \qquad P_2: \quad \Phi_0^{\bar{P}_1^i \bar{P}_2^j} \to (-)^j \Phi_0^{\bar{P}_1^i \bar{P}_2^j},
\tag{95}
$$

with the 1d anomaly $\eta \in H^2(\mathbb{Z}_2 \times \mathbb{Z}_2, U(1))$ encoded in the ordering of successively applying the $\mathbb{Z}_2 \times \mathbb{Z}_2$ symmetry action $(P_1^i P_2^j) \in \mathsf{Vec}_{\mathbb{Z}_2 \times \mathbb{Z}_2}$ as

$$
(P_1^i P_2^j) \otimes (P_1^k P_2^l) = \eta[(P_1^i P_2^j), (P_1^k P_2^l)] \, (P_1^{i+k} P_2^{j+l}).
\tag{96}
$$

- $(\mathcal{L}_{\mathsf{Vec}_{\mathbb{Z}_2 \times \mathbb{Z}_2}} \mid \mathcal{L}_{\mathrm{SPT}^+} \mid \mathcal{L}_{\mathrm{SPT}^-})$: The singlet boundary condition $\mathfrak{B}^{m_0, m_0}$ of the non-trivial SPT will give rise to local operators on the 1d boundary $\Phi_0^{\hat{P}_1^i \hat{P}_2^j} \in \mathrm{End}(\mathfrak{B}^{m_0, m_0})$ with $i, j = 0, 1$ as seen below:

$$
\begin{array}{c}
\mathsf{Vec}^+ \quad Q_2^{\mathrm{SPT}^+} \quad \mathsf{Vec}^- \\
\end{array}
\qquad = \qquad
\tag{97}
$$

The situation is now thus very similar to the previous case **b)** just above where $\Phi_0^{\hat{P}_1^i \hat{P}_2^j}$ is charged under $\mathbb{Z}_2 \times \mathbb{Z}_2$ as

$$P_1: \quad \Phi_0^{\hat{P}_1^i \hat{P}_2^j} \to (-)^i \Phi_0^{\hat{P}_1^i \hat{P}_2^j}, \qquad P_2: \quad \Phi_0^{\hat{P}_1^i \hat{P}_2^j} \to (-)^j \Phi_0^{\hat{P}_1^i \hat{P}_2^j}, \tag{98}$$

with the 1d anomaly $\beta \in H^2(\mathbb{Z}_2 \times \mathbb{Z}_2, U(1))$, encoded in the ordering of successively applying various $\mathbb{Z}_2 \times \mathbb{Z}_2$ symmetry actions, now coming from the bulk $SPT^-$ instead of from the boundary module category $Vec^-$.

- $(\mathcal{L}_{Vec_{\mathbb{Z}_2 \times \mathbb{Z}_2}} \mid \mathcal{L}_{SPT^+} \mid \mathcal{L}_{SPT^-})$: Finally, the four boundary conditions $\mathfrak{B}^{m_0, m_{s,s'}}$ of the non-trivial SPT transform in a singlet under $\mathbb{Z}_2 \times \mathbb{Z}_2$ and host the following junction operators

$$
\begin{array}{ccc}
Vec^- & Q_2^{SPT^+} & \mathcal{M}_{reg}
\end{array}
$$

$$
\begin{array}{c}
m_0 \\
m_0
\end{array}
\boxed{
\begin{array}{cc}
\bar{P}_1^i \bar{P}_2^j & \\
\phi_0^{\bar{P}_1^i \bar{P}_2^j} & \psi_{s,s'}^{\bar{P}_1^i \bar{P}_2^j}
\end{array}}
\begin{array}{c}
m_{(-)^i s, (-)^j s'} \\
m_{s,s'}
\end{array}
\quad = \quad
\begin{array}{l}
(m_0, m_{(-)^i s, (-)^j s'}) \\
\Phi_{s,s'}^{\bar{P}_1^i \bar{P}_2^j} \\
(m_0, m_{s,s'})
\end{array}
, \tag{99}
$$

and they are charged under $\mathbb{Z}_2 \times \mathbb{Z}_2$ as

$$P_1: \quad \Phi_{s,s'}^{\bar{P}_1^i \bar{P}_2^j} \to (-)^i \Phi_{s,s'}^{\bar{P}_1^i \bar{P}_2^j}, \qquad P_2: \quad \Phi_{s,s'}^{\bar{P}_1^i \bar{P}_2^j} \to (-)^j \Phi_{s,s'}^{\bar{P}_1^i \bar{P}_2^j}, \tag{100}$$

with the combined boundary symmetry action being anomaly-free.

### 3.5 Non-invertible symmetry: $Rep(S_3)$

As an example of a non-anomalous, non-invertible symmetry consider $Rep(S_3)$ in $(1+1)d$ generated by the invertible lines $1$ and $P$ and the non-invertible line $E$ with fusion rules

$$P \otimes E = E \otimes P = E, \qquad E \otimes E = 1 \oplus P \oplus E. \tag{101}$$

The SymTFT description of the gapped phases has appeared in [38], which we briefly recap here. The SymTFT in this case is the (untwisted) 3d $S_3$ Dijkgraaf-Witten theory $\mathfrak{Z}(Vec_{S_3})$.[7] The associated Drinfeld center has simple lines labeled by conjugacy classes of $S_3$ ($[id], [a], [b]$) and the irreducible representations of the centralizers of elements in these conjugacy classes:

$$\mathcal{Z}(Vec_{S_3}) = \left\{ Q_{[id],1}, Q_{[id],P}, Q_{[id],E}, Q_{[a],1}, Q_{[a],\omega}, Q_{[a],\omega^2}, Q_{[b],+}, Q_{[b],-} \right\}, \tag{102}$$

with $\omega = e^{\pm 2\pi i/3}$. The Drinfeld center has four Lagrangians given by

$$
\begin{aligned}
\mathcal{L}_{Vec} &= Q_{[id],1} \oplus Q_{[id],P} \oplus 2Q_{[id],E}, & \mathcal{L}_{reg} = \mathcal{L}_{Rep(S_3)} &= Q_{[id],1} \oplus Q_{[a],1} \oplus Q_{[b],+}, \\
\mathcal{L}_{\mathbb{Z}_2} &= Q_{[id],1} \oplus Q_{[id],E} \oplus Q_{[b],+}, & \mathcal{L}_{\mathbb{Z}_3} &= Q_{[id],1} \oplus Q_{[id],P} \oplus 2Q_{[a],1},
\end{aligned}
\tag{103}
$$

where the Lagrangian algebra $\mathcal{L}_a$ (or equivalently the 2-charge $Q_2^a$) are labelled by the module category separating $\mathcal{L}_{Rep(S_3)}$ and $\mathcal{L}_a$.

By fixing the symmetry boundary $B^{sym} = \mathcal{L}_{Rep(S_3)}$, we find the four $Rep(S_3)$-symmetric gapped phases by varying $\mathfrak{B}^{phys}$ of the SymTFT sandwich:

$$
\begin{aligned}
(\mathcal{L}_{Rep(S_3)} | \mathcal{L}_{Vec}) &= \text{Trivial phase}, & (\mathcal{L}_{Rep(S_3)} | \mathcal{L}_{Rep(S_3)}) &= Rep(S_3) \text{ SSB Phase}, \\
(\mathcal{L}_{Rep(S_3)} | \mathcal{L}_{\mathbb{Z}_2}) &= \mathbb{Z}_2 \text{ SSB Phase}, & (\mathcal{L}_{Rep(S_3)} | \mathcal{L}_{\mathbb{Z}_3}) &= Rep(S_3)/\mathbb{Z}_2 \text{ SSB Phase}.
\end{aligned}
\tag{104}
$$

---

[7] This SymTFT can be obtained from the 3d $\mathbb{Z}_3$ DW gauge theory by gauging the $\mathbb{Z}_2$ outer automorphism which exchanges the pairs $(e, m) \leftrightarrow (e^2, m^2)$.

**Module categories.** We have discussed the odule categories in (12). They are in 1-1 correspondence with Lagrangians. We furthmore indicate how the symmetry $\text{Rep}(S_3)$ acts on these module categories:

- $\mathcal{M}_{\text{Vec}} = \text{Vec} = \{m_0\}$ with trivial $S_3$ symmetry action and $\text{Rep}(S_3)$ action:

$$P: \quad m_0 \to m_0 \,, \qquad E: \quad m_0 \to 2m_0 \,. \tag{105}$$

- $\mathcal{M}_{\mathbb{Z}_2} = \text{Vec}_{\mathbb{Z}_2} = \{m_1, m_P\}$ with $\text{Rep}(S_3)$ symmetry action

$$P: \quad m_1 \leftrightarrow m_P \,, \qquad E: \quad m_1, m_P \to m_1 \oplus m_P \,. \tag{106}$$

- $\mathcal{M}_{\mathbb{Z}_3} = \text{Vec}_{\mathbb{Z}_3} = \{m_1, m_\omega, m_{\omega^2}\}$ with $S_3$ symmetry action

$$a: \quad m_{\omega^i} \to m_{\omega^{i+1}} \,, \qquad b: \quad m_{\omega^i} \to m_{\omega^i} \,, \tag{107}$$

and $\text{Rep}(S_3)$ symmetry action

$$P: \quad m_{\omega^i} \to m_{\omega^i} \,, \qquad E: \quad m_{\omega^i} \to m_{\omega^{i+1}} \oplus m_{\omega^{i+2}} \,. \tag{108}$$

Notice here that acting with $E$ from one side is equivalent to acting with $a + a^2$ from the other.

- $\mathcal{M}_{\text{Rep}(S_3)} = \mathcal{M}_{\text{reg}} = \text{Rep}(S_3) = \{m_1, m_P, m_E\}$ with $\text{Rep}(S_3)$ symmetry action

$$\begin{aligned} P: \quad & m_1 \leftrightarrow m_P \,, \qquad m_E \to m_E \,, \\ E: \quad & m_1, m_P \to m_E \,, \qquad m_E \to m_1 \oplus m_P \oplus m_E \,. \end{aligned} \tag{109}$$

There is also one more module category that we will need which is the regular module category for $S_3$ symmetry:

- $\mathcal{M}_{S_3} = \text{Vec}_{S_3} = \{m_1, m_a, m_{a^2}, m_b, m_{ab}, m_{a^2b}\}$ with $S_3$ symmetry action following the standard $S_3$ group multiplication rules.

**Trivial phase.** In the trivial phase, we find one vacuum $v_0$ and correspondingly one boundary condition $\mathfrak{B}^0$, while the symmetry lines trivialize as

$$P = 1 \,, \qquad E = 1 \oplus 1 \,. \tag{110}$$

By varying $\mathbf{Q}_2$ we find four Boundary SymTFTs depicted in figure 10.

**a)** $(\mathcal{L}_{\text{Rep}(S_3)} \mid \mathcal{L}_{\text{Vec}} \mid \mathcal{L}_{\text{Vec}})$: In this phase we find 6 singlet boundary conditions under $\text{Rep}(S_3)$. The difference between the BCs is captured in how the $\text{Rep}(S_3)$ lines end on the quantum mechanical systems living on the boundary (the 2d phase is trivial so the boundaries are QM systems). Recall that the similar case for non-anomalous $G$ symmetry leads to QM systems whose Hilbert spaces are irreducible representations of $G$.

Let $\mathfrak{B}$ be any of these singlet BCs. On all of them the action of $\text{Rep}(S_3)$ is

$$\begin{aligned} P \otimes \mathfrak{B} &= \mathfrak{B} \,, \\ E \otimes \mathfrak{B} &= \mathfrak{B} \oplus \mathfrak{B} \,. \end{aligned} \tag{111}$$

Three of these cases involve the end of $P$ line being realized as the number $+1$ on the trivial QM $\mathfrak{B}$. These three cases are distinguished by the end of $E$ being realized as the map

$$\begin{pmatrix} \omega^i \\ \omega^{2i} \end{pmatrix}: \quad \mathfrak{B} \to \mathfrak{B} \oplus \mathfrak{B} \,. \tag{112}$$

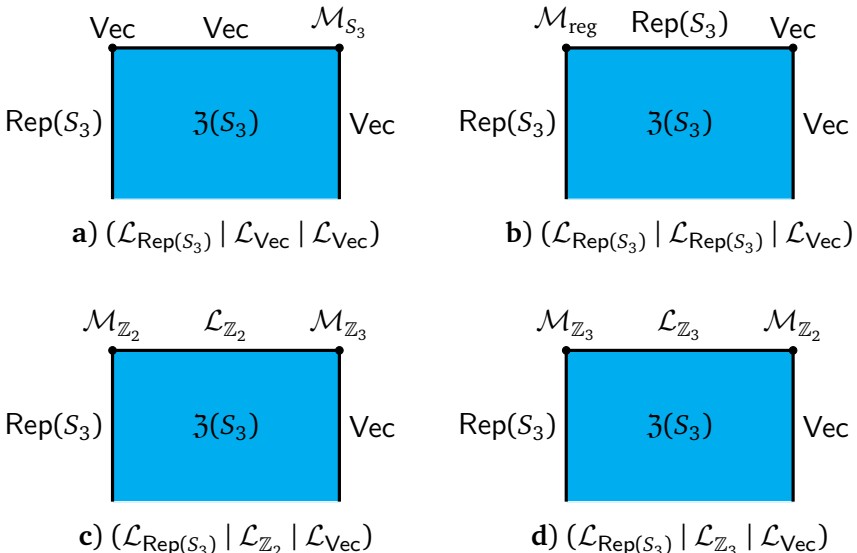

Figure 10: Boundary SymTFTs for the trivial gapped phase with $\mathrm{Rep}(S_3)$ symmetry.

The remaining three cases involve the end of $P$ line being realized as the number $-1$ on the trivial QM $\mathfrak{B}$. These three cases are distinguished by the end of $E$ being realized as the map

$$\begin{pmatrix} -\omega^i \\ \omega^{2i} \end{pmatrix}: \ \mathfrak{B} \to \mathfrak{B} \oplus \mathfrak{B}. \tag{113}$$

**b)** $(\mathcal{L}_{\mathrm{Rep}(S_3)} \mid \mathcal{L}_{\mathrm{Rep}(S_3)} \mid \mathcal{L}_{\mathrm{Vec}})$: There is a triplet of BCs transforming in $\mathcal{M}_{\mathrm{reg}}$ under $\mathrm{Rep}(S_3)$. However, as the underlying phase is trivial with one boundary invariant $\mathfrak{B}^0$, there is symmetry enrichment of the boundary such that there are now three boundaries

$$\mathfrak{B}^{1,0}, \mathfrak{B}^{P,0} = \mathfrak{B}^0, \qquad \mathfrak{B}^{E,0} = \mathfrak{B}^0 \oplus \mathfrak{B}^0, \tag{114}$$

with the $\mathrm{Rep}(S_3)$ symmetry action on these boundaries (or elements of the $\mathcal{M}_{\mathrm{reg}}$ module category according to (109)) sending

$$\begin{aligned} P: \qquad & \mathfrak{B}^{1,0} \leftrightarrow \mathfrak{B}^{P,0}, \qquad \mathfrak{B}^{E,0} \to \mathfrak{B}^{E,0}, \\ E: \quad & \mathfrak{B}^{1,0}, \mathfrak{B}^{P,0} \to \mathfrak{B}^{E,0}, \qquad \mathfrak{B}^{E,0} \to \mathfrak{B}^{1,0} \oplus \mathfrak{B}^{P,0} \oplus \mathfrak{B}^{E,0}, \end{aligned} \tag{115}$$

which is consistent with the symmetry action of the trivial phase in (110). Notice that the boundary $\mathfrak{B}^{E,0}$ has to be a composite of two boundaries $\mathfrak{B}^0$ for the symmetry action to be consistent.

**c)** $(\mathcal{L}_{\mathrm{Rep}(S_3)} \mid \mathcal{L}_{\mathbb{Z}_2} \mid \mathcal{L}_{\mathrm{Vec}})$: There are 3 doublets transforming in $\mathcal{M}_{\mathbb{Z}_2}$ under $\mathrm{Rep}(S_3)$, we can label the boundaries as

$$\mathfrak{B}^{1,\omega^i}, \mathfrak{B}^{P,\omega^i} = \mathfrak{B}^0, \tag{116}$$

where $i = 0, 1, 2$, with symmetry acting only on the first (symmetry) index, with the second labelling different multiplets transforming in the same 1-charge,

$$P: \quad \mathfrak{B}^{1,\omega^i} \leftrightarrow \mathfrak{B}^{P,\omega^i}, \qquad E: \quad \mathfrak{B}^{1,\omega^i}, \mathfrak{B}^{P,\omega^i} \to \mathfrak{B}^{1,\omega^i} \oplus \mathfrak{B}^{P,\omega^i}, \tag{117}$$

which is again consistent with (106) and (110). The three doublets are distinguished by the ends of $E$. There are four ends of $E$ that we label as $\mathcal{O}_{ab}$ for $a, b \in \{1, P\}$. The end $\mathcal{O}_{ab}$ takes $\mathfrak{B}^{a,\omega^i}$ to $\mathfrak{B}^{b,\omega^i}$. We have

$$[\mathcal{O}_{ab}] = \frac{1}{2} \begin{pmatrix} \omega^i + \omega^{2i} & \omega^i - \omega^{2i} \\ \omega^i - \omega^{2i} & \omega^i + \omega^{2i} \end{pmatrix}. \tag{118}$$

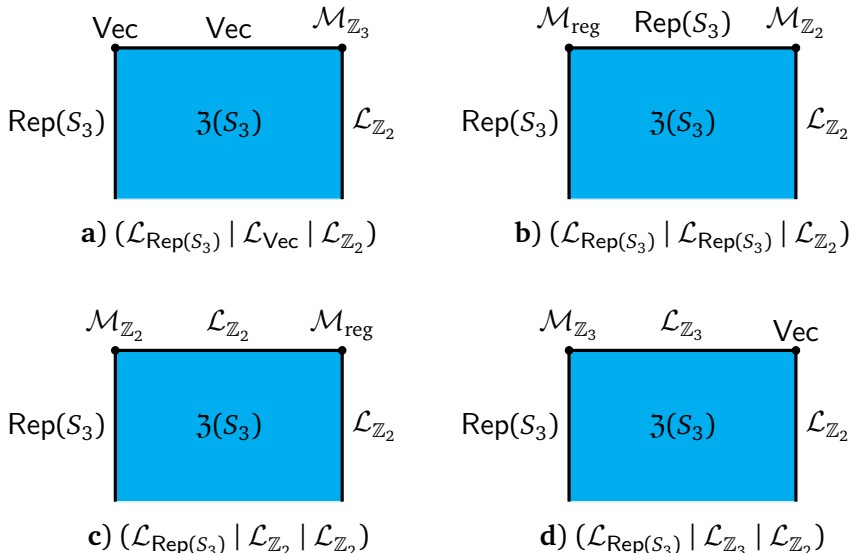

Figure 11: Four Boundary SymTFTs for the $\mathbb{Z}_2$ SSB phase of $\mathrm{Rep}(S_3)$, with boundary.

**d)** $(\mathcal{L}_{\mathrm{Rep}(S_3)} \mid \mathcal{L}_{\mathbb{Z}_3} \mid \mathcal{L}_{\mathrm{Vec}})$: Finally, this case is analogous to the one just above but with the roles of $\mathbb{Z}_2$ and $\mathbb{Z}_3$ switched, with us now having two triplets in $\mathcal{M}_{\mathbb{Z}_3}$ under $\mathrm{Rep}(S_3)$, we can similarly label the boundary elements as

$$\mathfrak{B}^{\omega^i, P^x} = \mathfrak{B}^0 \,, \tag{119}$$

with $i = 0, 1, 2$, and $x = 0, 1$, with $P^2 = 1$ and the symmetry action on these elements

$$P: \quad \mathfrak{B}^{\omega^i, P^x} \to \mathfrak{B}^{\omega^i, P^x} \,, \qquad E: \quad \mathfrak{B}^{\omega^i, P^x} \to \mathfrak{B}^{\omega^{i+1}, P^x} \oplus \mathfrak{B}^{\omega^{i+2}, P^x} \,, \tag{120}$$

which is consistent with (108) and (110). The two triplets are distinguished by the ends of $P$ being $\pm 1$.

**$\mathbb{Z}_2$ SSB phase.** In the $\mathbb{Z}_2$ SSB phase there are two vacua $v_1$ and $v_P$ and correspondingly two irreducible boundary conditions $\mathfrak{B}^1$ and $\mathfrak{B}^P$ which are exchanged by the broken subsymmetry $\mathbb{Z}_2 \subset \mathrm{Rep}(S_3)$, generated by $P$. The non-invertible $E$ symmetry line can be identified as

$$E = 1 \oplus P \,. \tag{121}$$

By varying the symmetry module (or equivalently $\mathbf{Q}_2$), one can track four cases of Boundary SymTFTs, as shown in figure 11.

**a)** $(\mathcal{L}_{\mathrm{Rep}(S_3)} \mid \mathcal{L}_{\mathrm{Vec}} \mid \mathcal{L}_{\mathbb{Z}_2})$: This case describes three singlets $\mathcal{M}_{\mathrm{Vec}}$ under $\mathrm{Rep}(S_3)$. However, the $\mathbb{Z}_2$ SSB phase has two BCs $\mathfrak{B}^1$ and $\mathfrak{B}^P$, neither of which is invariant under $\mathrm{Rep}(S_3)$. Hence each of the three boundaries in this phase must be a composite boundary

$$\mathfrak{B}^{0, \omega^i} = \mathfrak{B}^1 \oplus \mathfrak{B}^P \,. \tag{122}$$

The three singlets are distinguished by the ends of $E$ along $\mathfrak{B}^{0, \omega^i}$ which are the same as in (118).

**b)** $(\mathcal{L}_{\mathrm{Rep}(S_3)} \mid \mathcal{L}_{\mathrm{Rep}(S_3)} \mid \mathcal{L}_{\mathbb{Z}_2})$: Here we have two triplets $\mathcal{M}_{\mathrm{reg}}$ under $\mathrm{Rep}(S_3)$ which we can identify in terms of boundaries of the $\mathbb{Z}_2$ SSB phase $\mathfrak{B}^1$ and $\mathfrak{B}^P$ as

$$\mathfrak{B}^{1, P^x} = \mathfrak{B}^1 \,, \qquad \mathfrak{B}^{P, P^x} = \mathfrak{B}^P \,, \qquad \mathfrak{B}^{E, P^x} = \mathfrak{B}^1 \oplus \mathfrak{B}^P \,, \tag{123}$$

with $x = 0, 1$, which has consistent $\mathrm{Rep}(S_3)$ symmetry action with (109) and (121) as

$$P: \qquad \mathfrak{B}^{1,P^x} \leftrightarrow \mathfrak{B}^{P,P^x}, \qquad \mathfrak{B}^{E,P^x} \to \mathfrak{B}^{E,P^x},$$
$$E: \quad \mathfrak{B}^{1,P^x}, \mathfrak{B}^{P,P^x} \to \mathfrak{B}^{E,P^x}, \qquad \mathfrak{B}^{E,P^x} \to \mathfrak{B}^{1,P^x} \oplus \mathfrak{B}^{P,P^x} \oplus \mathfrak{B}^{E,P^x}. \tag{124}$$

The two triplets are distinguished by the sign of the end of $P$ along $\mathfrak{B}^{E,P^x}$.

**c)** $(\mathcal{L}_{\mathrm{Rep}(S_3)} \mid \mathcal{L}_{\mathbb{Z}_2} \mid \mathcal{L}_{\mathbb{Z}_2})$: There are 3 doublets transforming naturally under $\mathcal{M}_{\mathbb{Z}_2}$ in the $\mathbb{Z}_2$ SSB phase which we can label

$$\mathfrak{B}^{1,i} = \mathfrak{B}^1, \qquad \mathfrak{B}^{P,i} = \mathfrak{B}^P, \tag{125}$$

with multiplicity given by $i \in \{0, 1, 2\}$, and natural transformation under $\mathbb{Z}_2 \subset \mathrm{Rep}(S_3)$:

$$P: \quad \mathfrak{B}^{1,i} \leftrightarrow \mathfrak{B}^{P,i}, \qquad E: \quad \mathfrak{B}^{1,i}, \mathfrak{B}^{P,i} \to \mathfrak{B}^{1,i} \oplus \mathfrak{B}^{P,i}, \tag{126}$$

which is consistent with (106) and (121). These are distinguished by ends of $E$ between boundaries $\mathfrak{B}^{1,i}$ and $\mathfrak{B}^{P,i}$. The expressions for these ends is as in (118).

**d)** $(\mathcal{L}_{\mathrm{Rep}(S_3)} \mid \mathcal{L}_{\mathbb{Z}_3} \mid \mathcal{L}_{\mathbb{Z}_2})$: In this phase, we find a single triplet $\mathcal{M}_{\mathbb{Z}_3}$ under $\mathrm{Rep}(S_3)$. However, as the boundaries of the $\mathbb{Z}_2$ SSB phase $\mathfrak{B}^1$ and $\mathfrak{B}^P$ do not have natural transformation properties under $\mathbb{Z}_3$ or $\mathrm{Rep}(S_3)/Z_2$, we find the only possibility for the triplet to be

$$\mathfrak{B}^{\omega^i,0} = \mathfrak{B}^1 \oplus \mathfrak{B}^P, \tag{127}$$

with the enriched symmetry implemented to be consistent with (108) and (121) as

$$P: \quad \mathfrak{B}^{\omega^i,0} \to \mathfrak{B}^{\omega^i,0}, \qquad E: \quad \mathfrak{B}^{\omega^i,0} \to \mathfrak{B}^{\omega^{i+1},0} \oplus \mathfrak{B}^{\omega^{i+2},0}. \tag{128}$$

**$\mathrm{Rep}(S_3)/\mathbb{Z}_2$ SSB phase.** In the $\mathrm{Rep}(S_3)/\mathbb{Z}_2$ SSB phase there are three vacua $v_1$, $v_\omega$ and $v_{\omega^2}$, and correspondingly three irreducible boundary conditions $\mathfrak{B}^1, \mathfrak{B}^\omega$ and $\mathfrak{B}^{\omega^2}$ which are exchanged by the broken subsymmetry $\mathrm{Rep}(S_3)/Z_2 \subset \mathrm{Rep}(S_3)$, generated by the non-invertible line $E$, whereas the $\mathbb{Z}_2$ symmetry generated by $P = 1$ remains unbroken. One can indeed naturally label the boundaries by $\mathbb{Z}_3$ irreducible representations $\omega^i$ for $i = 0, 1, 2$, as the action of $E$ in this phase can be expressed using $\mathbb{Z}_3$ symmetry generators $a$ and $a^2$

$$E = a \oplus a^2: \quad \mathfrak{B}^{\omega^i} \to \mathfrak{B}^{\omega^{i+1}} \oplus \mathfrak{B}^{\omega^{i+2}}, \tag{129}$$

as discussed previously around (108). The four Boundary SymTFTs are shown in figure 12.

**a)** $(\mathcal{L}_{\mathrm{Rep}(S_3)} \mid \mathcal{L}_{\mathrm{Vec}} \mid \mathcal{L}_{\mathbb{Z}_3})$: This case describes two singlets $\mathcal{M}_{\mathrm{Vec}}$ under $\mathrm{Rep}(S_3)$. However, the $\mathrm{Rep}(S_3)/Z_2$ SSB phase has three BCs $\mathfrak{B}^1$, $\mathfrak{B}^\omega$ and $\mathfrak{B}^{\omega^2}$, neither of which is invariant under the full $\mathrm{Rep}(S_3)$. Hence both singlet boundaries in this phase must be composite boundaries

$$\mathfrak{B}^{0,P^x} = \mathfrak{B}^1 \oplus \mathfrak{B}^\omega \oplus \mathfrak{B}^{\omega^2}, \tag{130}$$

for $x = 0, 1$, and have consistent $\mathrm{Rep}(S_3)$ symmetry action with (105) and (129) as

$$P: \quad \mathfrak{B}^{0,P^x} \to \mathfrak{B}^{0,P^x}, \qquad E: \quad \mathfrak{B}^{0,P^x} \to 2\mathfrak{B}^{0,P^x}. \tag{131}$$

The two singlets are distinguished by the sign of the end of $P$ along these BCs.

**b)** $(\mathcal{L}_{\mathrm{Rep}(S_3)} \mid \mathcal{L}_{\mathrm{Rep}(S_3)} \mid \mathcal{L}_{\mathbb{Z}_3})$: Here we have three triplets $\mathcal{M}_{\mathrm{reg}}$ under $\mathrm{Rep}(S_3)$ which we can identify in terms of boundaries of the $\mathrm{Rep}(S_3)/Z_2$ SSB phase $\mathfrak{B}^{\omega^i}$ as

$$\mathfrak{B}^{1,\omega^i} = \mathfrak{B}^1, \qquad \mathfrak{B}^{P,\omega^i} = \mathfrak{B}^1, \qquad \mathfrak{B}^{E,\omega^i} = \mathfrak{B}^\omega \oplus \mathfrak{B}^{\omega^2}, \tag{132}$$

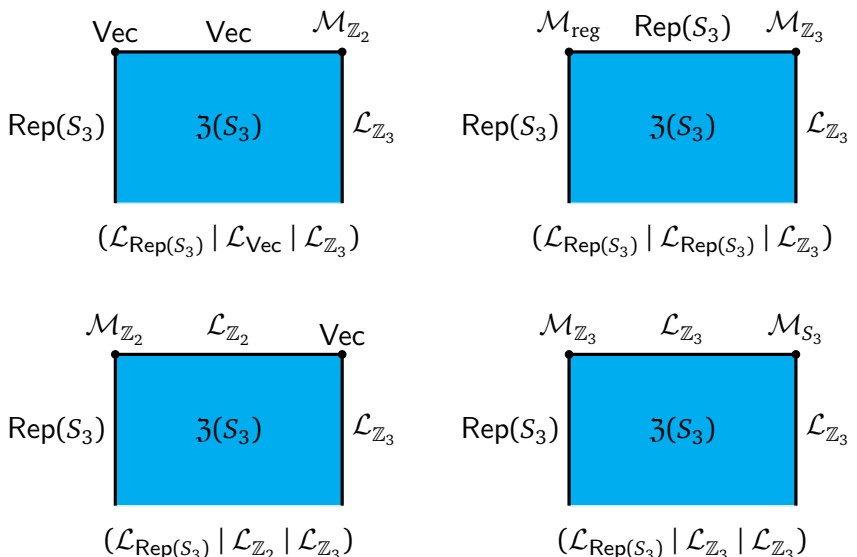

Figure 12: Four Boundary SymTFTs describing a (1+1)d $\text{Rep}(S_3)/\mathbb{Z}_2$ SSB phase with $\text{Rep}(S_3)$ symmetry.

which has consistent $\text{Rep}(S_3)$ symmetry action with (109) and (129) as

$$
\begin{aligned}
P: \quad & \mathfrak{B}^{1,\omega^i} \longleftrightarrow \mathfrak{B}^{P,\omega^i}, \qquad \mathfrak{B}^{E,\omega^i} \to \mathfrak{B}^{E,\omega^i}, \\
E: \quad & \mathfrak{B}^{1,\omega^i}, \mathfrak{B}^{P,\omega^i} \to \mathfrak{B}^{E,\omega^i}, \qquad \mathfrak{B}^{E,\omega^i} \to \mathfrak{B}^{1,\omega^i} \oplus \mathfrak{B}^{P,\omega^i} \oplus \mathfrak{B}^{E,\omega^i},
\end{aligned} \tag{133}
$$

particularly as

$$
\begin{aligned}
E: \quad & \mathfrak{B}^{1,\omega^i}, \mathfrak{B}^{P,\omega^i} \sim \mathfrak{B}^1 \to \mathfrak{B}^{\omega} \oplus \mathfrak{B}^{\omega^2} \sim \mathfrak{B}^{E,\omega^i}, \\
& \mathfrak{B}^{E,\omega^i} \sim \mathfrak{B}^{\omega} \oplus \mathfrak{B}^{\omega^2} \to 2\mathfrak{B}^1 \oplus \mathfrak{B}^{\omega} \oplus \mathfrak{B}^{\omega^2} \sim \mathfrak{B}^{1,\omega^i} \oplus \mathfrak{B}^{P,\omega^i} \oplus \mathfrak{B}^{E,\omega^i}.
\end{aligned} \tag{134}
$$

The three triplets can be distinguished by studying the ends of $E$ line along $\mathfrak{B}^{E,\omega^i}$ which take the form (118).

**c)** $(\mathcal{L}_{\text{Rep}(S_3)} \mid \mathcal{L}_{\mathbb{Z}_2} \mid \mathcal{L}_{\mathbb{Z}_3})$: In this phase, we find a doublet $\mathcal{M}_{\mathbb{Z}_2}$ under $\text{Rep}(S_3)$. However, as the boundaries of the $\text{Rep}(S_3)/Z_2$ SSB phase $B^{\omega^i}$ do not have natural transformation properties under $\mathbb{Z}_2$, we find the only possibility for the doublet to be

$$
\mathfrak{B}^{P^x,0} = \mathfrak{B}^1 \oplus \mathfrak{B}^{\omega} \oplus \mathfrak{B}^{\omega^2}, \tag{135}
$$

with the enriched symmetry implemented to be consistent with (106) and (129) as

$$
P: \quad \mathfrak{B}^{1,0} \longleftrightarrow \mathfrak{B}^{P,0}, \qquad E: \quad \mathfrak{B}^{P^x,0} \to \mathfrak{B}^{1,0} \oplus \mathfrak{B}^{P,0}. \tag{136}
$$

**d)** $(\mathcal{L}_{\text{Rep}(S_3)} \mid \mathcal{L}_{\mathbb{Z}_3} \mid \mathcal{L}_{\mathbb{Z}_3})$: In this phase, we find 18 boundaries transforming naturally in the triplet $\mathcal{M}_{\mathbb{Z}_3}$ under $\text{Rep}(S_3)$ in the $\text{Rep}(S_3)/Z_2$ SSB phase which we can label

$$
\mathfrak{B}^{\omega^i,g} = \mathfrak{B}^{\omega^i}, \tag{137}
$$

with $i = 0, 1, 2$, multiplicity given by $g \in S_3$, and natural transformation under $\text{Rep}(S_3)/\mathbb{Z}_2 \subset \text{Rep}(S_3)$ consistent with (108) and (129):

$$
P: \quad \mathfrak{B}^{\omega^i,g} \to \mathfrak{B}^{\omega^i,g}, \qquad E: \quad \mathfrak{B}^{\omega^i,g} \to \mathfrak{B}^{\omega^{i+1},g} \oplus \mathfrak{B}^{\omega^{i+2},g}. \tag{138}
$$

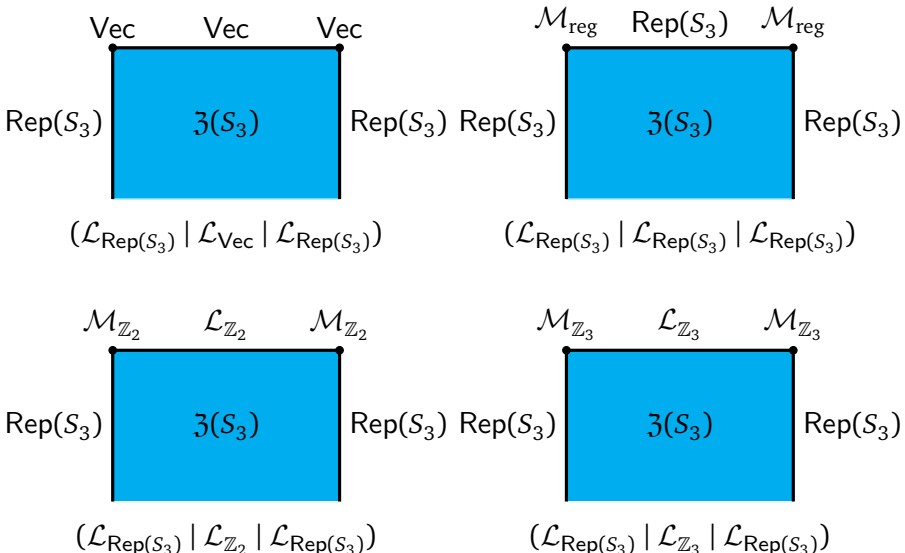

Figure 13: Four Boundary SymTFTs for the (1+1)d $\mathrm{Rep}(S_3)$ SSB phase of $\mathrm{Rep}(S_3)$.

**$\mathrm{Rep}(S_3)$ SSB phase.** In the $\mathrm{Rep}(S_3)$ phase there are three vacua similarly to the $\mathrm{Rep}(S_3)/Z_2$ SSB phase case but now the remaining $\mathbb{Z}_2$ is also spontaneously broken leading to physically distinguishable vacua. The corresponding irreducible three boundaries are $\mathfrak{B}^1$, $\mathfrak{B}^P$, $\mathfrak{B}^E$, labelled by the elements of $\mathrm{Rep}(S_3)$ as now these boundaries transform according to the $\mathrm{Rep}(S_3)$ fusion rules as in (109).

By varying the module categories (or equivalently $\mathbf{Q}_2$), there are again four cases of Boundary SymTFTs shown in figure 13).

**a)** $(\mathcal{L}_{\mathrm{Rep}(S_3)} \mid \mathcal{L}_{\mathrm{Vec}} \mid \mathcal{L}_{\mathrm{reg}})$: This case describes a single boundary forming a singlet $\mathcal{M}_{\mathrm{Vec}}$ under $\mathrm{Rep}(S_3)$. However, the $\mathrm{Rep}(S_3)$ SSB phase has three BCs $\mathfrak{B}^1$, $\mathfrak{B}^P$ and $\mathfrak{B}^E$, neither of which is invariant under $\mathrm{Rep}(S_3)$. Hence the singlet boundary in this phase must be a composite boundary

$$\mathfrak{B}^{0,0} = \mathfrak{B}^1 \oplus \mathfrak{B}^P \oplus 2\mathfrak{B}^E\,, \tag{139}$$

and have consistent $\mathrm{Rep}(S_3)$ symmetry action with (105) and (109) as

$$P:\quad \mathfrak{B}^{0,0} \to \mathfrak{B}^{0,0}\,, \qquad E:\quad \mathfrak{B}^{0,0} \to 2\mathfrak{B}^{0,0}\,, \tag{140}$$

particularly as

$$E:\quad \mathfrak{B}^{0,0} = \mathfrak{B}^1 \oplus \mathfrak{B}^P \oplus 2\mathfrak{B}^E \to \mathfrak{B}^E \oplus \mathfrak{B}^E \oplus 2(\mathfrak{B}^1 \oplus \mathfrak{B}^P \oplus \mathfrak{B}^E) = 2\mathfrak{B}^{0,0}\,. \tag{141}$$

The form of the singlet $\mathfrak{B}^{0,0}$ in (139) is not accidental as it represents the Lagrangian algebra $\mathcal{L}_{\mathrm{Vec}}$ which is the symmetry boundary of the SymTFT for $S_3$ symmetry.

**b)** $(\mathcal{L}_{\mathrm{Rep}(S_3)} \mid \mathcal{L}_{\mathrm{Rep}(S_3)} \mid \mathcal{L}_{\mathrm{reg}})$: Here we have a set of 9 boundaries transforming naturally in the triplet $\mathcal{M}_{\mathrm{reg}}$ under $\mathrm{Rep}(S_3)$ which we can label as

$$\mathfrak{B}^{R,R'} = \mathfrak{B}^R\,, \tag{142}$$

for $R, R' = 1, P, E$, which is consistent with the natural $\mathrm{Rep}(S_3)$ symmetry action of (109)

$$\begin{aligned} P:&\quad \mathfrak{B}^{1,R'} \leftrightarrow \mathfrak{B}^{P,R'}\,, \qquad \mathfrak{B}^{E,R'} \to \mathfrak{B}^{E,R'}\,, \\ E:&\quad \mathfrak{B}^{1,R'}, \mathfrak{B}^{P,R'} \to \mathfrak{B}^{E,R'}\,, \qquad \mathfrak{B}^{E,R'} \to \mathfrak{B}^{1,R'} \oplus \mathfrak{B}^{P,R'} \oplus \mathfrak{B}^{E,R'}\,. \end{aligned} \tag{143}$$

**c)** $(\mathcal{L}_{\mathsf{Rep}(S_3)} \mid \mathcal{L}_{\mathbb{Z}_2} \mid \mathcal{L}_{\mathsf{Rep}(S_3)})$: In this phase, we find four boundaries transforming in the doublet $\mathcal{M}_{\mathbb{Z}_2}$ under $\mathsf{Rep}(S_3)$. However, even though the boundaries $\mathfrak{B}^1$ and $\mathfrak{B}^P$ of the $\mathsf{Rep}(S_3)$ SSB phase have natural transformation properties under $\mathbb{Z}_2$, the last boundary $\mathfrak{B}^E$ is an odd man out as it is invariant under the $\mathbb{Z}_2$. We find the only possibility for the doublet to be

$$\mathfrak{B}^{1,P^y} = \mathfrak{B}^1 \oplus \mathfrak{B}^E, \qquad \mathfrak{B}^{P,P^y} = \mathfrak{B}^P \oplus \mathfrak{B}^E, \tag{144}$$

with multiplicity given by $y = 0, 1$, and the symmetry action consistent with (106) and (109) to be

$$P: \quad \mathfrak{B}^{1,P^y} \leftrightarrow \mathfrak{B}^{P,P^y}, \qquad E: \quad \mathfrak{B}^{1,P^y}, \mathfrak{B}^{P,P^y} \to \mathfrak{B}^{1,P^y} \oplus \mathfrak{B}^{P,P^y}. \tag{145}$$

Particularly, $P$ transformation is evident as $\mathfrak{B}^1 \leftrightarrow \mathfrak{B}^P$ and $\mathfrak{B}^E$ is invariant, whereas $E$ is again a bit less clear:

$$E: \quad \mathfrak{B}^{1,P^y} \sim \mathfrak{B}^1 \oplus \mathfrak{B}^E \to \mathfrak{B}^E \oplus (\mathfrak{B}^1 \oplus \mathfrak{B}^P \oplus \mathfrak{B}^E) \sim \mathfrak{B}^{1,P^y} \oplus \mathfrak{B}^{P,P^y}, \tag{146}$$

and similarly for $\mathfrak{B}^{P,P^y}$.

Notice again that in fact combining the doublet into a singlet under the $\mathbb{Z}_2$ symmetry produces the result in (139), i.e.

$$\mathfrak{B}^{1,P^y} \oplus \mathfrak{B}^{P,P^y} = \mathfrak{B}^1 \oplus \mathfrak{B}^P \oplus 2\mathfrak{B}^E, \tag{147}$$

which is the canonical Lagrangian algebra for $S_3$ symmetry, invariant under $\mathsf{Rep}(S_3)$.

**d)** $(\mathcal{L}_{\mathsf{Rep}(S_3)} \mid \mathcal{L}_{\mathbb{Z}_3} \mid \mathcal{L}_{\mathsf{reg}})$: Finally, in this case, we find 9 boundaries transforming in the triplet $\mathcal{M}_{\mathbb{Z}_3}$ under $\mathsf{Rep}(S_3)$. However, as the boundaries of the $\mathsf{Rep}(S_3)$ SSB phase $\mathfrak{B}^1$, $\mathfrak{B}^P$, do not have natural transformation properties under $\mathbb{Z}_3$ or $\mathsf{Rep}(S_3)/Z_2$, we find the triplet to be

$$\mathfrak{B}^{1,\omega^j} = \mathfrak{B}^1 \oplus \mathfrak{B}^P, \qquad \mathfrak{B}^{\omega,\omega^j}, \mathfrak{B}^{\omega^2,\omega^j} = \mathfrak{B}^E, \tag{148}$$

with multiplicity $j = 0, 1, 2$, and consistent symmetry action with (108) and (109) as

$$P: \quad \mathfrak{B}^{\omega^i,\omega^j} \to \mathfrak{B}^{\omega^i,\omega^j}, \qquad E: \quad \mathfrak{B}^{\omega^i,\omega^j} \to \mathfrak{B}^{\omega^{i+1},\omega^j} \oplus \mathfrak{B}^{\omega^{i+2},\omega^j}, \tag{149}$$

which particularly for the $E$ action can be seen as

$$
\begin{aligned}
E: \quad & \mathfrak{B}^{1,\omega^j} \sim \mathfrak{B}^1 \oplus \mathfrak{B}^P \to 2\mathfrak{B}^E \sim \mathfrak{B}^{\omega,\omega^j} \oplus \mathfrak{B}^{\omega^2,\omega^j}, \\
& \mathfrak{B}^{\omega,\omega^j} \sim \mathfrak{B}^E \to \mathfrak{B}^1 \oplus \mathfrak{B}^P \oplus \mathfrak{B}^E \sim \mathfrak{B}^{1,\omega^j} \oplus \mathfrak{B}^{\omega^2,\omega^j}, \\
& \mathfrak{B}^{\omega^2,\omega^j} \sim \mathfrak{B}^E \to \mathfrak{B}^1 \oplus \mathfrak{B}^P \oplus \mathfrak{B}^E \sim \mathfrak{B}^{1,\omega^j} \oplus \mathfrak{B}^{\omega,\omega^j}.
\end{aligned}
\tag{150}
$$

**Boundary-changing operators.** Rather than going over all 16 cases mentioned above with various multiplicities, let us concentrate on a few interesting examples to showcase the general characteristics of boundary-changing operators for boundary phases with non-invertible symmetry. We can organize these based on the module category $\mathcal{M}$ which is acted on from the left by the symmetry lines, which forms the top left corner of the Boundary SymTFT:

- $\mathcal{M} = \mathcal{M}_{\mathsf{Vec}}$: In this case, all six $S_3$ symmetry lines can end on $\mathcal{M}$ from the right or more precisely on the singlet boundary $m_0 \in \mathsf{Vec}$. If we label these junctions on which the topological lines $a^i b^x$, with $i \in \mathbb{Z}_3$, $x \in \mathbb{Z}_2$, can end as $\phi^{a^i b^x}$ then one can show these junctions transform under $\mathsf{Rep}(S_3)$ from the left as

$$
\begin{aligned}
P: \quad & \phi^{a^i} \to \phi^{a^i}, \quad \phi^b \to -\phi^b, \\
E: \quad & \phi^a \to \omega\phi^a \oplus \omega^2\phi^{a^2}, \quad \phi^{a^2} \to \omega^2\phi^a \oplus \omega\phi^{a^2}, \quad \phi^b \to 0, \quad \phi^1 \to 2\phi^1.
\end{aligned}
\tag{151}
$$

This is consistent with $\mathsf{Rep}(S_3)$ symmetry action on topological local operators found in [38], i.e. combining these junctions based on conjugacy classes as

$$\mathcal{O}_+^a = \phi^a \oplus \phi^{a^2}, \qquad \mathcal{O}^b = \phi^b \oplus \phi^{ab} \oplus \phi^{ab^2}, \tag{152}$$

one finds the same symmetry action

$$
\begin{aligned}
P: &\quad \mathcal{O}_+^a \to \mathcal{O}_+^a, &\quad \mathcal{O}^b \to -\mathcal{O}^b, \\
E: &\quad \mathcal{O}_+^a \to -\mathcal{O}_+^a, &\quad \mathcal{O}^b \to 0,
\end{aligned} \tag{153}
$$

as $\omega + \omega^2 = -1$.

The action (151) is interesting, especially as the non-invertible line $E$ maps, for example, $\phi^a \to \omega \phi^a \oplus \omega^2 \phi^{a^2}$, thus if we take this less general example where we fix our initial boundary to be $\mathfrak{B}^{0,1}$ then the picture is



$$\tag{154}$$

Then we find that the boundary-changing operator $\Phi_{1,a}^a$ transforms under $\mathsf{Rep}(S_3)$ as

$$P: \quad \Phi_{1,a}^a \to \Phi_{1,a}^a, \qquad E: \quad \Phi_{1,a}^a \to \omega \Phi_{1,a}^a \oplus \omega^2 \Phi_{1,a^2}^{a^2}, \tag{155}$$

hence we see that even though the boundaries are in a singlet under $\mathsf{Rep}(S_3)$, the non-invertible symmetry can still transform the boundary through the action on the boundary-changing operator, which in turn can even act on the non-symmetry index, which is not possible in invertible cases.

- $\mathcal{M} = \mathcal{M}_{\mathsf{Vec}_{\mathbb{Z}_2}}$: In this case, only the $[a] = (a \oplus a^2)_{S_3}$ symmetry line (of $\mathsf{Rep}(S_3)'$) can end on $\mathcal{M}$ from the right or more precisely on the doublet of boundaries $m_1, m_P \in \mathcal{M}_{\mathbb{Z}_2}$. If we label these junctions on which the topological line $[a]$ can end as $\phi^{[a]}$ then one can show this junctions transform under $\mathsf{Rep}(S_3)$ from the left as

$$P: \quad \phi^{[a]} \to \phi^{[a]}, \qquad E: \quad \phi^{[a]} \to -\phi^{[a]}, \tag{156}$$

which is consistent with $\mathsf{Rep}(S_3)$ symmetry action on topological local operators by simply identifying $\phi^{[a]} = \mathcal{O}_+^a$.

- $\mathcal{M} = \mathcal{M}_{\mathsf{Vec}_{\mathbb{Z}_3}}$: In this case, only the $b$ symmetry line (of $S_3'$) can end on $\mathcal{M}$ from the right or more precisely on the triplet of boundaries $m_1, m_\omega, m_{\omega^2} \in \mathcal{M}_{\mathbb{Z}_3}$. If we label this junction on which the topological line $b$ can end as $\phi^b$ as previously then one can show this junction transforms under $\mathsf{Rep}(S_3)$ from the left as

$$P: \quad \phi^b \to -\phi^b, \qquad E: \quad \phi^b \to 0, \tag{157}$$

which is consistent with what we have already seen for $\mathcal{M} = \mathcal{M}_{\mathsf{Vec}}$.

- $\mathcal{M} = \mathcal{M}_{\mathrm{reg}}$: This is just the case of having the boundaries transform naturally in the regular module category where no symmetry lines can end completely from the right on $\mathcal{M}$, hence the boundary-changing operators will transform under the standard $\mathsf{Rep}(S_3)$ symmetry rules.

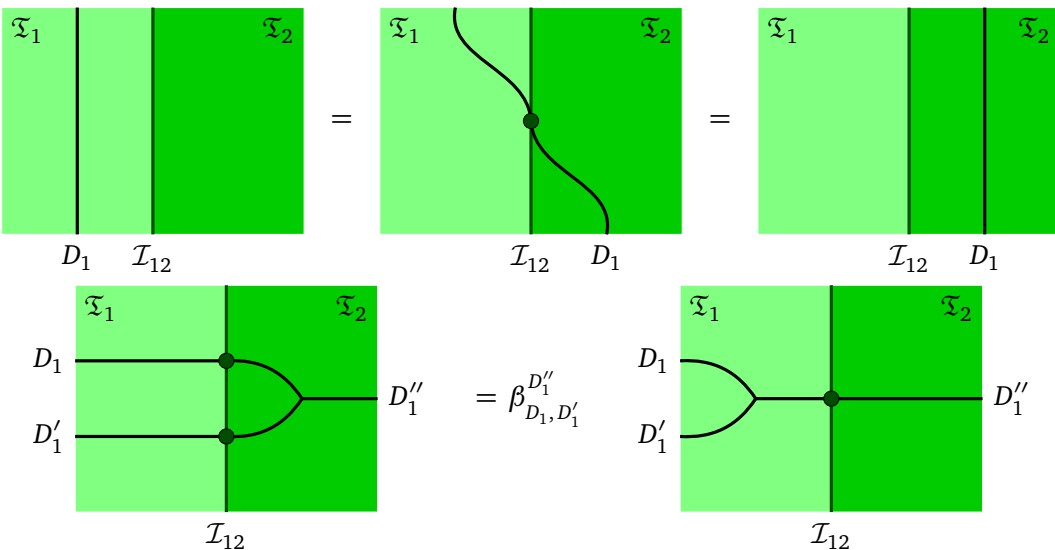

Figure 14: Above, symmetry defect $D_1$ can be freely moved through a symmetric interface $\mathcal{I}_{12}$ between two phases $\mathfrak{T}_1$ and $\mathfrak{T}_2$. Below, composition law for symmetry defects along the interface.

## 3.6 Interfaces between SPTs

As we have alluded in the Introduction, our formalism also allows the description of symmetric interfaces between phases, which we will denote as $\mathcal{I}$. Here by symmetric, we mean interfaces which are transparent to the symmetry action, as depicted in figure 14.

After folding, the symmetric interface is described by the following boundary sandwich:

$$
\mathfrak{T}_2 \bigg|_{\mathfrak{T}_1} \mathcal{I} = \mathfrak{B}^{\text{sym}}_{\mathcal{S}} \boxed{\mathfrak{Z}(\mathcal{S})} \mathcal{I} \begin{matrix} \mathfrak{B}^{\text{phys}}_{\mathfrak{T}_2} \\ \\ \mathfrak{B}^{\text{phys}}_{\mathfrak{T}_1} \end{matrix} = \mathfrak{B}^{\text{sym}}_{\mathcal{S}\boxtimes\bar{\mathcal{S}}} \boxed{\begin{matrix} \mathcal{I} & \boldsymbol{Q}^{\text{diag}}_2 & M \\ \mathfrak{Z}(\mathcal{S})\boxtimes\overline{\mathfrak{Z}(\mathcal{S})} \end{matrix}} \mathfrak{B}^{\text{phys}}_{\mathfrak{T}_1\boxtimes\mathfrak{T}_2} . \tag{158}
$$

Where $\boldsymbol{Q}^{\text{diag}}_2$ is the gapped BC corresponding to the universal diagonal Lagrangian algebra in $\mathfrak{Z}(\mathcal{S}) \boxtimes \overline{\mathfrak{Z}(\mathcal{S})}$ [84]. Further interfaces can be constructed from codimension-1 topological defects in $\mathfrak{Z}(\mathcal{S})$ which leave the gapped BC $\mathfrak{B}^{\text{sym}}$ invariant by simply stretching them through the bulk. We do not study these objects in this work.

Of special interest to us will be interfaces between gapped phases and, in particular, between invertible phases (SPTs). For invertible symmetry, such problem has a long history, while for non-invertible ones –especially for Tambara-Yamagami categories– an illuminating early study is [3], while a recent lattice construction is [85]. Of interest to use will be two key observables

- The associator for the symmetry defects on the interface. In the folded picture, these are the boundary F-symbols for the diagonal symmetry defects $D_1^{\mathcal{S}\otimes\bar{\mathcal{S}}}$.

- The number of edge modes on the interface –that is– the ground-state degeneracy of the interface Hilbert space.[8] Recall that, for an SPT for a discrete group, the symmetry

---

[8]This is defined by taking time to run in the vertical direction along the interface. Notice also that, in lattice

acts projectively on the 1d boundary, so the interface Hilbert space is always degenerate. Similarly, the nontriviality of the boundary F-symbols often forbids a nondegenerate interface Hilbert space.

The first observable has essentially been described in [24], for invertible symmetries. The main idea is the following: since the theories on both sides of the interface are SPTs, the symmetry defect can be trivialized. The symmetry action then localizes on the interface $\mathcal{I}_{12}$ and its algebra can be computed via the bulk data. To understand edge modes, we must give a construction of the interface Hilbert space $\mathcal{H}_{\mathcal{I}}$, on which the operators described above act. As usual in TFT, this is obtained by "closing" the (2+1)d bulk geometry (158). This gives rise to the following junction between gapped boundaries:

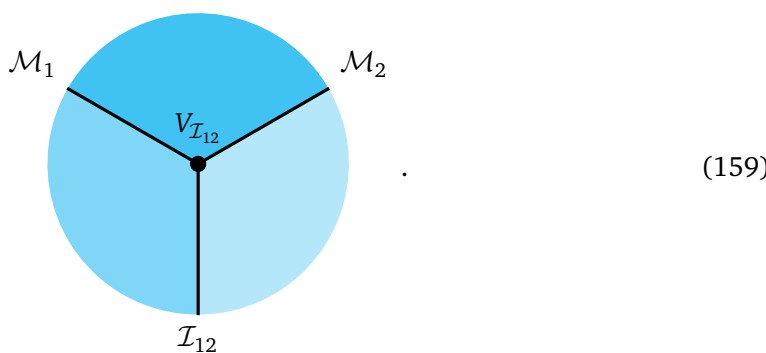

$$\tag{159}$$

The three-valent junction $V_{\mathcal{I}_{12}}$ is a vector space, which is isomorphic to the interface Hilbert space $\mathcal{H}_{\mathcal{I}_{12}}$.[9] Its dimension is in general hard to compute, but can sometimes be extracted from the description of $\mathcal{M}_1$, $\mathcal{M}_2$ and $\mathcal{I}_{12}$ as gauging interfaces using the techniques of [86]. We will now compute these data in two classes of examples, showcasing the power of the SymTFT.

**SPTs for discrete groups**    For simplicity we will study Abelian symmetries,[10] the non-Abelian generalization is also straightforward but tedious. We denote the underlying group by $G$. SPTs are classified by classes $\omega \in H^2(G, U(1))$. These are most easily described via their associated anti-symmetric bicharacter

$$\chi_\omega(a, b) = \frac{\omega(a, b)}{\omega(b, a)}, \tag{160}$$

which encodes the gauge-invariant information about $\omega$. Let us consider an interface between two SPTs, say $\text{SPT}_1$ and $\text{SPT}_2$, described by classes $\omega_1$ and $\omega_2$. By folding we obtain an interface between the SPT described by $\omega_1/\omega_2$ and the trivial theory. As discussed in section 2.2, the symmetry action on the boundary is labelled by the boundary F-symbol:

$$\hat{F}_{a,b} = \frac{\omega_1(a, b)}{\omega_2(a, b)}, \tag{161}$$

alternatively, the algebra of lines on the interface is:

$$D_1^a \otimes D_1^b = \frac{\chi_1(a, b)}{\chi_2(a, b)} D_1^b \otimes D_1^a. \tag{162}$$

---

constructions, one considers a periodic system with gapped phases $\mathfrak{T}_1$ and $\mathfrak{T}_2$ on the two halves of the circle. This way there are two insertions of the interface and the number of edge modes squares.

[9]More precisely, to its ground states.

[10]For this kind of symmetries, edge modes have also been studied in [24] using the SymTFT formulation.

Let us see how this is encoded in the SymTFT. We will follow the conventions of [39, 43] and denote a bulk anyon by $(a, \alpha) \in G \times G^\vee$. The symmetry boundary condition $\mathfrak{B}^{\text{sym}}$ is described by the magnetic algebra:

$$\mathcal{L}_{\text{sym}} = \bigoplus_\alpha (0, \alpha). \tag{163}$$

On the other hand, the SPTs are described by gapped BC [39, 76]:

$$\mathcal{L}_\omega = \bigoplus_a (a, \psi_\omega(a)), \tag{164}$$

where $\psi_\omega : G \to G^\vee$ is an invertible group homomorphism such that:

$$\psi_\omega(a)[b] = \chi_\omega(a, b). \tag{165}$$

In order to compute the commutator in the SymTFT we employ the following representation of the symmetry lines crossing the interface:

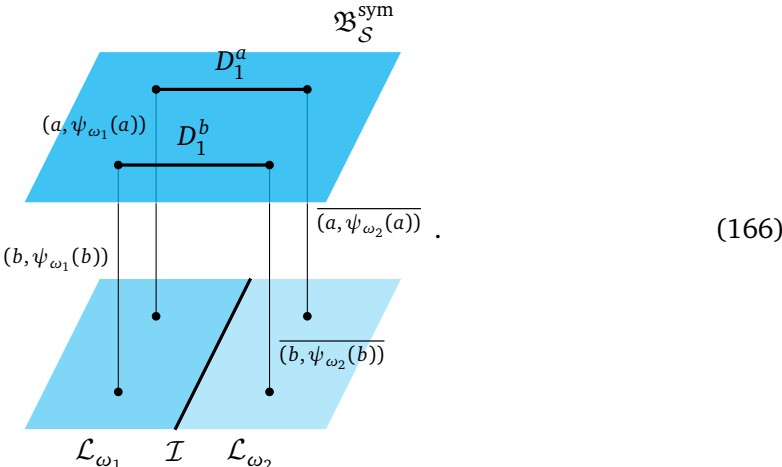

$$\tag{166}$$

Shrinking the $D_1^a$ line we obtain the anyon $(0, \psi_{\omega_1}(a) - \psi_{\omega_2}(a))$ stretching between the interface and the $\mathfrak{B}^{\text{sym}}$ boundary:

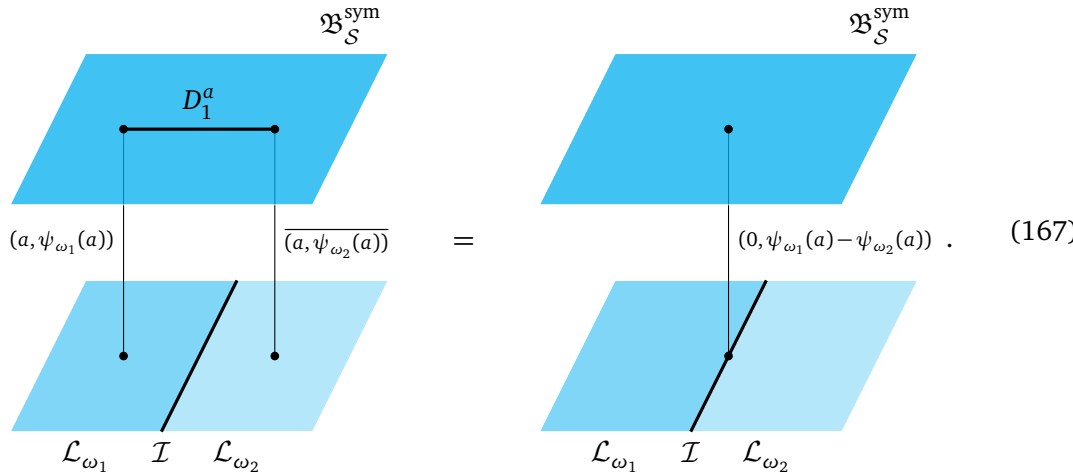

$$\tag{167}$$

Commuting this with the $D_1^b$ line gives a phase:

$$\frac{\psi_{\omega_1}(a)[b]}{\psi_{\omega_2}(a)[b]} = \frac{\chi_{\omega_1}(a, b)}{\chi_{\omega_2}(a, b)}, \tag{168}$$

reproducing the known QFT answer. Let us now focus on $G = \mathbb{Z}_2 \times \mathbb{Z}_2$, for which the interface Hilbert space is two-fold degenerate. The interfaces $\mathcal{M}_1$ and $\mathcal{M}_2$ implement the gauging of $\mathbb{Z}_2 \times \mathbb{Z}_2$, with or without discrete torsion, respectively. The same is true for $\mathcal{I}_{12}$. Since all three module categories have a single element the fusion of interfaces must be of the form:

$$\mathcal{M}_1 \otimes \mathcal{M}_2 = \dim(V_{\mathcal{I}_{12}}) \mathcal{I}_{12}, \tag{169}$$

following [86], notice that the quantum dimension on any gauging interface is:

$$d_{\mathcal{I}} = \sqrt{\dim(\mathcal{A}_{\mathcal{I}})}, \tag{170}$$

where $\mathcal{A}_{\mathcal{I}}$ is the algebra object implementing the generalized gauging. In our case $\dim(\mathcal{A}_{\mathcal{M}_1}) = \dim(\mathcal{A}_{\mathcal{M}_2}) = \dim(\mathcal{A}_{\mathcal{I}_{12}}) = 4$, so we conclude:

$$\dim(V_{\mathcal{I}_{12}}) = 2, \tag{171}$$

recovering the expected counting of edge modes.

**Non-invertible symmetry:** $\mathsf{Rep}(D_8)$    The paramount example of nontrivial interfaces between SPTs is given by the three SPTs under $\mathsf{Rep}(D_8)$ symmetry, which we dub:

$$\mathrm{SPT}_R, \mathrm{SPT}_G, \mathrm{SPT}_B. \tag{172}$$

We will denote invertible elements $g \in \mathsf{Rep}(D_8)$ by $P_1, P_2, P_1 P_2$ while $E$ will denote the non-invertible object, satisfying:

$$E^2 = 1 + P_1 + P_2 + P_1 P_2. \tag{173}$$

At the level of the invertible $\mathbb{Z}_2 \times \mathbb{Z}_2$ subcategory of $\mathsf{Rep}(D_8)$, the three SPTs are indistinguishable and identified with the $\mathbb{Z}_2 \times \mathbb{Z}_2$ SPT. They can however be distinguished via the action of the non-invertible element $E$, given by 1-cochain $\nu$ satisfying [3]:

$$\nu(g)^2 = 1, \qquad g \in \{P_1, P_2, P_1 P_2\}, \tag{174}$$

$$\nu(P_1)\nu(P_2)\nu(P_1 P_1) = -1. \tag{175}$$

the three numbers $\nu(P_1)$, $\nu(P_2)$, $\nu(P_1 P_2)$ furthermore need to satisfy:

$$\mathrm{sign}\left(1 + \nu(P_1) + \nu(P_2) + \nu(P_1 P_2)\right) = 1. \tag{176}$$

The solution is to assign a $-$ sign an element of the list $P_1, P_2, P_1 P_2$ describing three cochains $\nu_R, \nu_G, \nu_B$. The interface between two such SPTs $i$ and $j$ is characterized by the ratio of two co-chains, which we denote by $\nu_{ij} \equiv \nu_i / \nu_j$ whose values are:

Table 1: Table summary of three possible ratio combinations of 1-cochains distinguishing the three possible SPTs with $\mathsf{Rep}(D_8)$ symmetry.

|            | $P_1$ | $P_2$ | $P_1 P_2$ |
|------------|-------|-------|-----------|
| $\nu_{RG}$ | $-$   | $-$   | $+$       |
| $\nu_{GB}$ | $+$   | $-$   | $-$       |
| $\nu_{RB}$ | $-$   | $+$   | $-$       |

We will now recover this characterization from the SymTFT. Following the notation of [45] –to which we also refer the reader for further details about the structure of the Drinfeld center– the relevant Lagrangian algebras are:

$$\mathcal{L}_{\mathrm{sym}} = 1 \oplus e_{RGB} \oplus m_{GB} \oplus m_{RB} \oplus m_{RG}, \tag{177}$$

and

$$\mathcal{L}_R = 1 \oplus e_B \oplus e_G \oplus e_{BG} \oplus 2m_R \,, \tag{178}$$

$$\mathcal{L}_G = 1 \oplus e_B \oplus e_R \oplus e_{RB} \oplus 2m_G \,, \tag{179}$$

$$\mathcal{L}_B = 1 \oplus e_R \oplus e_G \oplus e_{RG} \oplus 2m_B \,. \tag{180}$$

The one-cochain can be extracted from the commutation relations between an invertible line and the non-invertible generator $E$:

$$D_1^g \otimes D_1^E = \nu_{ij}(g) D_1^E \otimes D_1^g \,. \tag{181}$$

We will focus on the interface $\mathcal{I}_{RG}$ between $\text{SPT}_R$ and $\text{SPT}_G$. The other interfaces $\mathcal{I}_{RB}$ and $\mathcal{I}_{GB}$ are obtained by permuting the $RGB$ labels. For this choice of interface the description of the symmetry action is as follows

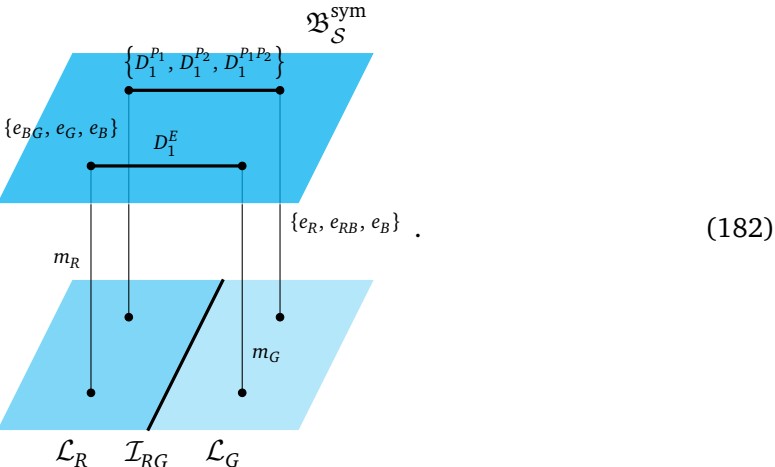

$$\tag{182}$$

Shrinking the upper line triplet $\left\{D_1^{P_1}, D_1^{P_2}, D_1^{P_1 P_2}\right\}$ gives rise to $\{e_{RGB}, e_{RGB}, 1\}$:

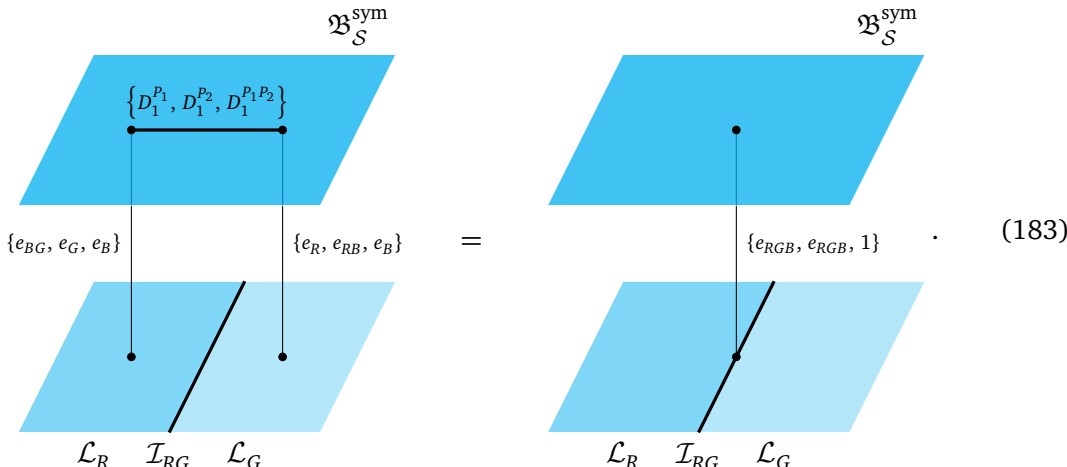

$$\tag{183}$$

Following [45], the endpoint of the $e_{RGB}$ line is charged under the $D_1^E$ symmetry, thus we find:

$$D_1^g \otimes D_1^E = \nu_{ij}(g) D_1^E \otimes D_1^g \,, \tag{184}$$

as expected. Cycling through the three interfaces we recover Table 1. Let us now describe the counting of edge modes. Since:

$$\mathcal{L}_R \cap \mathcal{L}_G = 1 \oplus e_B \,, \tag{185}$$

there are two (physically equivalent) choices for the interface $\mathcal{I}_{RG}$, let us denote them by $\mathcal{I}_{RG}^{\pm}$. Furthermore, since we are describing Fiber-Functors, it is known [86] that the algebras $\mathcal{A}_{R,G}$ take the "maximal" form:

$$\mathcal{A}_{R,G} = 1 + P_1 + P_2 + P_1 P_2 + 2E, \qquad \dim(\mathcal{A}_{R,G}) = 8. \tag{186}$$

Furthermore, $\mathcal{I}_{RG}$ implements the gauging of $\mathbb{Z}_2 \times \mathbb{Z}_2$,[11] so $\dim(\mathcal{A}_{RG}) = 4$. The fusion rule now takes the form:

$$\mathcal{M}_R \otimes \mathcal{M}_G = \dim(V_{\mathcal{I}_{RG}}) \left( \mathcal{I}_{RG}^+ \oplus \mathcal{I}_{RG}^- \right), \tag{187}$$

plugging in the numbers we again find:

$$\dim(V_{\mathcal{I}_{RG}}) = 2, \tag{188}$$

in accordance with the results of [85].

Of course, a more general exploration of these phenomena, especially in higher dimensions, would be of great interest. We believe the SymTFT to be the perfect tool for this.

### 3.7 Boundary club sandwich: Gapless phases

**General boundary SymTFT description** In the presence of categorical symmetries, (1+1)d gapless phases, describing critical points between two gapped phases, have a SymTFT description as well. More precisely, we can construct associated KT transformations, which transform a phase transition for a symmetry $\mathcal{S}'$ to one for a larger symmetry $\mathcal{S}$. In this way, only $\mathcal{S}'$ acts on the gapless degrees of freedom. This setup, also known as the "club sandwich" is defined by a symmetry and a physical boundary, as well as an interface between the topological order $\mathfrak{Z}(\mathcal{S})$ and $\mathfrak{Z}(\mathcal{S}')$. This is given in terms of a condensable –but not Lagrangian– algebra $\mathcal{A}$ in the Drinfeld center $\mathcal{Z}(\mathcal{S})$:

$$
\begin{array}{c}
\boxed{\quad \mathfrak{Z}(\mathcal{S}) \quad \big| \quad \mathfrak{Z}(\mathcal{S}') \quad} \\[2pt]
\mathfrak{B}_{\mathcal{S}}^{\text{sym}} \qquad \mathcal{A} \qquad \mathfrak{B}_{\mathfrak{T}}^{\text{phys}}
\end{array}
\tag{189}
$$

Gapless SPT (gSPT) phases are of particular interest. These are gapless phases, with a unique ground state on a circle on which only $\mathcal{S}'$ acts faithfully on the gapless degrees of freedom. They are characterized by condensable algebras $\mathcal{A}_{\text{gSPT}}$, such that

$$\mathcal{A}_{\text{gSPT}} \cap \mathcal{L}_{\mathcal{S}} = 1. \tag{190}$$

If a gSPT cannot be deformed to an SPT phase without breaking the symmetry, it is called an igSPT. These are particularly interesting gapless phases, as they correspond to symmetry-protected critical points. There are examples with group and non-invertible symmetries [24, 25, 45, 60–66] as well as in higher dimensions [42, 43]. Typically such phases are reached via RG flow from a UV, $\mathcal{S}$-symmetric theory $\mathfrak{T}^{UV}$. If we keep in mind such picture, it is natural to ask how the boundary condition multiplets $\mathfrak{B}^{\mathcal{M}}$ in the UV are mapped to $\mathcal{S}'$-symmetric multiplets $\mathfrak{B}^{\mathcal{M}'}$. This is not a trivial problem. Indeed, even for flows between unitary minimal models, the complete map between boundary conditions between the UV and IR is only understood for certain RG flows [87–89].

---

[11] A way to understand this is that $\text{Rep}(D_8)$ is Morita equivalent to a $\mathbb{Z}_2^R \times \mathbb{Z}_2^G \times \mathbb{Z}_2^B$ symmetry with 't Hooft anomaly $A_R A_G A_B$. The gapped BC corresponding to the three SPTs can be reached from the Dir boundary condition for this symmetry by gauging a single $\mathbb{Z}_2$. This follows e.g. from the discussion in [85], or can be derived directly. Thus to connect any two of these boundaries we must gauge a $\mathbb{Z}_2 \times \mathbb{Z}_2$ symmetry without discrete torsion, where one of the $\mathbb{Z}_2$s is a magnetic symmetry.

**Club-sandwich with boundary.** Adding boundaries to the club sandwich is described as follows: again there is a $Q_2$ charge that characterizes the boundary condition of the SymTFT:

$$
\begin{array}{ccccc}
\mathcal{M} & Q_2 & \mathcal{I} & Q_2' & M \\
& \mathfrak{Z}(\mathcal{S}) & & \mathfrak{Z}(\mathcal{S}_{\text{low}}) & \\
\mathfrak{B}_{\mathcal{S}}^{\text{sym}} & & \mathcal{A} & & \mathfrak{B}_{\mathfrak{T}}^{\text{phys}}
\end{array}
\tag{191}
$$

and $\mathcal{M}$ is again a module category. The $\mathcal{S}'$-symmetric boundary condition is instead described by a 1-charge $Q_2'$. The interface $\mathcal{I}$ is a trivalent junction between $Q_2$, $Q_2'$ and $\mathcal{A}$. Given $Q_2$ and $\mathcal{A}$, not all the charges $Q_2'$ are allowed to appear in the IR. To this end consider the parallel fusion:

$$
\begin{array}{ccc}
\begin{array}{cc} \\ Q_2 & \mathcal{A} \end{array} & = \sum_{Q_2'} n_{\mathcal{A}Q_2}^{Q_2'} & \begin{array}{c} \\ Q_2' \end{array}
\end{array}
\tag{192}
$$

The integer $n_{\mathcal{A}Q_2}^{Q_2'}$ counts the number of independent boundary interfaces $\mathcal{I}$, and the allowed choices of $Q_2'$ are those for which $n_{\mathcal{A}Q_2}^{Q_2'}$ is non-vanishing. The club sandwich has an equivalent description in terms of the gapped boundary condition via the folding trick:

$$
\begin{array}{ccccc}
Q_2 & \mathcal{I} & Q_2' & & Q_2 \boxtimes \overline{Q_2'} & \mathcal{I} \\
\mathfrak{Z}(\mathcal{S}) & & \mathfrak{Z}(\mathcal{S}') & = & \mathfrak{Z}(\mathcal{S}) \boxtimes \overline{\mathfrak{Z}(\mathcal{S}')} & \mathcal{L}_{\mathcal{A}} \\
& \mathcal{A} & & & &
\end{array}
\tag{193}
$$

where $\mathcal{L}_{\mathcal{A}}$ is a Lagrangian algebra of the folded topological order $\mathfrak{Z}(\mathcal{S}) \boxtimes \overline{\mathfrak{Z}(\mathcal{S}')}$, which is a lift of the algebra $\mathcal{A}$ (see [44] for details). In particular in $\mathcal{I}$ is a module category for this folded setup, describing the interface between two gapped boundary conditions $\mathcal{L}_{\mathcal{A}}$ and $Q_2 \boxtimes \overline{Q_2'}$. The physical properties of $\mathcal{I}$, allow for instance the determination of the mapping between boundary conditions and boundary changing operators from the UV, $\mathcal{S}$-symmetric theory, into the IR.

**Mapping between BC: UV to IR** Consider an $\mathcal{S}$-symmetric gapless RG flow. Typically the faithfully acting symmetry in the IR fits in a group extension

$$
1 \longrightarrow \mathcal{S}_{\text{gap}} \longrightarrow \mathcal{S} \longrightarrow \mathcal{S}' \longrightarrow 1,
\tag{194}
$$

where $\mathcal{S}$ is the full symmetry of the UV theory–which for simplicity we take to be a group– and $\mathcal{S}'$ the quotient acting on the gapless sector. This picture can be extended to arbitrary fusion categories using the SymTFT picture we introduce below.

We now consider a UV, $\mathcal{S}$-symmetric boundary condition $\mathfrak{B}^{UV}$ associated to a $\mathcal{S}$ module category $\mathcal{M}$. Its image in the $\mathcal{S}'$-symmetric theory can be extracted by pushing the $\mathcal{I}$ interface of (191) onto $\mathcal{M}$. The interface $\mathcal{I}$ thus provides a map between the set of UV boundary

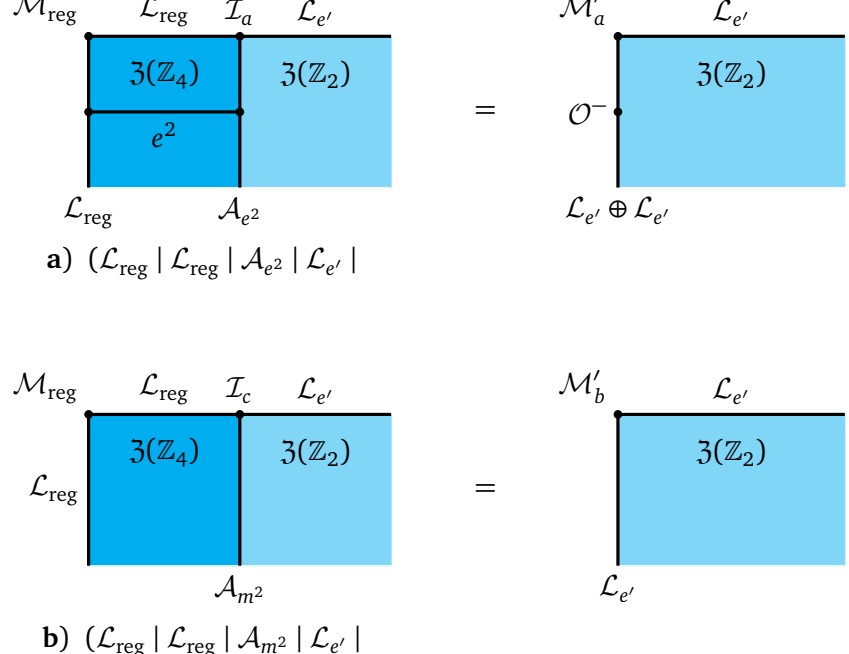

a) $(\mathcal{L}_{\text{reg}} \mid \mathcal{L}_{\text{reg}} \mid \mathcal{A}_{e^2} \mid \mathcal{L}_{e'} \mid$

b) $(\mathcal{L}_{\text{reg}} \mid \mathcal{L}_{\text{reg}} \mid \mathcal{A}_{m^2} \mid \mathcal{L}_{e'} \mid$

Figure 15: Boundary SymTFTs for $\mathbb{Z}_4$-symmetric irreducible (non-intrinsic) gapless phases. Diagram a) denotes $\mathbb{Z}_4$ gSSB phase with two universes while diagrams b) denotes $\mathbb{Z}_4$ gSPT phase.

conditions $\mathfrak{B}^{UV}$ and the set of IR boundary conditions $\mathfrak{B}^{IR}$, in much the same way as the bulk interface $\mathcal{A}$ provides a map between $\mathcal{S}$ and $\mathcal{S}'$-symmetric QFTs. In typical RG flows, the set of IR boundary conditions is strictly smaller than the set of UV boundary conditions, so the map $\mathcal{I}$ is not injective. We will see, however, that the opposite phenomenon can also take place: a single UV boundary condition can become non-simple in the IR and decompose into a multiplet under $\mathcal{S}'$. This in particular happens if the RG flow describes an intrinsically gapless SPT (igSPT) phase. This can arise if the group extension (194) is non-trivial. We will give an explicit example of this phenomenon, together with its physical interpretation, in the examples below.

**Example: gSPT and gSSB for $\mathbb{Z}_4 \to \mathbb{Z}_2$.** A standard and simple example is to consider $\mathcal{S} = \mathbb{Z}_4$ and $\mathcal{S}' = \mathbb{Z}_2$, with a trivial 't Hooft anomaly, where we will denote the $\mathbb{Z}_4$ symmetry generator as $P$ and $\mathbb{Z}_2$ symmetry generator as $P'$. This is achieved by condensing the bulk algebras

$$\mathcal{A}_{e^2} = 1 \oplus e^2, \quad \text{or} \quad \mathcal{A}_{m^2} = 1 \oplus m^2, \tag{195}$$

while fixing the symmetry boundary

$$\mathcal{L}_{\text{sym}} = \mathcal{L}_{\text{reg}} = 1 \oplus e \oplus e^2 \oplus e^3. \tag{196}$$

Condensing the algebra $\mathcal{A}_{e^2}$ is equivalent to describing a $\mathbb{Z}_2$ gSSB phase for the $\mathbb{Z}_4$ symmetry whereas condensing the algebra $\mathcal{A}_{m^2}$ amounts to depicting a gSPT phase [44, 45].

Let us determine the fate of the regular $\mathbb{Z}_4$ module category under RG flow. For this we must choose

$$\mathbf{Q}_2 = \mathcal{L}_{\text{reg}}, \tag{197}$$

and the two possible maps are determined by the IR multiples given by the choice of either

condensing $\mathcal{A}_{e^2}$ or $\mathcal{A}_{m^2}$:

$$\mathcal{L}_{\text{reg}} \otimes \mathcal{A}_{e^2} = 2\mathcal{A}_{e'}, \qquad \mathcal{L}_{\text{reg}} \otimes \mathcal{A}_{m^2} = \mathcal{A}_{e'}, \tag{198}$$

where the factor of 2 tells us there are two inequivalent choices of interface for $\mathcal{I}_a$ in contrast to $\mathcal{I}_b$ which only has one.

Starting with $\mathcal{A}_{e^2}$, the bulk theory splits into two $\mathbb{Z}_2$-symmetric universes (+) and (−) in the IR, exchanged by the $\mathbb{Z}_4$ symmetry line $P \sim m$ (projection of $m$ on $\mathcal{L}_{\text{reg}}$), denoting a gSSB phase. In each of these universes we have a doublet of boundary conditions for the $\mathbb{Z}_2$ symmetry $P'$ which is a projection of the bulk line $m'$ on $\mathcal{L}_{e'}$ or equivalently of $m^2$ on $\mathcal{L}_{\text{reg}}$, $P' \sim m' \sim m^2$. The map $\mathcal{I}_a : \mathcal{M}_{\text{reg}} \to \mathcal{M}'_a$ is described by

$$\begin{aligned}
(\mathfrak{B}^{1'})_+ &= \mathfrak{B}^1, & (\mathfrak{B}^{m'})_+ &= \mathfrak{B}^{m^2}, \\
(\mathfrak{B}^{1'})_- &= \mathfrak{B}^m, & (\mathfrak{B}^{m'})_- &= \mathfrak{B}^{m^3},
\end{aligned} \tag{199}$$

with the symmetry action

$$\begin{aligned}
P : \quad &(\mathfrak{B}^{1'})_+ \to (\mathfrak{B}^{1'})_- \to (\mathfrak{B}^{m'})_+ \to (\mathfrak{B}^{m'})_- \to (\mathfrak{B}^{1'})_+, \\
P^2 = P' : \quad &(\mathfrak{B}^{1'})_+ \leftrightarrow (\mathfrak{B}^{m'})_+, \qquad (\mathfrak{B}^{1'})_- \leftrightarrow (\mathfrak{B}^{m'})_-.
\end{aligned} \tag{200}$$

We can clearly see that $\mathcal{M}'_a \simeq (\mathcal{M}_{\mathbb{Z}_2})_+ \oplus (\mathcal{M}_{\mathbb{Z}_2})_-$ with $P^2 = P'$ being the natural action on the $\mathbb{Z}_2$ module category $(\mathcal{M}_{\mathbb{Z}_2})_\pm$ and $P$ providing a map between the two copies of $\mathcal{M}_{\mathbb{Z}_2}$.

Considering instead $\mathcal{A}_{m^2}$, the $\mathbb{Z}_2$ symmetry line $P'$ acts trivially in the IR of the single universe describing a gSPT phase and the regular representations map between each other through $\mathcal{I}_b : \mathcal{M}_{\text{reg}} \to \mathcal{M}'_b$ as

$$\mathfrak{B}^{1'} = \mathfrak{B}^1 \oplus \mathfrak{B}^{m^2}, \qquad \mathfrak{B}^{m'} = \mathfrak{B}^m \oplus \mathfrak{B}^{m^3}, \tag{201}$$

with $P$ symmetry action exchanging the two boundary conditions as

$$P : \quad (\mathfrak{B}^{1'}) \leftrightarrow (\mathfrak{B}^{m'}). \tag{202}$$

We can thus see that $\mathcal{M}'_b \simeq \mathcal{M}_{\mathbb{Z}_2}$ with $P$ providing the natural action on the $\mathbb{Z}_2$ module category.

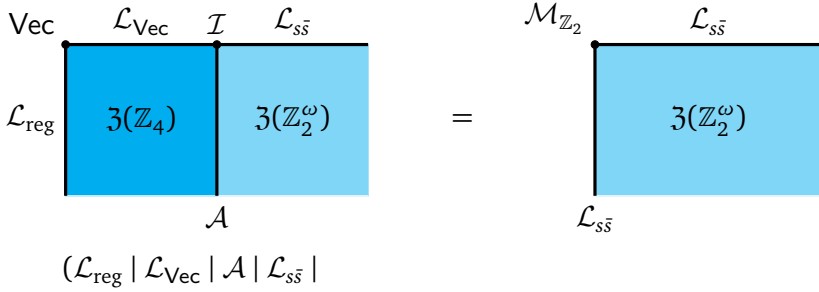

Figure 16: Boundary SymTFT for the $\mathbb{Z}_4$-symmetric intrinsically gapless phase (igSPT).

**Example: igSPT for $\mathbb{Z}_4 \to \mathbb{Z}_2^\omega$.** An enticing example of the splitting of boundary conditions along RG flows comes by considering a flow from a $\mathbb{Z}_4$-symmetric system to a $\mathbb{Z}_2^\omega$-symmetric one with nontrivial 't Hooft anomaly, implementing the non-trivial $\mathbb{Z}_4$ igSPT. In this case the condensable algebra is $\mathcal{A} = 1 \oplus e^2 m^2$ and $\mathfrak{Z}(\mathcal{S}')$ is the double semion model, whose nontrivial lines we again denote (as in section 3.4.2) by $s$ and $\bar{s}$, respectively. This model has a unique gapped boundary condition: $\mathcal{L}_{s\bar{s}} = 1 + s\bar{s}$. The symmetry boundary for $\mathbb{Z}_4$ is $\mathcal{L}_{\text{reg}} = 1 \oplus e \oplus e^2 \oplus e^3$ and we choose the boundary $\boldsymbol{Q}_2$ to be the singlet $\mathcal{L}_{\text{Vec}} = 1 \oplus m \oplus m^2 \oplus m^3$. The module category $\mathcal{M}$ is then given by Vec, with a single boundary state $\mathfrak{B}^0$. Then

$$
\begin{aligned}
\mathcal{L}_{\boldsymbol{Q}_2'} &= 1 \oplus s\bar{s}\,, \\
\mathcal{L}_{\mathcal{A}} &= 1 \oplus e^2 m^2 \oplus (em^3 + e^3 m)s \oplus (em + e^3 m^3)\bar{s} \oplus (e^2 + m^2)s\bar{s}\,,
\end{aligned}
\tag{203}
$$

or, in the folded picture

$$
\mathcal{L}_{\boldsymbol{Q}_2 \boxtimes \overline{\boldsymbol{Q}_2'}} = (1 \oplus m \oplus m^2 \oplus m^3)(1 \oplus s\bar{s})\,.
\tag{204}
$$

The intersection between $\mathcal{L}_{\mathcal{A}} \cap \mathcal{L}_{\boldsymbol{Q}_2 \boxtimes \overline{\boldsymbol{Q}_2'}} = 1 \oplus m^2 s\bar{s}$ gives rise to a module category $\mathcal{M}_{\mathbb{Z}_2}$ of two indecomposable objects, describing a doublet of boundary conditions $\mathfrak{B}^{1'}$ and $\mathfrak{B}^{P'}$ in the IR. Explicitly, the map $\mathcal{I}$ depicted in Fig. 16 is then $\mathcal{I} : \text{Vec} \to \mathcal{M}_{\mathbb{Z}_2}$ where

$$
\mathfrak{B}^0 = \mathfrak{B}^{1'} \oplus \mathfrak{B}^{P'}\,,
\tag{205}
$$

with the UV $P' = P^2$ line responsible for the splitting. This set of ideas can be developed in fuller generality and hopefully allow to address interesting physical boundary RG flows between (1+1)d symmetric CFTs. A prime candidate are current-current deformations of WZW models, whose rich phenomenology has been studied using integrability methods [90].

**An explanation for the splitting** The splitting of BC can be given an enticing physical interpretation in our setup. Consider, in the UV, a symmetry line $D_1^S$, $S \in \mathcal{S}_{\mathcal{M}}^*$, ending on the singlet BC, thus defining a topological junction $\varphi_{UV}^S$. We now assume that $S \in \mathcal{S}_{\text{gap}}$, so that it does not act on the gapless IR degrees of freedom. Under RG flow the topological junction $\varphi_{UV}^S$ is mapped to a (possibly trivial) topological local boundary operator $\varphi_{IR}^S$:

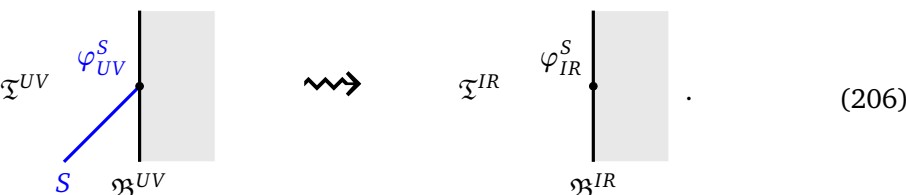

$$\tag{206}$$

We conclude that, from the IR perspective, the boundary condition $\mathfrak{B}_{\partial}^{IR}$ can split in many universes, depending on whether or not $\varphi_{IR}^S$ is trivial. The precise functor $\mathcal{I}_{IR}^{UV}$ implementing the mapping can be extracted from the SymTFT by studying the interface $\mathcal{I}$ described above, and in particular its number of indecomposable components. In the case of an igSPT we have shown that the interface $\mathcal{I}$ has two components, even if the original module cateogy has a single simple object. Indeed, as the IR symmetry $\mathcal{S}'$ has no Fiber Functor, we expect that $\mathfrak{B}^{\mathcal{M}}$ must split. This is a consequence of the fact that an anomalous symmetry does not admit an invariant BC [48, 91]. This immediately follows from the SymTFT picture, as for anomalous symmetries the module category $\mathcal{M} \boxtimes_{\mathcal{S}_{\mathcal{M}}^*} \mathcal{I}$ cannot have a single object due to the $\mathcal{S}'$ 't Hooft anomaly.

# 4 (1+1)d lattice models with boundary conditions

(1+1)d lattice models with fusion category symmetries $\mathcal{S}$ provide a UV description of categorical phases. The anyon chain models [67–75] provide a systematic way to construct such lattice models with any fusion category symmetry $\mathcal{S}$. In particular, the anyon chain allows for a systematic construction of Hamiltonians, whose ground states realize $\mathcal{S}$-symmetric phases – gapped and gapless [75]. Here we discuss how they can be extended to construct lattice models with boundary conditions transforming in arbitrary 1-charges of $\mathcal{S}$.

Let us first review the construction of the anyonic chain with periodic boundary conditions following [75], which we will then extend to anyonic chain on a segment. The construction begins with an input fusion category $\mathcal{C}^{12}$ and an input module category $\mathcal{M}$ of $\mathcal{C}$ such that the dual fusion category formed by $\mathcal{C}$-endofunctors of $\mathcal{M}$, $\mathcal{C}^*_{\mathcal{M}}$, is the symmetry fusion category $\mathcal{S}$ that one wants to consider, i.e.

$$\mathcal{S} = \mathcal{C}^*_{\mathcal{M}}. \tag{207}$$

One moreover chooses an arbitrary object $\rho \in \mathcal{C}$ that is not necessarily simple. The data so far fixes the Hilbert space of the model, with a canonical basis specified by all possible configurations of the form

$$\text{(208)}$$

where $m_i$ run over arbitrary simple objects of $\mathcal{M}$ and $\mu_{i+\frac{1}{2}}$ run over arbitrary morphisms in a fixed basis of $\mathrm{Hom}(m_i, \rho \otimes m_{i+1}) \in \mathcal{M}$. As the system is on a circle, the $m_1$ on the left is identified with the one on the right.

The Hamiltonian of the (1+1)d system is specified by choosing a morphism

$$h : \rho \otimes \rho \to \rho \otimes \rho, \tag{209}$$

which acts on two adjacent sites as

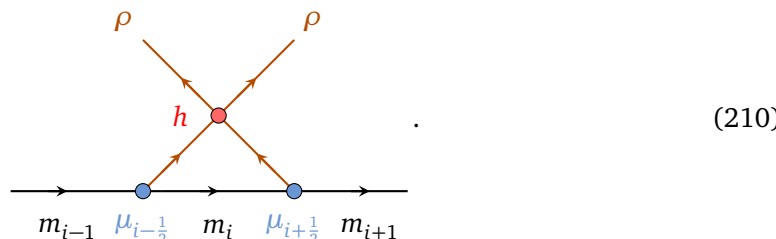

$$\text{(210)}$$

The full Hamiltonian is obtained by summing over the above map over all pairs of sites.

In the pictures of the model drawn above, $\mathcal{C}$ acts on $\mathcal{M}$ from the top. Correspondingly, $\mathcal{S}$ acts on $\mathcal{M}$ from the bottom. That is, the action of a symmetry $s \in \mathcal{S}$ on the Hilbert space of

---

$^{12}$This is not the symmetry category!

the model is obtained simply by fusing the $s$ line into $(m_i, \mu_{i+\frac{1}{2}})$ in the figure below

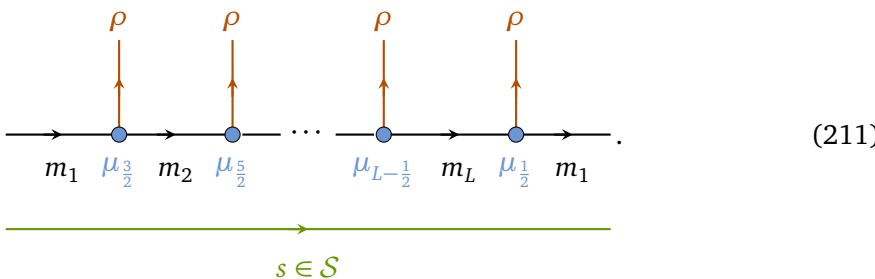

(211)

The fact that the action of symmetry commutes with the Hamiltonian is encoded in the fact that they act on $\mathcal{M}$ from the opposite (bottom and top respectively) directions.

**Boundary conditions.** We can include boundary conditions by defining the model on a segment rather than a circle. Let us say we want to have a BC transforming in 1-charge $\mathbf{Q}_2^L$ on the left, and a BC transforming in 1-charge $\mathbf{Q}_2^R$ on the right. These correspond respectively to some module categories $\mathcal{M}_L$ and $\mathcal{M}_R$ of $\mathcal{S}$. In particular, let us say that we want to construct a BC corresponding to a (not necessarily simple) object $m_L \in \mathcal{M}_L$ on the left and a BC corresponding to a (not necessarily simple) object $m_R \in \mathcal{M}_R$ on the right. The construction involves choosing (not necessarily simple) objects

$$\rho_L \in \mathcal{M}'_L, \qquad \rho_R \in \mathcal{M}'_R, \tag{212}$$

where $\mathcal{M}'_x$ for $x \in \{L, R\}$ is the module category for $\mathcal{S}^*_{\mathcal{M}_x}$ whose dual category is $\mathcal{C}$, i.e.

$$(\mathcal{S}^*_{\mathcal{M}_x})^*_{\mathcal{M}'_x} = \mathcal{C}. \tag{213}$$

Then the Hilbert space of the model is generated by configurations of the form

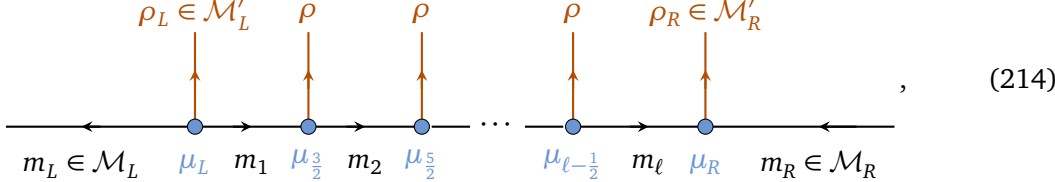

(214)

where $m_L$ and $m_R$ "stretch to infinity" in the above figure. $\mu_L$ and $\mu_R$ are 2-morphisms in the 2-category $\mathrm{Mod}(\mathcal{S})$ formed by module categories of $\mathcal{S}$, which physically is the 2-category formed by topological BCs of the 3d SymTFT $\mathfrak{Z}(\mathcal{S})$.

To describe the Hamiltonian, we need to choose boundary pieces $h_L$ and $h_R$ of the Hamiltonian along with the bulk piece $h$ discussed above. These are

$$\begin{aligned} h_L &: \rho_L \otimes \rho \to \rho_L \otimes \rho, \\ h_R &: \rho \otimes \rho_R \to \rho \otimes \rho_R, \end{aligned} \tag{215}$$

which should be viewed as 2-morphisms in $\mathrm{Mod}(\mathcal{S})$.

The $\mathcal{S}$ symmetry acts from below, and modifies $(m_L, m_R)$, thus changing this model with BCs to another model with BCs, where all the input data remains the same except for the modification of $m_L$ and $m_R$.

## Acknowledgments

We thank Lea Bottini and Alison Warman for discussions. CC thanks Lucia Cordova, Shota Komatsu, Kantaro Ohmori and Yifan Wang for discussions. We thank the author of [92–96] for coordinating submission on related results.

**Funding information**   LB is funded as a Royal Society University Research Fellow through grant URF\R1\231467. CC, and in part SSN, are supported by STFC grant ST/X000761/1. The work of SSN, is supported by the UKRI Frontier Research Grant, underwriting the ERC Advanced Grant "Generalized Symmetries in Quantum Field Theory and Quantum Gravity".

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
