# Peer review of "Boundary SymTFT"

_SciPost Physics, doi:SciPost Phys. 19, 061 (2025)_

## Round 1 · Referee Report · Anonymous (Referee 1) · 2025-6-24

Report

The authors have accounted for all of my comments on the first version. The paper can therefore be published in its present form.

Recommendation

Publish (easily meets expectations and criteria for this Journal; among top 50%)

---

## Round 1 · Referee Report · Anonymous (Referee 2) · 2025-7-16

Report

The authors have addressed all relevant points. The paper is ready to be published.

Recommendation

Publish (easily meets expectations and criteria for this Journal; among top 50%)

---

## Round 1 · Author Response

We thank both referees for their reviews and insightful comments. We reply to reach referee below.

Reply to referee 1:

We thank the referee for their comments and confirm all mentioned points have been addressed in the text.

Regarding a general higher-categorical framework, we state clearly what is known for fusion (d-1)-categories, module categories, their duals and Drinfeld centres, especially for n≥3.

Changes: A new paragraph entitled has been inserted near the end of the Introduction (p. 5). It now distinguishes the rigorous cases n=1,2 from the conjectural case n>2; cites all foundational papers mentioned: Etingof–Nikshych–Ostrik (2005), Ostrik (2003), Douglas–Reutter (arXiv 1812.11933), Décoppet (arXiv 2107.11037), and Bhardwaj–Décoppet–Schäfer-Nameki–Yu (arXiv 2408.13302); states explicitly that any claim for n>2 is presented only as a conjectural extrapolation but based on physical considerations.

Regarding the specific comments:

1) At instances like the list (2.11), “module categories” were changed to “indecomposable module categories” as requested.

2) Added footnote 3 with references that explains the internal-Hom construction invoked after (2.32) is well established for d=2 and d=3 but for d>3 the relevant mathematical theory remains unproven and is thus conjectural at this point in time.

3) Added footnote 2 that explains why the category N appearing in (2.33) is a module category over S^*_M.

4) Corrected equation (3.2) to reflect that Hom in this instance is not Hom_S but Hom_M as pointed out by the referee.

5) "Example: igSPT for Z_4 -> Z_2^omega" paragraph has been rephrased/corrected to ensure equation (3.169) is indeed a map, similar to "Example: gSPT and gSSB for Z4 -> Z2". Figure 16 has been updated to reflect the changes and make the map more explicit.

We also thank the referee for spotting some smaller typos which have been pointed out and now corrected.

With regards to some references appearing in the introduction but not later in the text, we believe these introductory citations serve as a concise “literature map.” In our field it is customary to open with a compact string of references that positions the paper within the broader program of higher-categorical TFT and symmetry. Each entry highlights a milestone—higher-form symmetries, categorical centers, anomaly inflow, lattice realizations, etc.—that motivated the present work, even though none is required for the subsequent proofs. We would therefore like to retain the full list for context, but we will shorten it if the journal’s style guidelines so direct.

Reply to referee 2:

We thank the referee for their comments and appreciate the positive assessment of our work. We also thank the referee for spotting the small grammatical slip in the sentence below Eq. (3.122): Line two below (3.122) – “It’s dimension” has been corrected to “Its dimension.”

All additional points referred to in Referee 1’s report have been fully addressed in the latest revision (see our reply to Referee 1 above for details).

We hope we have addressed all the comments of the referees and that the paper can now be recommended for publication.

---

## Editorial Decision

published